# Narrative Knowledge Weaver: A Multi-Agent Framework for Knowledge Graph Construction and Analysis from Complex Narratives

## Abstract

Long-form narratives such as screenplays and novels require reasoning over evolving characters, multi-stage events, and long-range temporal and causal structure. Although recent LLM-based methods can extract surface entities and relations, automatically induced knowledge graphs often lack the coherence and interpretability needed for narrative understanding and downstream tasks such as continuity checking or character timeline analysis. We introduce **Narrative Knowledge Weaver**, a multi-agent framework for constructing high-quality, human-readable knowledge graphs from complex narratives. The system combines adaptive schema induction, reflection-augmented extraction, and a normalization-before-merge pipeline that performs type refinement, scope convergence, and LLM-guided disambiguation. A dedicated module conducts **adaptive attribute enrichment** for narrative entities, aggregating multi-granular evidence and reflection-guided feedback to incrementally refine and expand schema-defined properties. An **event-centric refinement** stage further transforms raw event mentions into structured event cards and causally organized Event Plot Graphs (EPGs). All outputs are stored with fine-grained provenance and leveraged by a **tool-augmented reasoning agent** for temporal, causal, and structural queries. Evaluations on Re-DocRED, a NarrativeQA-derived benchmark, and a Practitioner Screenplay QA dataset show substantial improvements in entity normalization, relation accuracy, and event-level reasoning over strong baselines including EDC, Hybrid Retrieval, and GraphRAG. Beyond quantitative gains, the resulting graphs provide interpretable, application-ready representations of story worlds, supporting detailed analyses of narrative dynamics—from character states to causal chains and scene-level progression.

## 1 Introduction

Long-form narratives such as novels and screenplays contain rich, multiscale structure: characters evolve across scenes, events unfold through intertwined temporal and causal processes, and storylines span hundreds of pages (Ji et al., 2021; Huang et al., 2023). Current LLM-based systems struggle with these long-range dependencies, often losing coherence when reasoning across extended contexts.

Knowledge graph (KG) construction provides a promising avenue for organizing narrative information into explicit, queryable structures. Recent approaches—including sequence-to-KG generation (Huguet Cabot & Navigli, 2021), schema-light pipelines (Sun et al., 2024b; Mo et al., 2025), and multi-stage frameworks such as EDC (Zhang & Soh, 2024)—demonstrate significant progress. However, automatically induced KGs remain difficult to use for narrative reasoning: entities are often inconsistently typed or fragmented, relations loosely defined, and narrative dynamics blurred by community-level summarization. Such representations may suffice for broad retrieval, but they fail to support tasks that require explicit tracking of entity continuity, event progression, and causal structure.

A second challenge stems from the extraction process itself. Long-form narratives require coordinated, multi-step decisions—extraction, attribute assignment, disambiguation, temporal and causal

adjudication—that exceed what a single LLM pass can reliably perform. Recent work on agentic LLMs shows that stability improves when tasks are decomposed across specialized agents or tool-augmented workflows (Yao et al., 2023; Schick et al., 2023; Shen et al., 2023). Complementary studies emphasize the importance of *context engineering*—careful design of prompts, memory, and retrieval—to enable consistent behavior in long-context settings (Ji, 2025; LangChain, 2025). These trends suggest that narrative understanding requires not only structured representations, but also a *role-specialized, multi-agent construction process*.

Motivated by these observations, we aim to build narrative representations that are both *structurally coherent* and *human-readable*. In this work, we introduce **Narrative Knowledge Weaver**, a multi-agent framework that constructs high-quality knowledge graphs and leverages them for structured, tool-augmented reasoning. Our approach integrates:

1. **Narrative-tailored extraction and normalization:** a reflection-augmented pipeline for entity–relation extraction, adaptive schema probing, and a multi-stage normalization-before-merge process (type refinement, scope convergence, LLM-guided disambiguation) that yields globally coherent entities.
2. **Adaptive attribute enrichment for narrative entities:** a degree-aware and reflection-guided enrichment module that aggregates multi-granular evidence across all occurrences of an entity, incrementally refines and expands its schema-defined attributes, and produces coherent, interpretable, and provenance-aware entity profiles tailored to long-form narratives.
3. **Event-centric refinement for structured story modeling:** conversion of raw event mentions into structured event cards, followed by temporal and causal adjudication to form coherent *Event Plot Graphs (EPGs)* capturing plot-level structure.
4. **Provenance-aligned storage and tool-augmented reasoning:** unified graph, vector, and tabular backends linked via fine-grained `chunk_id` provenance, enabling reliable temporal, causal, and structural narrative QA.

Together, these components yield coherent, interpretable, and application-ready narrative graphs, supporting deeper temporal, causal, and character-centric reasoning than existing retrieval-based or community-level approaches.

## 2 RELATED WORK

### 2.1 LLM-BASED KNOWLEDGE GRAPH CONSTRUCTION

Early work on document-level information extraction emphasized cross-sentence reasoning and global coherence. Datasets and systems such as *DocRED* (Yao et al., 2019), *DyGIE++* (Wadden et al., 2019), and *OneIE* (Lin et al., 2020) demonstrated the benefits of joint modeling across entities, relations, and events. For cross-document scenarios, research on entity and event coreference highlighted the need for normalization before merging mentions (Barhom et al., 2019; Cattan et al., 2020), a practice we adopt in our normalization-before-merge pipeline for long-form narratives.

More recently, the advent of Large Language Models (LLMs) has catalyzed a shift towards more flexible and powerful knowledge graph construction (KGC) pipelines. A prominent line of work focuses on structured, multi-phase frameworks: the *Extract-Define-Canonicalize (EDC)* framework (Zhang & Soh, 2024) proposes a three-stage process of open information extraction, schema definition, and post-hoc canonicalization, while *Docs2KG* (Sun et al., 2024b) explores unified KGC from heterogeneous documents, and *KGValidator* (Boylan et al., 2024) introduces automated validation of extracted triples. Our framework extends this line of work by combining schema probing, disambiguation, and event-centric refinement into coherent multi-chapter graphs tailored to long narrative sources.

### 2.2 KNOWLEDGE-AUGMENTED GENERATION AND STRUCTURED RETRIEVAL

Retrieval-Augmented Generation (RAG) grounds LLMs in external knowledge (Lewis et al., 2020), with extensions such as Fusion-in-Decoder (FiD) improving evidence integration (Izacard & Grave, 2021). Multi-hop datasets like HotpotQA (Yang et al., 2018) further motivated structured retrieval methods for compositional reasoning. A major development is the integration of graphs with RAG,

exemplified by GraphRAG, which performs community-level organization and summarization to support retrieval (Edge et al., 2024). This has inspired graph-enhanced approaches (Peng et al., 2024; Zhu et al., 2025) and graph-grounded reasoning methods such as RoG (Luo et al., 2024) and ToG (Sun et al., 2024a). Complementary agentic systems like KG-Agent (Jiang et al., 2025) further highlight the value of structured knowledge in tool-based reasoning. Knowledge-Augmented Generation (KAG) contributes another key insight: explicitly tracking the provenance of supporting evidence (Liang et al., 2025).

Our framework draws from both lines of work, but focuses on enriching the graph at a finer granularity. For regular entity nodes, we attach structured attributes that evolve dynamically: the attribute set can expand, contract, or reorganize as the system integrates new, entity-specific evidence. Each attribute is further grounded with fine-grained `chunk_id` provenance to ensure interpretability and traceability. Events, in contrast, are treated as first-class narrative units: each event is converted into a structured event card and later refined through temporal and causal adjudication before being integrated into the Event Plot Graph (EPG). This design enables precise, provenance-aware retrieval while preserving the narrative structure necessary for temporal, causal, and plot-level reasoning.

### 2.3 AGENTIC FRAMEWORKS AND CONTEXT ENGINEERING

The agentic paradigm—casting LLMs as controllers of multi-step workflows—has advanced rapidly. Early systems such as *ReAct* (Yao et al., 2023) and *Toolformer* (Schick et al., 2023) demonstrated the benefits of interleaving reasoning with tool use, inspiring more complex multi-agent frameworks for collaborative tasks (Wu et al., 2023; Li et al., 2023; Qian et al., 2024).

Recent studies highlight that the effectiveness of such systems hinges on *Context Engineering* (Mei et al., 2025), which governs how information is written, selected, compressed, and isolated in the context window. Techniques include scratchpads and external memory (Nye et al., 2021; Packer et al., 2023), reflection-based summarization (Shinn et al., 2024), and role isolation via specialized agents.

Our framework applies these principles through reflection-driven memory, provenance-aware retrieval, and modular role separation across extraction agents. Further details appear in Appendix B.

## 3 THE NARRATIVE KNOWLEDGE WEAVER FRAMEWORK

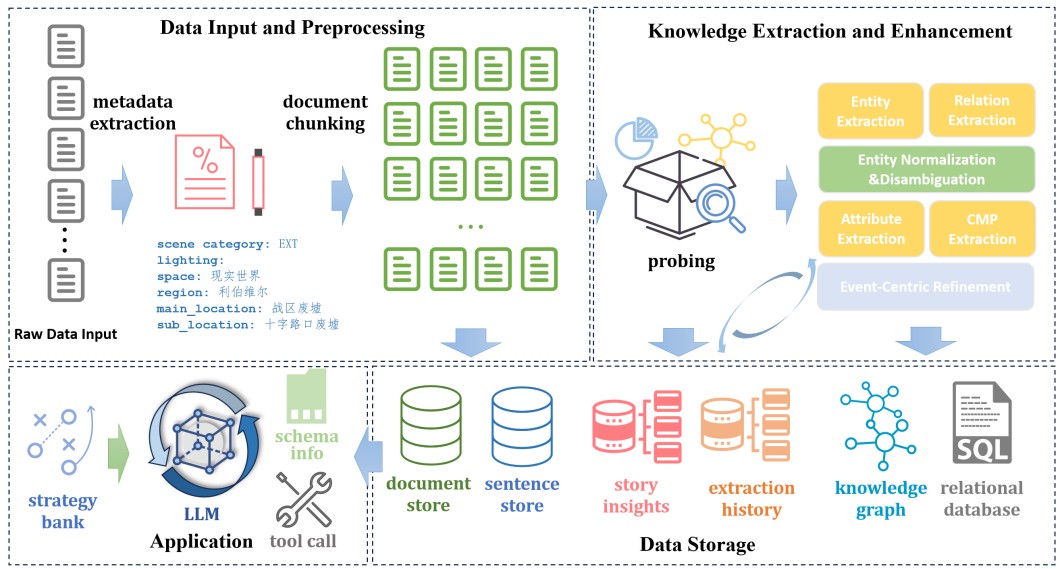

Figure 1: **Overall architecture of Narrative Knowledge Weaver.** The system is organized into four layers: *Data Input and Preprocessing*, *Knowledge Extraction and Enhancement*, *Data Storage*, and *Application*.

We propose **Narrative Knowledge Weaver**, a framework that transforms raw narrative text into multi-layered knowledge graphs for downstream reasoning. Figure 1 summarizes the system, which consists of a data preprocessing pipeline, a multi-agent knowledge extraction and enhancement module, a hybrid storage backend, and an application layer for tool-augmented reasoning.

## 3.1 Data Input and Preprocessing

Before knowledge extraction, raw narrative documents are first partitioned into discourse-coherent units and then annotated with continuity-aware summaries and structured metadata. This stage comprises two steps:

**Sliding Semantic Splitter.** Fixed-size chunking often disrupts narrative coherence by cutting across dialogues or stage directions. We therefore use a content-aware splitter that pairs a recursive text divider with a semantic boundary detector. Within a sliding window, preliminary spans are merged into a candidate text $t$, which is kept intact if $|t| < \tau$ or subdivided only at discourse-consistent boundaries (e.g., event or time shifts). Over-segmentation is limited by bounding the number of sub-segments $k$ and enforcing a minimum length $\ell_{\min}$. The resulting chunks remain LLM-friendly while preserving narrative units. The full algorithm and illustration appear in Appendix C.1.

**Continuity-Aware Summary and Metadata Extraction.** Each segmented unit (chapter or scene) receives a continuity-aware summary and structured metadata. Using a sliding-window setup, partition $i$ is summarized in 200 tokens conditioned on its content and all prior summaries, ensuring cross-unit consistency. This synopsis also guides domain-specific metadata extraction—locations, regions, and time for novels; and for screenplays, scene ID, INT/EXT category, lighting, and sub-location. Implementation details and prompt templates are provided in Appendix C.2.

## 3.2 Knowledge Extraction and Enhancement

his layer builds the narrative knowledge graph through four modules—adaptive schema induction, reflection-based extraction, entity normalization, and event refinement—forming a compact pipeline that incrementally stabilizes and enriches the story world.

### 3.2.1 Iterative Knowledge Extraction with Reflection

To construct the narrative knowledge graph, we employ a set of **reflection-augmented extraction agents**. Narrative text often contains fragmented evidence and partially specified descriptions, making single-pass extraction brittle. Each agent therefore operates under a recurrent extraction–reflection loop: an initial prediction is generated, immediately critiqued for schema adherence and evidential gaps, and then refined using structured feedback. This closed-loop mechanism stabilizes outputs over successive rounds and yields more complete, ontology-aligned representations.

**Reflection-Augmented Entity–Relation Extraction.** The **Knowledge Graph Extraction Agent** detects typed entities (e.g., `Character`, `Event`, `Location`), canonicalizes mentions, and instantiates relations drawn from a fixed ontology. A reflection module identifies type mismatches, invalid argument roles, and coverage gaps, enabling targeted revisions that produce ontology-consistent graphs even from noisy or sparse descriptions. Implementation details are shown in Appendix D.4.

- **Costume-Makeup-PropItems (CMP) Extraction:** For screenplay-specific properties (e.g., wardrobe, styling, props), a dedicated agent extracts structured scene-level details and stores them in a relational format.

**Adaptive Attribute Enrichment** We enrich the graph at the level of individual entities by attaching structured attributes only where they provide meaningful additional detail. A preliminary entity–relation graph is first constructed to compute node degrees. More than 85 percent of nodes have degree less than or equal to 2; these peripheral entities retain their extraction-time descriptions, which already suffice for grounding and retrieval. Only high-degree entities (degree greater than 2) and all Event nodes proceed to the enrichment workflow illustrated in Fig. 2.

For these selected entities, the Attribute Extraction Agent applies a reflection-guided process that aggregates all source chunks containing the entity, extracts passages that directly reference it, and merges them into a summary suitable for language model processing. Attributes predicted under the default schema are then evaluated by a reflection module, which determines whether the result is insufficient or mismatched. If refinement is needed, the agent processes the related passages block by block (around 600 tokens) and updates the attribute set in place. New attributes may be introduced, weak ones revised, and the schema itself can adapt dynamically to entity-specific evidence. Reflection feedback accumulates throughout, guiding the attribute set toward a stable and coherent representation.

This degree-aware and adaptive approach enriches central entities with precise, evidence-backed attributes while avoiding unnecessary processing for simpler nodes, producing a graph that is both efficient and closely aligned with narrative detail. (Implementation details are shown in Appendix D.6)

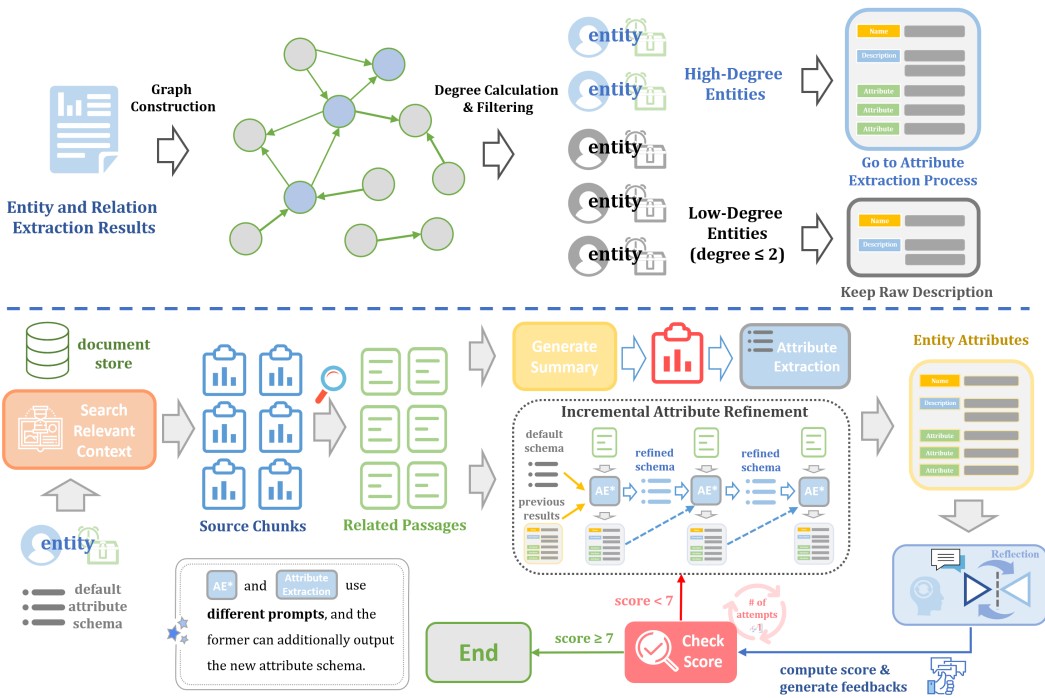

Figure 2: **Degree-aware entity enrichment.** Upper: selecting entities by node degree. Lower: adaptive workflow for assembling structured attributes.

### 3.2.2 ADAPTIVE SCHEMA INDUCTION VIA GRAPH PROBING AGENT

Narrative domains frequently deviate from fixed ontologies, giving rise to idiosyncratic entity types and unconventional relations. To accommodate such variability, we employ a **Graph Probing Agent** that induces a corpus-specific schema prior to large-scale extraction. Rather than adopting a static, manually designed ontology, the agent performs a small number of lightweight probe rounds that integrate corpus signals and reflection feedback from downstream extractors (§3.2.1) until the schema stabilizes.

**High-level process.** The agent draws on two complementary sources of contextual information: (i) coarse narrative scaffolding derived from chapter/scene summaries (§3.1); and (ii) dynamically sampled and reranked insights from the narrative text. Using these signals, it iteratively hypothesizes, evaluates, and refines candidate entity and relation types.

**Role in the full pipeline.** Each probe iteration triggers a set of trial extractions, collects schema-related feedback from the reflective modules of the extraction agents, and applies targeted updates.

Once stabilized, the resulting schema package—comprising refined type definitions, updated glossaries, and other contextual guidance—is passed to all downstream modules.

A complete description of the probing loop and its workflow diagram is provided in Appendix D.3.

### 3.2.3 ENTITY NORMALIZATION AND DISAMBIGUATION

Chunk–level extraction often yields heterogeneous and partially redundant entity candidates: a character may appear under multiple surface forms, while generic or ill-typed mentions can pollute the graph if merged prematurely. To bridge the gap between such noisy instance-level outputs and stable KG nodes, we introduce an *entity normalization and disambiguation* module that operates between local extraction and final graph construction.

We employ a three-stage *normalization–before–merge* pipeline that combines lightweight heuristics with LLM adjudication to regularize types and scopes and to resolve coreference into canonical entity identities.

1. **Entity type refinement.** We aggregate all observed type labels and apply simple pruning rules—e.g., (i) drop ACTION when {ACTION, EVENT} co-occur; (ii) drop CONCEPT when paired with a more specific label. Remaining ambiguities are resolved by an LLM that inspects local usage and assigns a narrative-consistent canonical type.

2. **Scope convergence.** Each entity is labeled as `global` (recurring) or `local` (scene-bound), which determines whether cross-scene merging is allowed. When scope disagrees across mentions, an LLM reviews evidence such as frequency, roles, temporal span, and co-participants to select a canonical scope.

3. **Name disambiguation.** After type and scope normalization, name–description embeddings are clustered via eigengap analysis (Appendix D.7) to produce coreference candidates. An LLM validates clusters using three rules: (i) avoid merging category-like groups; (ii) keep versioned identifiers distinct; (iii) choose canonical, well-formed names. The resulting rename map consolidates true aliases while rejecting spurious merges.

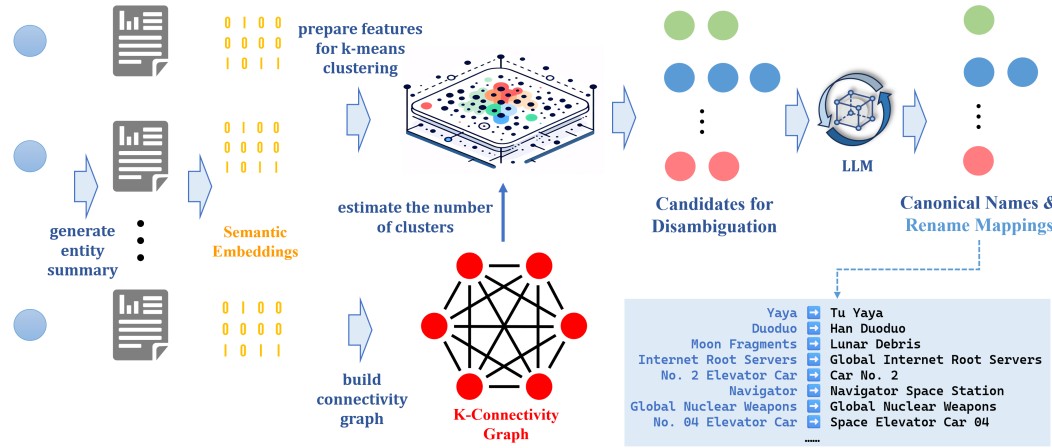

Figure 3: **Name disambiguation stage.** The normalization–before–merge module first refines entity types and scopes, then applies clustering and LLM adjudication to consolidate aliases while rejecting spurious merges (details in Appendix D.7).

### 3.2.4 EVENT-CENTRIC GRAPH REFINEMENT

Although extraction modules yield an initial entity–relation KG, narrative understanding requires modeling event dynamics and their aggregation into storylines. We therefore introduce an *Event-Centric Graph Refinement* pipeline that organizes raw events into a temporally and causally structured *Event Plot Graph (EPG)*. Inspired by community-level augmentation in GraphRAG (Edge

et al., 2024) but tailored to narratives, it constructs coherent event units that anchor causal reasoning and plot induction.

1. **Event card construction.** Each event is distilled into a compact *event card* containing participants, actions, temporal cues, outcomes, and provenance (Appendix B.4), providing stable and interpretable units for downstream reasoning.

2. **Candidate generation.** To reduce quadratic comparisons, candidate event pairs are restricted to the same community (Louvain) and further filtered by structural proximity (hop $\leq 3$) or semantic similarity ($\geq 0.7$).

3. **LLM-based causal adjudication.** Filtered pairs are classified into `CAUSES`, `INDIRECT_CAUSES`, `PART_OF`, or `None`, with temporal order, rationale, and confidence scores predicted for each (Appendix E.1).

4. **Graph pruning (SABER).** The initial causal graph is refined using **SABER**, which removes redundant or low-confidence edges when more coherent indirect paths exist, producing an interpretable DAG aligned with narrative logic (Appendix E.2).

5. **Promotion to plots.** High-confidence causal chains are aggregated, cleaned, and enriched with event-card context, then validated by an LLM against narrative criteria. Validated chains form plot units and inter-plot relations, yielding the final Event Plot Graph (Appendix E.3).

This pipeline transforms noisy event extractions into coherent, interpretable plots, establishing the Event Plot Graph as a stable backbone for downstream reasoning and QA.

## 3.3 Hybrid Storage and Application Layer

At the base of the architecture (Figure 1) is a hybrid data substrate paired with a tool-driven execution layer. Following the cross-indexing principles of KAG Liang et al. (2025), narrative information is organized across symbolic, semantic, and tabular representations, which the application layer exposes to the LLM through structured tools.

**Hybrid storage and cross-indexing.** The storage layer integrates three backends—a knowledge graph, a vector store, and relational tables—unified by a shared `chunk_id`. Each coherent text segment receives an embedding and `chunk_id` propagated to all entities, relations, events, and tabular records, enabling fine-grained bidirectional provenance. Beyond segments, the KG introduces `Scene`/`Chapter` supernodes that aggregate their constituent chunks, providing a coarse-to-fine index linking unit-level queries to chunk-level evidence and vice versa.

Naturally tabular information (e.g., wardrobe, props) is stored in relational tables, while the vector store maintains semantic memories such as insights and extraction histories. With all components aligned through `chunk_id`, the system supports seamless integration of graph traversal, semantic retrieval, and structured lookup within a single reasoning workflow.

**Application layer.** Sitting above the hybrid substrate, the application layer provides a tool-augmented interface for narrative reasoning. At inference time, the Tool-Augmented Reasoning Agent (TARA) composes tools from three families (Appendix H): graph utilities for entity lookup, relation/path queries, and plot exploration; vector utilities for chunk- and sentence-level retrieval; and relational utilities for SQL-style access to structured tables. By orchestrating these tools, TARA generates multi-step, provenance-traceable reasoning traces capable of answering temporal, causal, spatial, and attribute-focused questions.

## 4 Experiments and Results

### 4.1 Experimental Setup

**Datasets.** For **knowledge-graph construction**, we use the *Re-DocRED* dataset (Tan et al., 2022), a refined version of DocRED featuring improved annotation coverage and corrected coreference and logical inconsistencies. The schema contains six entity types: PERSON, ORGANIZATION, LOCATION, TIME, NUMBER, and MISCELLANEOUS.

For **Question Answering**, we evaluate on two benchmarks:

(i) a *NarrativeQA-derived* benchmark containing 20 screenplays and 10 novels (Kočiský et al., 2018), yielding 883 short-answer questions. Because NarrativeQA answers are typically *short phrases* rather than full sentences, retrieved evidence is passed through an additional *answer refinement prompt* that instructs the model to "based on the retrieved context, directly answer the question in a short phrase." This ensures comparability with the ground-truth answer format.

(ii) a **Practitioner Screenplay QA (Chinese)** benchmark (PSQA-CN; ours), containing 303 questions authored by film directors, screenwriters, and script supervisors. Questions span eight categories (Appendix H.5) covering localization, objects, character states, causal reasoning, timeline reasoning, and fine-grained production details (will be released after publication).

**Baselines.** For **KG construction**, we compare against: (i) a **Zero-shot LLM** extractor; (ii) the **EDC** framework (Zhang & Soh, 2024), which decomposes extraction into span detection, definition, and canonicalization; (iii) **GraphRAG** (Edge et al., 2024), using its KG induction module as a retrieval-oriented baseline under the Re-DocRED schema.

For **QA**, we evaluate three systems: (i) a **Hybrid Parent–Child Retriever** (**Hybrid Retriever**), combining BM25 and two-stage dense retrieval over parent and child segments; (ii) a **GraphRAG retrieval baseline** using the community-structured graph generated by its KG builder as query-time memory; (iii) our tool-augmented agent (**TARA**), which performs multi-step tool calling, evidence calibration, and iterative verification (Appendix A).

- All baselines use a **600-token chunk size**.
- All systems—including our Narrative Knowledge Weaver, GraphRAG, EDC, and the zero-shot LLM—use the same **Qwen3-235B-A22B-FP8** backbone for fairness.

**Evaluation Metrics.** For **KG construction**, we report Precision (P), Recall (R), and F1.

For **QA correctness**, we use an **LLM-based evaluator**. Given (question, answer, reference), we query the evaluator **five times** with independent sampling and report **majority-vote semantic correctness**. Evaluator consistency is high (Appendix F.4), indicating a stable evaluation signal. NarrativeQA additionally includes BLEU-1, BLEU-4, METEOR, ROUGE-L, and BERTScore. PSQA-CN responses are long and descriptive; therefore we report *only* LLM-evaluated correctness.

## 4.2 KNOWLEDGE GRAPH CONSTRUCTION

We evaluate KG construction under a *fixed* ontology aligned with Re-DocRED. To ensure comparability with EDC (Zhang & Soh, 2024), which does not model entity types explicitly, we disable Graph Probing and maintain a constant schema across all methods.

Table 1: KG construction under the fixed Re-DocRED schema.

| Method | Entity | | | Relation | | |
|---|---|---|---|---|---|---|
| | Recall | Precision | F1 | Recall | Precision | F1 |
| LLM-zeroshot | 0.7260 | 0.7337 | 0.7298 | 0.2637 | 0.4099 | 0.3209 |
| EDC | 0.6937 | **0.7897** | 0.7386 | 0.3369 | 0.4365 | 0.3803 |
| GraphRAG | 0.7897 | 0.1534 | 0.2569 | **0.3872** | 0.2715 | 0.3198 |
| NarrativeKnowledgeWeaver (Ours) | **0.8120** | 0.7190 | **0.7627** | 0.3360 | **0.5740** | **0.4239** |

NarrativeKnowledgeWeaver achieves the strongest overall performance. Compared with EDC, it yields higher entity recall without sacrificing precision and substantially higher relation precision with similar recall. GraphRAG, in contrast, exhibits high recall but very low precision—consistent with its coverage-oriented extraction strategy—leading to markedly lower F1.

## 4.3 QUESTION ANSWERING

We evaluate QA performance on the NarrativeQA-derived benchmark and the PSQA-CN benchmark, using a combination of automatic metrics and LLM-based pairwise comparisons.

**NarrativeQA.** Table 2 summarizes performance on the NarrativeQA-derived benchmark. TARA achieves the strongest overall results, with the highest LLM-judged correctness and competitive scores across all automatic metrics. GraphRAG performs particularly well on semantic overlap measures such as BLEU-4 and BERTScore, reflecting the strengths of its community-level aggregation. Hybrid Retriever provides a solid extractive baseline but lags behind on metrics that require deeper narrative understanding.

Table 2: QA results on the NarrativeQA-derived benchmark using automatic metrics and LLM-evaluated correctness (5-sample majority).

| System | BLEU-1 | BLEU-4 | METEOR | ROUGE-L | BERTScore | Correctness |
|---|---|---|---|---|---|---|
| TARA (Ours) | **24.10** | 4.95 | **37.80** | **45.70** | 41.85 | **76.4%** |
| GraphRAG | 23.85 | **5.02** | 36.10 | 44.20 | **42.10** | 71.2% |
| Hybrid Retriever | 23.40 | 4.78 | 29.95 | 39.85 | 40.10 | 63.8% |

**Practitioner QA (PSQA-CN).** Each question is answered five times with independent stochastic decoding, and we report both *question-level correctness* (majority-vote over the five samples) and *answer-level correctness* (per-sample accuracy). Details of evaluation procedures including prompts can be found in Appendix F. As shown in Table 3, TARA outperforms both baselines by a substantial margin under both metrics.

GraphRAG achieves relatively high question-level correctness (72.6%) but much lower answer-level correctness (61.3%), reflecting its high variance. This behavior is consistent with its *community* $\rightarrow$ *chunk* $\rightarrow$ *map* $\rightarrow$ *reduce* pipeline, where small fluctuations in partial-answer scoring or evidence distribution can lead to divergent final outputs. Hybrid Retriever shows far smaller variance (65.8%$\rightarrow$63.2%) due to its deterministic sparse–dense retrieval design and its tendency to produce short, extractive responses, but lacks the structured reasoning needed for temporal or causal queries.

TARA achieves the highest accuracy but also shows the largest gap between question- and answer-level correctness. Its agent-based reasoning selects tools and composes multi-step plans, leading to inherently variable trajectories. To improve stability, a lightweight *strategy library* (Appendix H.4) provides soft guidance on tool selection by question type, reducing variance without constraining capability.

Table 3: LLM-evaluated correctness on the PSQA-CN benchmark (5-sample majority).

| Method | Question Acc. | Answer Acc. |
|---|---|---|
| Hybrid Retriever | 65.8% | 63.2% |
| GraphRAG | 72.6% | 61.3% |
| TARA (Ours) | **89.8%** | **74.3%** |

**Pairwise LLM Preferences.** We further perform pairwise comparative evaluation across four criteria: *Comprehensiveness*, *Directness*, *Diversity*, and *Empowerment*. Figure 4 reports the resulting matrices. TARA is preferred across the more holistic criteria—reflecting its ability to synthesize multi-source evidence using structured tools—while Hybrid Retriever excels in Directness due to short extractive responses. GraphRAG generally lies in between, benefiting from structured retrieval but still hampered by variance in map–reduce inference.

**Tool Usage Analysis.** TARA operates over a unified tool layer combining graph utilities, semantic retrieval, and lightweight SQL-based lookups. Figure 5 shows tool usage distributions. Entity-centric tools dominate across datasets, confirming the importance of identifying characters and objects early in reasoning. A long tail of event-centric and structured utilities— though individually less frequent—plays a critical role in timeline reasoning, causal interpretation, and production-related queries. Overall, TARA dynamically integrates symbolic, semantic, and structured operations based on question needs.

Beyond the main results, we provide additional analyses in the appendix. These include ablations on sliding semantic splitting (Appendix G.5), adaptive attribute enrichment (Appendix G.3), event-

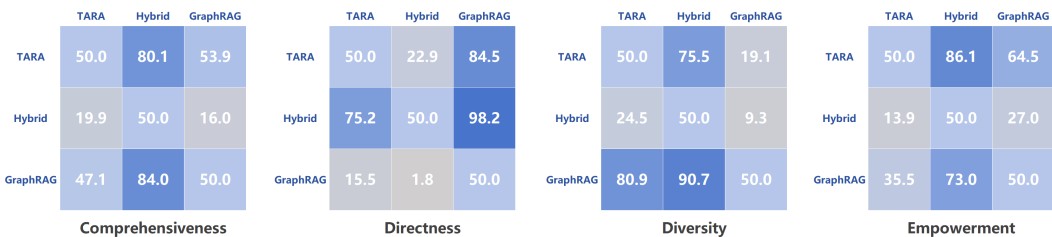

Figure 4: Pairwise LLM comparative evaluation across systems under four criteria: **Comprehensiveness**, **Directness**, **Diversity**, and **Empowerment**. Values denote mean head-to-head scores (0–100); 50 denotes a tie.

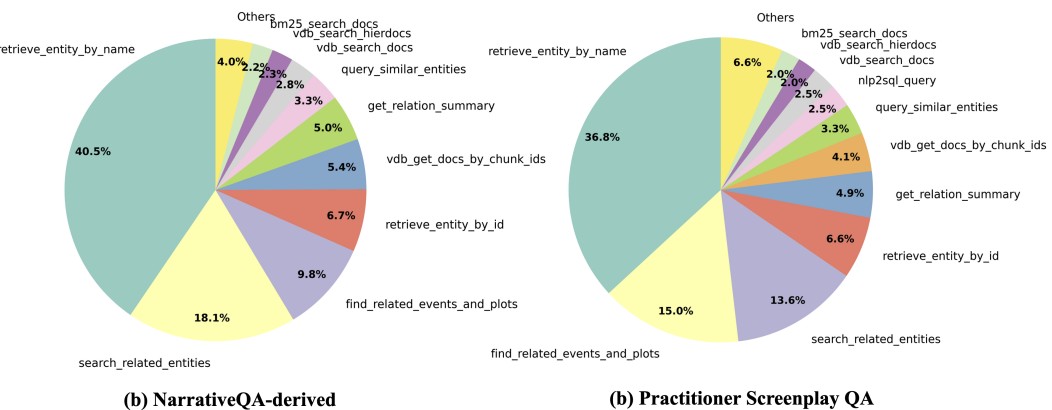

**(b) NarrativeQA-derived**          **(b) Practitioner Screenplay QA**

Figure 5: **Distribution of tool usage.** (a) NarrativeQA and (b) PSQA-CN benchmark

centric refinement (Appendix G.4), and low-frequency tools (Appendix H.3), as well as a detailed error analysis of low-correctness questions (Appendix H.6). We also present downstream applications enabled by the resulting narrative KGs, including production continuity checking and character–state extraction (Appendix I).

## 5 CONCLUSION

We presented **Narrative Knowledge Weaver**, a multi-agent framework for constructing coherent, human-readable knowledge graphs from long-form narratives. The system integrates reflection-augmented entity–relation extraction, adaptive schema induction, and a normalization-before-merge pipeline with an *adaptive attribute enrichment module* that dynamically updates entity profiles as new evidence appears, extending beyond the default attribute schema. Together, these modules yield stable, richly typed entities and structured event representations, which are further organized into causally grounded *Event Plot Graphs (EPGs)* for story-level reasoning.

Experiments show consistent improvements in KG quality, relation accuracy, and narrative question answering across NarrativeQA-derived data and practitioner screenplays. The interpretability and provenance of the constructed graphs also support applications requiring explicit narrative structure, such as continuity checking and character–state tracking (§I).

Despite these advantages, the system has two practical limitations. First, the multi-agent extraction pipeline incurs substantial token usage and runtime, especially for long and event-dense screenplays (Appendix G.1). Second, tool-augmented reasoning introduces stochasticity: different inference trajectories may select different toolchains, yielding variance in single-run answers. Both issues motivate future work. We plan to expand the **Practitioner Screenplay QA** benchmark with more fine-grained annotations, and to explore reinforcement learning to stabilize TARA's tool selection and multi-step plans, improving the scalability and reliability of narrative-grounded reasoning systems.

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

APPENDIX TABLE OF CONTENTS

# A  ADDITIONAL NOTES ON IMPLEMENTATION

**Use of Large Language Models.**   During the preparation of this paper, Large Language Models (LLMs) were used to [aid writing/polish text] and [assist in literature search]. No parts of the paper were solely generated by an LLM; all technical contributions, experiments, and analyses were designed and verified by the authors.

**Models and deployment.**   All language models are deployed *on-premises* to protect in-production screenplays. The primary backbone is **Qwen3-235B-A22B-FP8**, hosted with *vLLM* on four NVIDIA H20 GPUs (96 GiB each) using tensor parallelism, FP8 inference, KV caching, and batch scheduling for long contexts. We use **bge-m3** for dense embeddings and **Qwen3-Reranker-8B** for listwise reranking (experiments related to embedding selection is shown in G.6). No prompts, documents, or telemetry leave the secured environment.

**Agent runtime.**   The Tool-Augmented Reasoning Agent (TARA) is implemented with the *Qwen-Agent* framework[1], which provides dialogue state, function-style tool invocation, and decision tracing. Custom tools are registered through the framework's standardized interface; agent prompts, traces, and tool I/O are logged locally for auditability and reproducibility.

**Knowledge extraction pipeline.**   Entity, relation, and event extraction are orchestrated as a multi-agent pipeline with **LangGraph**, which coordinates agent nodes, retry budgets, and memory passing. This pipeline is used in our reflection-augmented extraction setting (Section 3.2) and elaborated in Appendix D.

**Retrieval stack.**   We adopt a hybrid retrieval design: sparse retrieval via **LangChain**'s BM25 components and dense retrieval over persistent vector stores, followed by listwise reranking with *Qwen3-Reranker-8B*. All retrieval steps attach provenance (e.g., chunk and scene identifiers) that is threaded through downstream reasoning and answer generation.

**Registered tools.**   Within the agent, we expose a curated set of tools implemented on top of *Qwen-Agent* (for registration/dispatch) and **LangChain** (for retrievers and SQL utilities). These cover: (i) sparse/dense retrieval at sentence and document granularity, (ii) vector-store search over hierarchical (parent–child) indices, and (iii) SQL accessors for structured production metadata. The tools follow uniform request/response contracts to ensure deterministic behavior and easy debugging (detailed interfaces are described later in the appendix  H).

**Graph store.**   Structured knowledge is persisted in **Neo4j**. Transactional access uses Cypher; analytics rely on Neo4j Graph Data Science (GDS), including reachability, cycle checks, weakly connected components, and transitive reduction used by SABER/causal pruning. We maintain versioned subgraphs for intermediate states and publish stabilized snapshots for QA and plot induction.

**Vector store.**   Semantic memories and document embeddings are stored in **Chroma**. We maintain persistent collections for (i) narrative chunks, (ii) extraction logs, and (iii) reflection memories, enabling targeted retrieval during revision while bounding long-context costs.

**Enhanced JSON handling.**   To make LLM tool calls robust, we add a lightweight *JSON reliability layer* that (i) corrects minor formatting issues, (ii) validates required fields with pluggable per-field validators, and (iii) invokes structured *repair prompts* with bounded retries when a response is invalid or incomplete. This layer guarantees that downstream components receive syntactically valid JSON (or an explicit error payload) and materially reduces failure cascades in multi-tool plans.

---

[1]https://github.com/QwenLM/Qwen-Agent

## B   DETAILS OF CONTEXT ENGINEERING

We group the context engineering components of our system into four categories, reflecting the strategies of *write*, *select*, *compress*, and *isolate* (Mei et al., 2025). (1) **Correction and reflection** introduces scoring, targeted feedback, and JSON repair to refine outputs while keeping error traces visible for future steps. (2) **Memory management** maintains extraction histories, insights, and feedbacks, enabling persistent scratchpad-style context that can be selectively retrieved. (3) **Global context management** stores background knowledge, terminology, and procedural rules as long-term shared memory across agents. (4) **Structured event representations** compress long narrative passages into reusable evidence units that support causal reasoning and plot-level inference. Together, these mechanisms ensure that the agent operates over tractable, verifiable, and contextually consistent state throughout narrative processing.

### B.1   CORRECTION AND REFLECTION

To ensure that all downstream components operate on structurally sound and semantically coherent outputs, our system combines two complementary components: a *correction module* that enforces strict JSON validity and a *reflection module* that evaluates extraction quality and provides targeted feedback for improvement.

**Correction.**   The correction module guarantees that every extraction stage produces a well-formed and schema-compliant JSON object. It follows a two-step pipeline. First, a deterministic formatter (`correct_json_format`) normalizes model outputs by standardizing quotes, stripping code blocks, repairing minor syntax issues, and preparing the structure for validation. Next, an enhanced validator performs *structured failure diagnosis* using a small taxonomy of common issues ("empty response", "invalid after correction", "missing required fields", "invalid field values"). It checks required fields, applies field-level validators, and determines whether the output is recoverable. If validation fails, the system activates `process_llm_response_with_retry`, which constructs a task-specific repair prompt (covering entities, relations, attributes, reflection logs, and causal checks) and performs a bounded number of LLM repair attempts, validating each candidate in turn. Regardless of whether repair succeeds, the module *always returns* a deterministically formatted JSON string, preserving error traces and the last corrected content for provenance.

**Reflection.**   Where correction enforces structural validity, the reflection module evaluates the *quality* of a valid extraction. It scores intermediate outputs along dimensions such as accuracy, consistency, and redundancy, assigns a discrete quality score (0–10; rubric in Table 4), and records narrative-level insights. Its checks cover both instance-level issues (e.g., invalid triples, duplicate or ambiguous entities) and schema-level diagnostics (coverage gaps, type–boundary confusions, incorrect direction or arity), but without proposing unconstrained schema edits. If the score is high ($\geq 8$), the module may intentionally leave narrative insights empty to avoid overfitting; otherwise, it populates a set of explicit repair points (e.g., incorrect relation types, duplicate mentions, missing core entities) and enriches the output with story-aware observations such as character dynamics or key event developments. These feedback traces enable *targeted retries* rather than full re-processing, improving quality with minimal token overhead.

Table 4: Reflection score rubric (0–10 scale).

| Score | Interpretation |
|---|---|
| 0 | Extraction failed or logs missing; unusable output |
| 3 | Partially usable but error-prone; requires re-extraction |
| 5 | Acceptable quality; minor issues present |
| 7 | Good quality; rare inconsistencies without critical errors |
| 10 | Excellent quality; no actionable flaws |

Together, correction and reflection implement a context-engineering strategy that stabilizes the extraction pipeline. Correction ensures that every step produces syntactically reliable, schema-compatible JSON; reflection provides selective, content-level feedback that directs when and how retries occur. Instead of discarding faulty generations, the system isolates errors, preserves their

provenance, and incrementally improves extractions while keeping context churn and token usage low.

## B.2 Memory Management

To stabilize iterative extraction, we employ a memory-augmented controller that records both evidence-level traces and higher-level reflections, ensuring that subsequent steps are grounded in accumulated experience rather than treated as isolated generations. Our design integrates two complementary mechanisms: a **DynamicReflector** for the extract–reflect–revise loop, and a probing-oriented memory layer that supports schema induction and pruning.

**Dynamic reflection.** The DynamicReflector maintains three stores together with a reranker. (i) **History memory** logs sentence-level evidences annotated with extracted entities, relations, and reflection scores, enabling provenance-aware retrieval. (ii) **Insight memory** accumulates narrative-level observations such as story development cues, character interactions, or emotional shifts, which transcend individual sentences and promote coherent revisions. (iii) **Entity extraction memory** is a lightweight dictionary that tracks prior mentions of entity names to stabilize naming and surface coreference links. (iv) A **reranker** rescores candidate recalls and filters them by relevance. On the write path, reflection outputs (scores, issues, insights) are decomposed into sentence-level audit notes and stored in the history memory, while high-level insights are preserved in the insight memory and normalized entity mentions are recorded in the entity extraction memory. On the read path, the agent queries these stores before the next extraction attempt; high-scoring examples (typically $\geq 7$) are highlighted as positive references, while low-scoring cases ($\leq 5$) serve as negative counterexamples. This selective recall improves stability and traceability, with reranking ensuring that retrieved context remains compact and focused.

**Probing-oriented memory.** In addition to the extraction loop, we incorporate memory into a schema probing workflow that adaptively refines the set of entity and relation types. Here, the agent retrieves prior **insight memory** entries relevant to background, abbreviations, entity schema, and relation schema, and reranks them to construct task-specific conditioning contexts. After each batch of test extractions, frequency distributions over entity and relation types are aggregated, and types falling below configurable thresholds (`entity_prune_threshold`, `relation_prune_threshold`) are flagged as pruning candidates. To avoid premature deletion, the system applies guardrails such as core type whitelists and feedback summaries before pruning. Reflection results and summarized feedbacks are fed back into the insight memory, providing a persistent record of schema evolution. This probing-oriented memory layer thus enables schema convergence by combining evidence accumulation, frequency-based pruning, and reflective consolidation. The full probing pipeline and its interaction with memory are illustrated in Figure 8.

**Pipeline integration.** As illustrated in Figure 9, memory management underpins both entity–relation extraction and schema probing. In the extraction loop, the sequence is: (1) generate logs and reflect; (2) write reflection outputs into the memory stores; (3) retrieve evidences and insights to enrich the next round; (4) retry selectively until convergence or budget exhaustion. In schema probing, the sequence is: (1) retrieve prior insights to condition background and schema updates; (2) generate candidate schema with feedback integration; (3) test extractions and aggregate distributions; (4) prune rare types under thresholded criteria; (5) reflect on schema quality and iterate until the score threshold is met.

By separating evidences from insights, attaching structured scores, and applying probing-time pruning with guardrails, the memory management module ensures that context remains verifiable, reusable, and tractable throughout narrative knowledge extraction.

## B.3 Global Context Management

Beyond transient feedback and reflection, our system relies on a layer of **global context management** that provides long-term shared memory across agents. This layer maintains background knowledge, terminology, procedural rules, and schema definitions that remain stable throughout the pipeline, ensuring coherence and consistency across multiple extraction and probing cycles.

**Procedural rules.** A set of general extraction rules is globally injected into every stage, serving as persistent guardrails for entity and relation identification. These rules enforce principles such as: avoiding category names or pronouns as entities; requiring clear semantics, directionality, and validity for relationships; restricting relation types to a closed set of enumerated labels; prohibiting the introduction of unseen entities in relation extraction; prioritizing core entities relevant to narrative progression; and excluding narrative techniques (e.g., montage, subtitles, transitions) from being treated as entities or events. They also regulate entity naming (e.g., stripping titles or modifiers to recover the core character), require entity descriptions, and enforce a semantic scope distinction between *global* recurring entities and *local* one-off mentions. These constraints act as structural invariants that reduce hallucinations, ensure cross-agent consistency, and promote extractability into a verifiable schema.

**Background and terminology.** Global memory also stores narrative background information and domain-specific abbreviations. During the probing phase, these elements can be updated or refined: additional insights retrieved from memory may adjust background descriptions or extend the abbreviation set. Once the schema converges, however, background and terminology are frozen for the main extraction stage, ensuring that all agents operate over a stable and consistent knowledge base.

**Schema definitions.** Schema definitions serve as the canonical vocabulary of the system. They specify entity types such as `Character`, `Event`, `Action`, `Object`, `Concept`, `Location`, `TimePoint`, and `Emotion`, and relation groups capturing semantic, spatiotemporal, causal, and role-specific links. In the probing stage, schema is iteratively adjusted: new candidate types may be proposed, rare or noisy types pruned, and descriptions refined. In the main extraction stage, the schema is locked as a fixed reference, preventing drift and enabling reproducible, verifiable knowledge graph construction.

**Integration.** Global context management complements reflection-based memory (subsection B.2) by providing an enduring layer of shared rules, background knowledge, terminology, and schemas. During probing, these elements remain adaptive, shaped by evidence and feedback (Figure 8); during extraction, they are frozen, ensuring aligned conventions across agents and preventing schema drift in long pipelines. In implementation, the global prompt is instantiated with background rules and terminology, while reflection outputs such as prior suggestions, identified issues, and scored results are selectively folded back into the context. This design guarantees that each round of extraction is simultaneously grounded in stable conventions and informed by dynamic evidence, yielding both consistency and adaptability across extended pipelines.

### B.4 STRUCTURED EVENT REPRESENTATIONS

To provide consistent and context-rich inputs for causal reasoning, each event is distilled into a **structured event card**. An event card encodes a compact summary, time hints, participants, actions, outcomes, and one supporting evidence sentence, derived jointly from the knowledge graph and source text. This construction exemplifies our *context engineering* strategy: by filtering context to the minimal but sufficient attributes of an event, cards reduce noise and hallucination while preserving verifiable grounding.

**Design.** Each card is designed as a lightweight, auditable record of an event. It balances expressivity and compactness: fields such as `name`, `summary`, and `action` provide semantic clarity, while `time_hint`, `locations`, `participants`, and `outcomes` capture the minimal situational context. The `evidence` field anchors the card to a verbatim sentence, ensuring reproducibility and traceability.

**Integration.** Event cards serve as standardized evidence units across stages of reasoning. In causality judgment, pairs of cards are evaluated together to decide temporal precedence and causal relations. In plot abstraction, chains of cards are grouped into coherent storylines. Because all higher-level judgments are grounded in event cards, downstream graphs remain interpretable, verifiable, and reproducible.

**Efficiency.** Event cards are precomputed in parallel and cached as persistent artifacts, avoiding redundant generations and stabilizing provenance across iterations. This ensures that causal and plot-level inferences can be scaled without repeated raw text processing, while every decision remains auditable through its underlying card.

---

**Event Card Construction Prompt Template**

You are an event graph modeling engineer. Your task is to construct a standardized event card from the provided information.

Rules (must follow):

- Consider only evidence directly describing the target event or its attributes such as time, location, participants, action, and outcomes.
- If multiple levels of granularity appear, preserve the description that best represents the event itself and ignore unrelated super- or sub-events.
- Express the action clearly as a verb phrase.
- Extract explicit outcomes if available; otherwise leave empty.
- Select the most direct textual evidence sentences.

Target Event Information:
{EVENT_INFO}
Related Context (may include noise):
{RELATED_CONTEXT}

Return the result strictly in the following JSON format:

```
{
  "event_card": {
    "name": "Event name (from input or inferred)",
    "summary": "Concise evidence-based summary",
    "time_hint": "unknown | relative phrase | time span",
    "locations": ["Location1", "Location2"],
    "participants": ["Actor1", "Actor2"],
    "action": "Main action verb phrase",
    "outcomes": ["Result1", "Result2"],
    "evidence": "Most direct evidence sentence (verbatim)"
  }
}
```

---

**ID: ent_417068**

- **Name:** Hao Xiaoxi reminds Zhou Zhezhi to go on stage
- **Summary:** Hao Xiaoxi reminds Zhou Zhezhi to prepare for the stage and check materials.
- **Time:** 50 hours before the Moon's disintegration
- **Location:** Small meeting room
- **Participants:** Zhou Zhezhi, Hao Xiaoxi
- **Action:** Reminder to prepare for going on stage
- **Outcomes:** Zhou Zhezhi begins preparing his speech and notices the surveillance camera
- **Evidence:** Hao Xiaoxi reminds Zhou Zhezhi to go on stage and confirm materials are ready; Zhou subsequently notices the surveillance camera.

Figure 6: **Event cards construction.** Each event is distilled into a compact card encoding summary, time hints, participants, action, outcomes, and supporting evidence. Standardized cards provide minimal yet sufficient inputs for causal classification and plot promotion, reducing noise and aiding reproducibility.

# C  IMPLEMENTATION DETAILS OF NARRATIVE-AWARE PREPROCESSING

Before knowledge extraction, raw narrative documents are normalized through a preprocessing pipeline that ensures both discourse alignment and structured context enrichment. The pipeline consists of two main stages: (i) *semantic chunking* to produce discourse-consistent text chunks, and (ii) *summary-guided metadata extraction* to attach continuity-aware synopses and standardized fields. Figure 7 illustrates these two stages.

## C.1  SEMANTIC CHUNKING

To prepare narrative documents for extraction, we first segment them into discourse-consistent units. Naïve fixed-size chunking often cuts across dialogues, stage directions, or coherent paragraphs, producing incoherent inputs for LLMs. To avoid this, we combine a recursive text splitter with a boundary detector that is sensitive to temporal and event shifts. Segments are merged within a sliding window into candidate text $t$. If $|t| < \tau$, the candidate is retained; otherwise, $t$ is subdivided at discourse-consistent breakpoints. Segmentation is further constrained by a maximum number of sub-segments $k$ and a minimum length $\ell_{\min}$ to prevent over-fragmentation.

The procedure is summarized in Algorithm 1.

---

**Algorithm 1** Sliding Semantic Splitting

---
1: **Input:** Preliminary segments $\mathcal{S}$; max sub-segments $k$; threshold $\tau$
2: **Output:** Refined segments $\mathcal{S}'$
3: Initialize $\mathcal{S}' \leftarrow [\,]$, carry segment $c \leftarrow \emptyset$
4: **for** each $s_i \in \mathcal{S}$ **do**
5:     $t \leftarrow c \,\|\, s_i$
6:     **if** $|t| < \tau$ **then**
7:         Append $t$ to $\mathcal{S}'$; $c \leftarrow \emptyset$; **continue**
8:     **end if**
9:     $\ell_{\min} \leftarrow |t|/k$
10:    subs $\leftarrow$ Splitter$(t, k, \ell_{\min})$
11:    Append subs$[:-1]$ to $\mathcal{S}'$; $c \leftarrow$ subs$[-1]$
12: **end for**
13: **if** $c \neq \emptyset$ **then**
14:    Append $c$ to $\mathcal{S}'$
15: **end if**

---

**Semantic segmentation prompt.** In practice, we implement boundary detection via an LLM prompt that asks for semantic breakpoints (e.g., shifts in time, events, or characters). The model outputs strictly formatted JSON, ensuring safe parsing and concatenation for downstream processing.

---

**Semantic Paragraph Segmentation Prompt**

You are a language expert skilled in text structure analysis. Your task is to semantically and logically split a longer narrative text.

**I. Task Description**

- Input: a narrative text segment (e.g., novel passage, screenplay excerpt).
- Identify semantic or plot breakpoints and divide into no more than {MAX_SEGMENTS} paragraphs.
- Each paragraph should be semantically complete (e.g., time progression, event transition, character relationship change, narrative focus shift).
- The last paragraph may be an incomplete fragment, reserved for concatenation with subsequent text.
- Each fragment should contain at least {MIN_LENGTH} words.

**II. Output Format** Return strict JSON:

{

---

```
  "segments": [
    "First semantically complete paragraph",
    "(Optional) Second semantically complete paragraph",
    ...,
    "Last paragraph (can be a fragment or empty string)"
  ]
}
```

**III. Additional Requirements**

- If no natural split is possible, return only one paragraph and set others as empty strings.
- Do not add explanations or formatting outside JSON.
- All paragraphs must be strings to ensure valid JSON parsing.

**IV. Text to be Processed**

`{TEXT}`

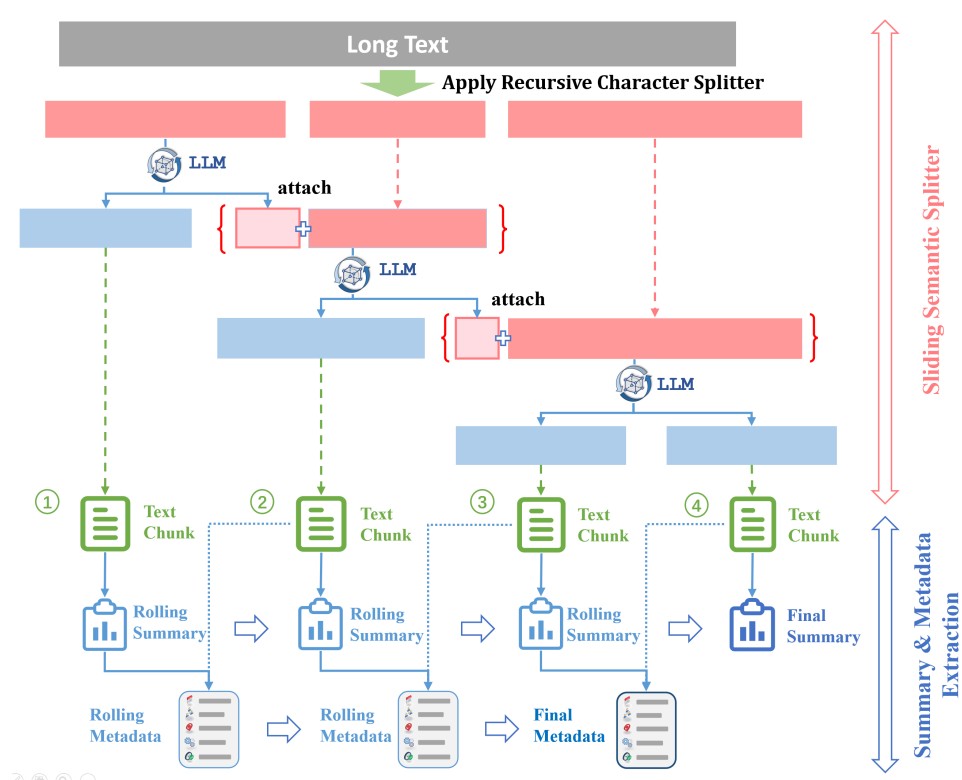

Figure 7: **Narrative-aware preprocessing.** Raw text is first segmented into discourse-consistent units through recursive splitting and LLM-based boundary detection. Each unit is then enriched with a sliding-window summary and structured metadata (e.g., scene ID, location, lighting), yielding narrative-aligned chunks for downstream KG construction and QA. For ablation results on the sliding semantic splitter, see Appendix G.5.

## C.2   SUMMARY AND METADATA EXTRACTION

Each segmented unit (a chapter or scene) is then augmented with a continuity-aware summary and standardized metadata. We use a *sliding-window summarization* strategy: for partition $i$, the model generates a $\sim$200-token synopsis conditioned on both the current chunk and the concatenated summaries of preceding ones. This lightweight form of *context engineering* (Ji, 2025) preserves long-range coherence and provides concise synopses for downstream retrieval and QA.

**Summary extraction prompt.** We use the following prompt to generate short continuity-aware summaries. The output must be a single coherent synopsis ($\leq$ 200 tokens) without speculation.

---

**Continuity-Aware Summary Extraction Prompt**

You are tasked with generating a short continuity-aware synopsis of the following text.

**Inputs**
Current chunk: `{CHUNK}`
Cumulative summary of previous chunks: `{HISTORY_SUMMARY}`

**Instructions**

- Write a single coherent summary ($\leq$ 200 words).

- Maintain narrative continuity by integrating key information from both the current chunk and the provided history.

- Do not speculate or invent unseen content.

- Use concise natural language suitable for downstream retrieval.

**Return strictly the following JSON**

```
{
   "summary": "a single coherent summary (no more than 200 words)"
}
```

---

**Screenplay scene metadata prompt.** For screenplays, we additionally parse standardized metadata. The inputs are the `title` and the sliding-window `summary`. The output must strictly follow the JSON schema.

---

**Screenplay Scene Metadata Extraction Prompt**

You are tasked with parsing a screenplay scene title (with optional continuity-aware summary) and extracting standardized metadata.

**Inputs**
Title: `{TITLE}`
Summary: `{SUMMARY}`

**Fields and Rules**

- `scene_id`: scene number, usually at the beginning of the title (e.g., "71", "17-13"); can be null.

- `scene_category`: only `"INT"`, `"EXT"`, or `null`.

- `lighting`: only `"Day"`, `"Night"`, `"No Day-Night"`, or `null`.

- `space`: `"Real World"`, `"Digital Space"`, `"Dream"`, or `null`.

- `region`: larger area (e.g., "New York City", "Paris").

- `main_location`: primary location (e.g., "Apartment", "Police Station", "Hospital").

- `sub_location`: specific sub-location (e.g., "Living Room", "Interrogation Room", "Emergency Ward").

- `summary`: short purpose/atmosphere phrase ($\leq$ 20 chinese characters/english words) or null.

**Return strictly the following JSON**

```
{
   "metadata": {
      "scene_id": "71 or 17-13 or null",
      "scene_category": "INT / EXT / null",
      "lighting": "Day / Night / No Day-Night / null",
      "space": "Real World / Digital Space / Dream / null",
      "region": "Region information or null",
      "main_location": "Main location or null",
      "sub_location": "Sub location or null",
      "summary": "One-sentence summary or null"
   }
}
```

---

## D  IMPLEMENTATION DETAILS OF MULTI-AGENT KNOWLEDGE EXTRACTION FRAMEWORK

This appendix details our multi-agent framework for knowledge extraction and enhancement, designed as a collaborative workflow with a clear division of labor. We first provide an architectural overview and then elaborate on the two core agents—the **Graph Probing Agent** and the **Knowledge Graph Extraction Agent**—highlighting their dynamic interaction which enables both instance-level refinement and global schema co-evolution. This structure directly addresses how feedback is shared and reconciled, demystifying the agent interaction process.

### D.1  ARCHITECTURAL OVERVIEW: A DIVISION OF LABOR

Our framework operates as a multi-agent system where each agent has a specialized role, contributing to a larger, cohesive workflow. The primary agents include:

- **Graph Probing Agent**: The "schema architect" of the system. Its responsibility is to define, evaluate, and iteratively evolve the global graph schema (i.e., the types of entities and relations). It operates at a macro level, ensuring the schema remains robust and well-adapted to the narrative domain.
- **Knowledge Graph Extraction Agent**: The "field worker" that performs fine-grained extraction from individual text chunks. It operates at the instance level, adhering to the schema provided by the `GraphProbingAgent` and striving for high-quality, compliant extractions.
- **Attribute Extraction Agent**: A "profiler" that enriches entities created by the extraction agent with detailed, schema-defined attributes, adding depth and context.
- **Other Specialized Agents**: Additional agents, such as the `CMP Extraction Agent` for production-specific details (costumes, props) and modules for entity disambiguation, handle other downstream tasks.

While these agents function with a degree of autonomy, they are not isolated. The linkage between the `GraphProbingAgent` and the `KnowledgeGraphExtractionAgent`, mediated by a shared memory system, forms a critical feedback loop that is central to the framework's adaptability and performance.

### D.2  THE TWO-TIER REFLECTION FRAMEWORK

To demystify the "black box" of agent interaction, our framework is built upon a **two-tier reflection architecture**. This design moves beyond isolated, single-pass extractions by establishing a structured process for how agents learn from their actions and how this learning is shared and reconciled globally. The entire process is orchestrated by a central memory component, the `DynamicReflector`, which facilitates two distinct but interconnected levels of reflection.

**The Dynamic Reflector: A Shared Long-Term Memory.**  The `DynamicReflector` serves as the central nervous system for knowledge sharing, preventing each extraction task from being an isolated "cold start." It maintains two persistent, vector-indexed memories:

- **History Memory**: Archives concrete extraction exemplars, linking source text to extracted triples and their quality scores. It functions as a dynamic casebook of successful ("positive examples") and unsuccessful ("negative examples") extractions.
- **Insight Memory**: Stores high-level, abstract knowledge, such as story-level insights (e.g., character relationships, plot points) generated by agents during reflection.

This shared memory allows any agent to query and learn from the collective experience of the entire system, making the feedback-sharing process transparent and reproducible.

**Instance-Level Reflection: Bottom-Up Feedback Generation.**  This first tier of reflection occurs within each `KnowledgeGraphExtractionAgent` and constitutes the bottom-up portion of our feedback loop. It follows an internal *extract–reflect–revise* cycle designed to maximize the quality of local outputs while generating valuable feedback for the global system.

- **Extract**: Following a top-down directive, the agent performs an initial extraction based on the current global schema provided by the `GraphProbingAgent`.
- **Reflect**: A critic module then evaluates this output. Using a specialized prompt, it assesses compliance and accuracy. Crucially, it also identifies fundamental mismatches between the text and the schema, generating two forms of feedback: (1) immediate, instance-specific corrections for revision, and (2) higher-level suggestions for schema improvement, which are logged as `current_issues`.
- **Revise**: If the quality score is below a threshold, the agent revises its extraction. It conditions its revision on both the critic's direct feedback and on relevant historical examples retrieved from the `DynamicReflector`.

This instance-level loop guarantees high-quality local extractions and serves as the primary source of the raw, bottom-up feedback that fuels schema evolution.

**Schema-Level Reflection: Top-Down Strategy Evolution.** The second tier is managed by the `GraphProbingAgent`, which reflects upon the system's performance at a macro level to guide the co-evolution of the global strategy. This top-down process aggregates and makes sense of the distributed feedback generated by the instance-level agents.

- **Aggregate Distributed Feedback**: The `GraphProbingAgent` first dispatches numerous `KnowledgeGraphExtractionAgent` instances to perform "probing" extractions. It then centrally collects all the schema-related `current_issues` and high-level `insights` generated during their instance-level reflections.
- **Synthesize and Analyze**: This mass of raw, bottom-up feedback is then synthesized. The agent summarizes and de-duplicates the suggestions to identify systemic problems with the current schema (e.g., recurring coverage gaps, consistent type confusion). This analysis is augmented by a statistical review of entity and relation frequencies from the probing run.
- **Reflect and Evolve**: Finally, the `GraphProbingAgent` reflects on this synthesized evidence to make strategic decisions. It prunes, merges, or refines elements of the schema. This updated schema is then propagated back down to all extraction agents for the next iteration, thus closing the loop with a new top-down directive.

This two-tiered reflection framework ensures that agent interactions are transparent and purposeful. Learning is systematically shared and escalated, allowing the framework to refine both its specific outputs and its global strategy in a data-driven, bi-directional manner.

### D.3 GRAPH PROBING AGENT

This subsection details the workflow of the `GraphProbingAgent`, which induces a narrative-specific schema before large-scale extraction.

**Inputs.** The agent initializes with two complementary sources of information:

- **Scene/Chapter Summaries**: fixed global scaffolding that sketches the main storyline, major factions, and high-level stakes.
- **Reranked Insights**: narrative insights sampled from roughly 35% of chunks per iteration and reranked for relevance, ensuring that schema induction remains grounded in the actual corpus.

These inputs are packaged into a *background bundle* that combines a short narrative synopsis with an abbreviation list derived from the source material.

**Example: Background bundle for schema induction.** Below is a concise example of the JSON structure passed to the probing agent (content adapted from a science-fiction disaster narrative):

```
Example background package for schema induction (truncated)

{
  "background": "The story describes a near-future crisis where the Sun is
  rapidly expanding. A unified Earth government launches a 'wandering planet'
```

```
     project: building thousands of planetary engines to push Earth away from
     the solar system while constructing underground cities for long-term
     survival. A space-based quantum AI system orchestrates backup plans and
     emergency separation protocols.",
   "abbreviations": [
     {
       "abbr": "UEG",
       "full": "United Earth Government",
       "description": "Global authority coordinating the planetary escape project."
     },
     {
       "abbr": "MOSS",
       "full": "Quantum AI System",
       "description": "Station-based AI controller that executes backup plans."
     },
     {
       "abbr": "Ark",
       "full": "Ark Project",
       "description": "Civilization backup initiative for extreme failure cases."
     }
   ]
 }
```

The agent uses such packages to align domain-specific terminology and to construct a schema that is sensitive to narrative conventions (e.g., factions, AI systems, planetary-scale devices).

**Iterative probe–refine cycle.**   Each iteration consists of six steps:

1. **Insight search**: retrieve and rerank narrative insights to surface corpus-specific world knowledge and stylistic cues.
2. **Background update**: expand glossaries, align abbreviations (e.g., UEG, MOSS, Ark), and reconstruct the system prompt with enriched context.
3. **Schema update**: propose or adjust candidate entity and relation types informed by the updated background and insights.
4. **Trial extraction**: run probe extractions using the reflective framework, collecting schema-related feedback without retries.
5. **Feedback aggregation**: collect reflection scores, synthesize critic comments, and prune low-frequency types (e.g., occurring in $< 5\%$ of trials).
6. **Reflection**: if the schema is not stable, return to Step 1 with updated feedback; otherwise finalize and export the schema.

**Outputs.**   At convergence, the probing loop emits a schema package containing:

• refined entity and relation schemas tailored to the narrative domain;
• updated glossaries and abbreviation lists shared across agents;
• summarized reflective notes that document design choices and known limitations.

**Example: Narrative schema induced by the probing agent.**   The following example illustrates the style of entity and relation schemas produced by the GraphProbingAgent for narrative-heavy domains. In practice, the exact types and properties vary by corpus and may differ from this template.

Example: Entity type schema (truncated)

```
[
  {
    "type": "Character",
    "description": "A concrete narrative character (human or anthropomorphized).",
    "properties": {
      "name": "Name or designation",
      "role": "Narrative role or profession (optional)",
      "affiliation": "Organization/faction (optional)"
    }
  },
  {
```

```
      "type": "Event",
      "description": "A narrative event involving actions or state changes.",
      "properties": {
        "name": "Event name",
        "description": "One-sentence summary of what happens",
        "cause": "Trigger or cause (optional)",
        "result": "Outcome or impact (optional)"
      }
    },
    {
      "type": "Location",
      "description": "A salient physical place or setting.",
      "properties": {
        "name": "Place name",
        "loc_type": "Type (city/station/underground base, optional)"
      }
    },
    {
      "type": "Object",
      "description": "A narrative-relevant item, device, or artifact.",
      "properties": {
        "name": "Object name",
        "obj_type": "Category (weapon/engine/document, optional)"
      }
    },
    {
      "type": "Concept",
      "description": "An abstract entity such as an organization, plan, or ideology.",
      "properties": {
        "name": "Concept or organization name",
        "category": "Domain (political/technological/cultural, optional)"
      }
    }
  ]
```

Example: Relation type schema (truncated)

```
[
    {
      "type": "kinship_with",
      "description": "Kinship or marital relation (Character <-> Character)"
    },
    {
      "type": "social_with",
      "description": "Friendship or collegial relation (Character <-> Character)"
    },
    {
      "type": "participates_in",
      "description": "Participation in an event (Character/Object -> Event)"
    },
    {
      "type": "causes",
      "description": "Direct causal trigger (Event/Action -> Event/Action)"
    },
    {
      "type": "possesses",
      "description": "Possession of an object (Character/Concept -> Object)"
    },
    {
      "type": "located_in",
      "description": "Spatial containment or location (Character/Object/Event -> Location)"
    }
  ]
```

## D.4 KNOWLEDGE GRAPH EXTRACTION AGENT

The Knowledge Graph Extraction Agent is responsible for instantiating entities and relations under the narrative-specific schema induced by the `GraphProbingAgent` (Appendix D.3). Its workflow (Figure 9) integrates iterative reflection: each extraction pass is followed by a critic stage that evaluates schema compliance, accuracy, and coverage. If the reflection score falls below an acceptance threshold, the agent revises its output by conditioning on feedback and prior exemplars. This loop yields schema-aligned, high-quality triples rather than error-prone one-shot predictions.

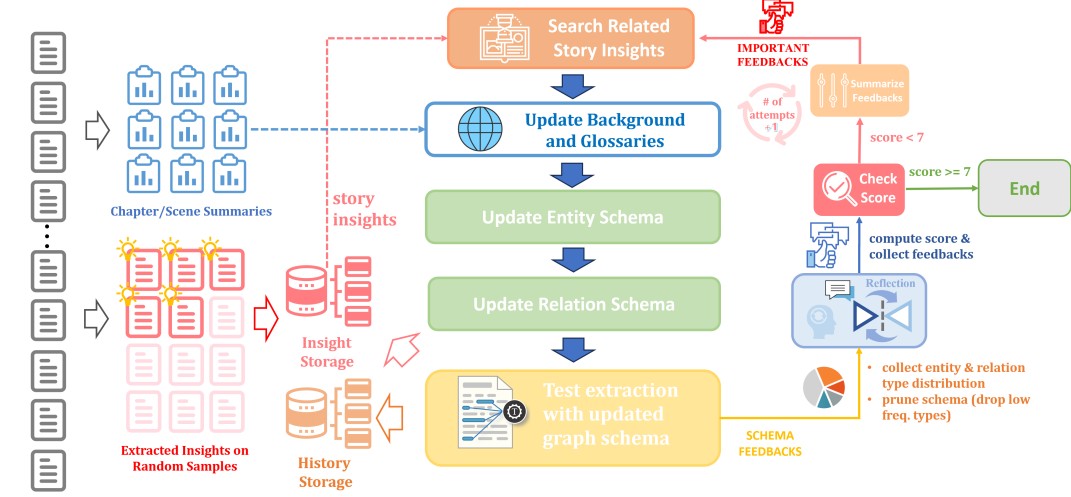

Figure 8: **Workflow of the Graph Probing Agent.** The agent iteratively incorporates narrative insights, runs probe extractions under a provisional schema, aggregates reflection feedback, and refines the schema until convergence.

**Use of the induced schema.** At run time, the extractor receives:

- the current *entity* and *relation* type definitions (cf. Example D.3);
- a small glossary and abbreviation list exported by the probing agent;
- chunk-level text and metadata (e.g., scene index, speaker tags).

These components are compiled into prompts that enumerate admissible types and specify task constraints.

**Entity extraction prompt.** The entity extraction step is parameterized by a prompt that enumerates allowed types and enforces strict output structure:

---

**Entity Extraction Prompt Template**

You are tasked with identifying and extracting entities from the given text.
**Type enumeration (field `type`):** choose strictly from the following English, case-sensitive list (no custom values or translations):
{ENTITY_TYPE_DESCRIPTION_TEXT}

**Rules**

- The field `type` must exactly match one of the enumerated values above (case-sensitive).
- If uncertain, default to `Concept`.
- Do *not* invent new values, use non-English names, or lowercase/camelCase variants.

**Scope field**

- The field `scope` must be either `"global"` or `"local"`.
- `"global"`: named entities or concepts with stable identity and potential recurrence.
- `"local"`: one-off or generic references lacking a stable identity.

**Additional constraints**

- If no aliases, return an empty array `aliases:[]`.
- Focus on entities central to events: characters, key objects, organizations, locations.
- Ignore irrelevant objects, background scenery, and natural phenomena.
- Do not add explanations or extra text beyond JSON.

**Text**

---

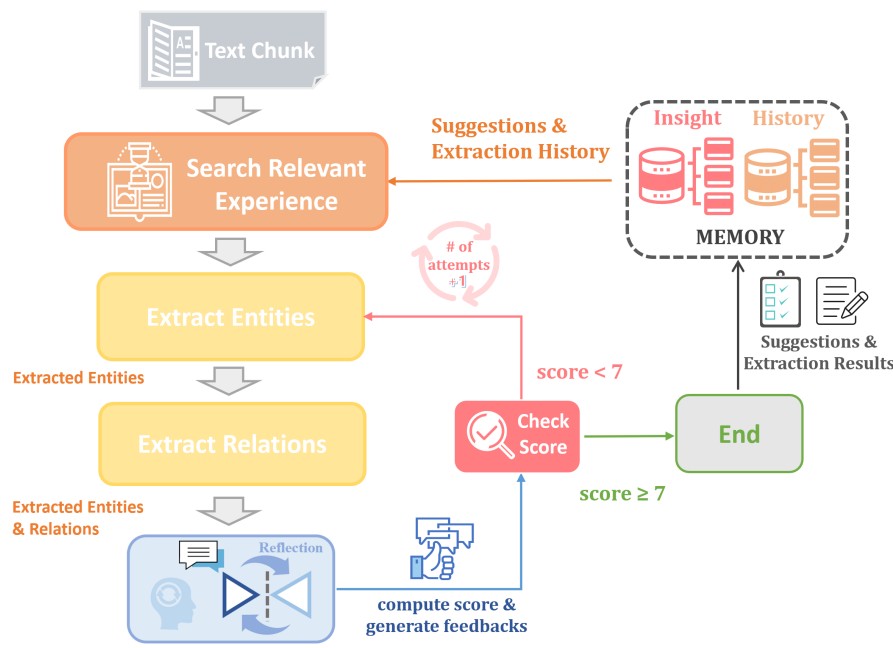

Figure 9: **Knowledge Graph Extraction Agent.** Guided by an induced schema, the agent performs document-level entity and relation extraction under an *extract → reflect → revise* loop. A critic scores accuracy, consistency, and redundancy; if the score is below threshold, targeted revisions are applied using feedback and reflection memory exemplars until convergence or budget exhaustion.

```
{TEXT}
```
**Return strictly the following JSON**
```
{
  "entities": [
    {
      "name": "Entity name",
      "type": "Entity type (must match enum)",
      "scope": "global or local",
      "description": "Concise description and evidence",
      "aliases": ["Alias1", "Alias2"]
    }
  ]
}
```

**Relation extraction prompt.** Relation extraction is constrained to the entities already detected in the same chunk, using the relation schema exported by the probing agent:

---

**Relation Extraction Prompt Template**

Identify relations *only among the listed entities (with types)* from the given text.
**Known entities (with types)**
{ENTITY_LIST}

**Allowed relation types (enumeration)**
{RELATION_TYPE_DESCRIPTION_TEXT}

**Rules**

- Use only the enumerated English relation types above (case-sensitive).
- Consider relations only among the listed entities. If none are listed, return empty `relations`.

---

- If a relation cannot be clearly inferred, prefer not to extract it.
- Ignore low-value, ambiguous, or narrator/meta elements.
- Output JSON only (no additional text).

**Input Text**
`{TEXT}`

**Return strictly the following JSON**

```
{
  "relations": [
    {
      "subject": "Subject entity",
      "relation_type": "One of the allowed types",
      "relation_name": "Concrete relation name",
      "object": "Object entity",
      "description": "Optional rationale/evidence"
    }
  ]
}
```

**KG extraction reflection prompt.** To close the extract–reflect–revise loop, the extractor is paired with a reflection prompt that evaluates and critiques the current output:

**KG Extraction Reflection Prompt Template**

You are a senior KG engineer and story analyst. Reflect on the quality of entity/relation extraction across three axes: *accuracy*, *consistency*, and *redundancy*.
**Type & relation enumerations (for reference only)**

- Entity types: `{ENTITY_TYPE_DESCRIPTION_TEXT}`
- Relation types: `{RELATION_TYPE_DESCRIPTION_TEXT}`

**Important**

- You *may* flag the use of undefined or incorrect relation types.
- You *must not* propose schema modifications.

**Evaluation axes**

- **Accuracy**: logical validity; adherence to the schema; use of only allowed relation types.
- **Consistency**: unified entity naming; no coreference confusion or near-duplicates.
- **Redundancy**: avoid low-value or repetitive entities/relations.

**Scoring guideline (0–10)**

- 10: excellent; no changes needed.
- 7–9: good; minor possible improvements.
- 3–6: usable but with notable issues.
- 0–2: largely wrong or missing.

If the extraction `logs` are missing or empty, return `score = 0`, set `current_issues = ["Missing extraction logs"]`, and leave `insights` empty.

**Return strictly the following JSON**

```
{
  "current_issues": ["List concrete issues to fix"],
  "insights": ["Optional story-level insights"],
  "score": 0-10
}
```

**Extraction logs to evaluate**
`{LOGS}`

**Reflection implementation.** The reflection prompt is implemented by a lightweight controller (`DynamicReflector`) that manages memory, logs, and retrieval. It combines prompt-driven critique with structured evidence from prior extractions:

- **Log synthesis.** After each extraction pass, entities and relations are converted into natural-language logs. Entities include name, type, scope, and description; relations include subject, relation name/type, object, and optional evidence. Prior mentions are surfaced from an `entity_extraction_memory` map.

- **Memory write.** Two vector memories are maintained: a `history_memory` storing sentence-level evidence with notes and current scores, and an `insight_memory` storing critic-produced insights. New sentences that mention extracted entities or subject–object pairs are indexed into `history_memory`; insights are indexed into `insight_memory`.

- **Targeted retrieval.** For a new context, each sentence queries both memories. Retrieved items are re-ranked by an LLM reranker; only items with `relevance_score` $\geq 0.5$ are kept to condition the next revision step.

- **Revision.** The agent revises its extraction conditioned on retrieved history and insights, continuing until convergence or retry budget exhaustion.

This design grounds the reflection loop in accumulated evidence rather than treating each round as isolated, resulting in more consistent and schema-compliant extractions.

## D.5 CMP EXTRACTION AGENT

For screenplay-specific details such as wardrobe, styling, and props, we implement the CMP Extraction Agent. Unlike the Knowledge Graph or Attribute agents, this component outputs flat records that are written into a relational SQL database for downstream production analysis. Its workflow (Figure 10) targets highly granular extraction: multiple candidates are proposed in parallel, overlapping results are merged, and a lightweight reflection step filters spurious mentions and enforces consistency.

**Unified CMP Extraction Prompt (Wardrobe / Styling / Prop).**

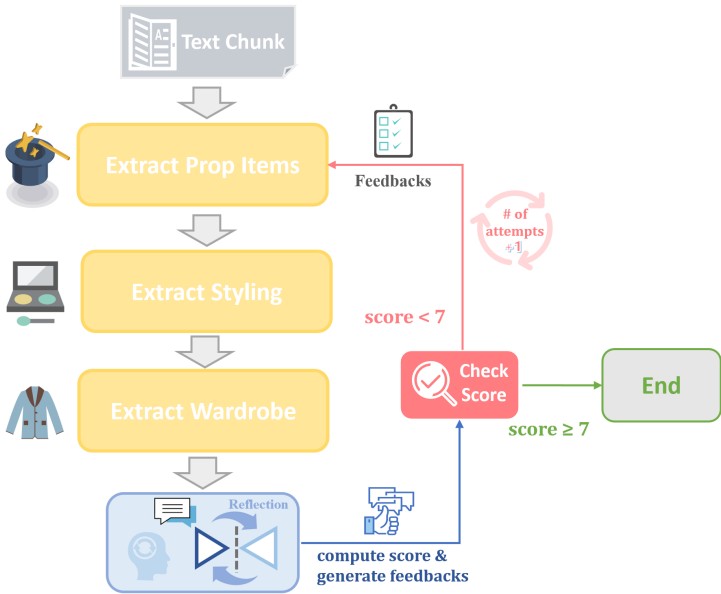

Figure 10: **CMP Extraction Agent (costume, makeup, props).** The agent extracts wardrobe, styling, and prop records in parallel, merges overlapping candidates, applies lightweight reflection, and writes standardized results to a SQL database.

**Unified CMP Extraction (Chain-of-Thought)**

Think internally following these steps, but output *only* JSON.
1) Identify characters, scene context, and key actions from the text.
2) Determine the item category:

- **wardrobe** (worn items): clothing nouns linked to a person; clothing verbs (*wear, put on, zip up, change into*); explicit scene + clothing (*change into uniform*); strong scene-specific uniforms (e.g., spacewalk → spacesuit). Avoid weak guesses (e.g., generic weather).

- **styling** (makeup/hair/SFX makeup): infer concise styling traits relative to state or scene (e.g., light makeup, messy hair, blood SFX makeup, old-age makeup).

- **prop** (held/operated/scene-operated or necessarily implied objects): character-props (e.g., umbrella, phone, glass), scene-props (door, curtain, lamp, console), or verb-implied objects (*unlock* → key/keypad).

3) Filtering rules: do not extract locations/organizations/abstract concepts; do not keep background scenery unless operated on or undergoing change; large vehicles/facilities only if directly controlled or key to the action.
4) Standardize names/subcategories; fill appearance/status; add notes for model/codes, etc.
5) One record corresponds to one *character–item* pair.
**Output JSON**

```
{
  "results": [
    {
      "name": "",
      "category": "wardrobe | styling | prop",
      "subcategory": "",
      "appearance": "",
      "status": "",
      "character": "",
      "evidence": "",
      "notes": ""
    }
  ]
}
```

*Note.* The previously separated wardrobe/styling/prop prompts are subsumed by this unified template; they follow the same schema and decision rules with only `category` and `subcategory` differing.

**CMP Reflection Prompt.**

**Unified CMP Extraction (Chain-of-Thought)**

Think internally following these steps, but output *only* JSON.
1) Identify characters, scene context, and key actions from the text.

2) Determine the item category:

- **wardrobe** (worn items): clothing nouns linked to a person; clothing verbs (*wear, put on, zip up, change into*); explicit scene + clothing (*"change into uniform"*); strong scene-specific uniforms (e.g., spacewalk → spacesuit). Avoid weak guesses (e.g., generic weather).

- **styling** (makeup/hair/SFX makeup): infer concise styling traits relative to state or scene (e.g., light makeup, messy hair, blood SFX makeup).

- **prop** (held/operated/scene-operated or implied objects): character-props (umbrella, phone, glass), scene-props (door, curtain, console), verb-implied objects (*unlock* → *key*).

3) Filtering rules: do not extract locations/organizations/abstract concepts; do not keep background scenery unless operated on or undergoing change; large vehicles/facilities only if directly controlled or key to the action.

4) Standardize names/subcategories; fill appearance/status; add notes for model/codes, etc.

5) One record corresponds to one *character–item* pair.

**Output JSON**

```
{
  "results": [
    {
      "name": "",
      "category": "wardrobe | styling | prop",
      "subcategory": "",
      "appearance": "",
      "status": "",
      "character": "",
      "evidence": "",
      "notes": ""
    }
  ]
}
```

*Note.* The previously separated wardrobe/styling/prop prompts are subsumed by this unified template; they follow the same schema with only `category` and `subcategory` differing.

**Implementation notes (SQL writing).**   Each CMP record is a flat row with fixed fields (`name`, `category`, `subcategory`, `appearance`, `status`, `character`, `evidence`, `notes`). After reflection, validated rows are inserted into the SQL schema:

- Tables: `wardrobe`, `styling`, `prop` (same columns as above), plus audit columns (`scene_id`, `source_doc`, `timestamp`).

- Normalization: standardized `name`/`subcategory` and consistent casing; `character` links to the character dictionary (foreign key or soft link).

- De-duplication:   overlap-aware merging by (`scene_id`, `character`, `name`, `subcategory`) with conflict resolution preferring entries with stronger evidence or higher reflection scores.

This integration enables production-oriented queries (e.g., costume continuity, per-character prop inventories) and supports downstream analytics.

## D.6   ATTRIBUTE EXTRACTION AGENT

Following entity normalization and disambiguation, the **Attribute Extraction Agent** enriches each canonical entity with schema-defined properties (e.g., identity, affiliation, physical characteristics, organizational role). Whereas the Knowledge Graph Extraction Agent supplies structural relations, the Attribute Extraction Agent provides the fine-grained semantic content needed for temporal reasoning, causal modeling, downstream QA, and cross-entity consistency checks.

**Degree-aware processing.**   As shown by the degree statistics in Table 5, nearly 90% of canonical entities have total degree $\leq 2$ and serve primarily as peripheral anchors in the narrative graph. For these nodes, the extraction-time descriptions are already sufficient for grounding and retrieval, and further enrichment provides limited benefit. Consequently, only two categories of entities enter the enrichment workflow: (i) nodes with degree $> 2$, which participate in multi-scene interactions, and (ii) all Event nodes, which remain semantically central regardless of degree.

**Extraction modes.**   For entities selected for enrichment, the agent supports two complementary extraction modes:

- **Single-round (standard) extraction**: used when the evidence for an entity is localized and self-contained. The system aggregates goal-conditioned snippets into a compact summary and fills all schema-defined attributes in one pass.
- **Incremental extraction**: used for long-form narratives segmented into chunks. The agent incrementally updates the entity profile as new evidence appears, incorporating reflection feedback to refine, extend, or correct attribute values while maintaining stable definitions across iterations.

**Reflection and selective retry.**    After each extraction pass, a critic evaluates the completeness, correctness, and evidence grounding of the predicted attributes. Rather than regenerating the entire profile, the system performs *selective retries* only for fields that fail the reflection criteria, using targeted repair prompts and bounded retry budgets. This strategy minimizes unnecessary computation, preserves provenance, and ensures high-quality, semantically coherent entity representations.

DEGREE-AWARE ENTITY SELECTION FOR ATTRIBUTE ENRICHMENT

To support the degree-aware enrichment strategy described in Section 3.2.1, we analyze the structural properties of canonical entities across ten full-length NarrativeQA screenplays. Our goal is to determine which entities benefit from adaptive attribute enrichment and which can safely retain their extraction-time descriptions without loss of utility.

**Motivation.**    Attribute enrichment is most valuable for *structurally central* entities— those that participate in many relations, recur across scenes, or interact with multiple narrative threads. Conversely, peripheral nodes (low degree) tend to serve as local anchors or one-off mentions, and enriching them yields minimal downstream benefit. To justify this design, we compute total graph degree (in + out) for all canonical entities after entity normalization and relation consolidation.

**Degree statistics.**    Across the ten NarrativeQA screenplays, we observe a highly skewed distribution. Most nodes exhibit extremely low connectivity:

- **Mean degree**: 1.76
- **Median degree**: 1.0
- **Standard deviation**: 5.15

Percentile analysis further highlights the sparse structure:

- 25th percentile: 0.0
- 50th percentile: 1.0
- 75th percentile: 1.0
- 90th percentile: 3.0
- 95th percentile: 6.0
- 99th percentile: 19.0

**Distribution.**    Figure 11 visualizes the degree distribution. Nearly 80% of nodes fall into the 0–1 degree range, and more than 88% have degree at most 2. Only a long tail of nodes—with degrees above 3—represent structurally important entities with rich cross-scene interactions.

Table 5 shows the proportion of nodes at each observed degree value.

Table 5: Grouped degree distribution of canonical entities (NarrativeQA, 10 scripts).

| Degree Range | Percentage of Nodes |
|---|---|
| 0–1 (isolated or peripheral) | 78.98% |
| 2 (lightly connected) | 8.41% |
| 3–5 (moderately connected) | 7.13% |
| 6–20 (high connectivity) | 3.51% |
| >20 (long-tail hubs) | 1.97% |

**Implications for enrichment.**    Based on this distribution, we enrich only:

1. entities with **degree** $> 2$, and
2. **all Event nodes**, regardless of degree,

since events frequently participate in causal chains even when their local connectivity appears low. At this stage, however, only entity and relation mentions from the initial extraction pipeline are available; the dense event–event and event–plot linkages used in downstream causal reasoning are introduced later during the event-centric refinement phase. This degree-aware selection avoids unnecessary computation on 88% of peripheral nodes while focusing enrichment on the structurally central 12% that meaningfully benefit from adaptive refinement.

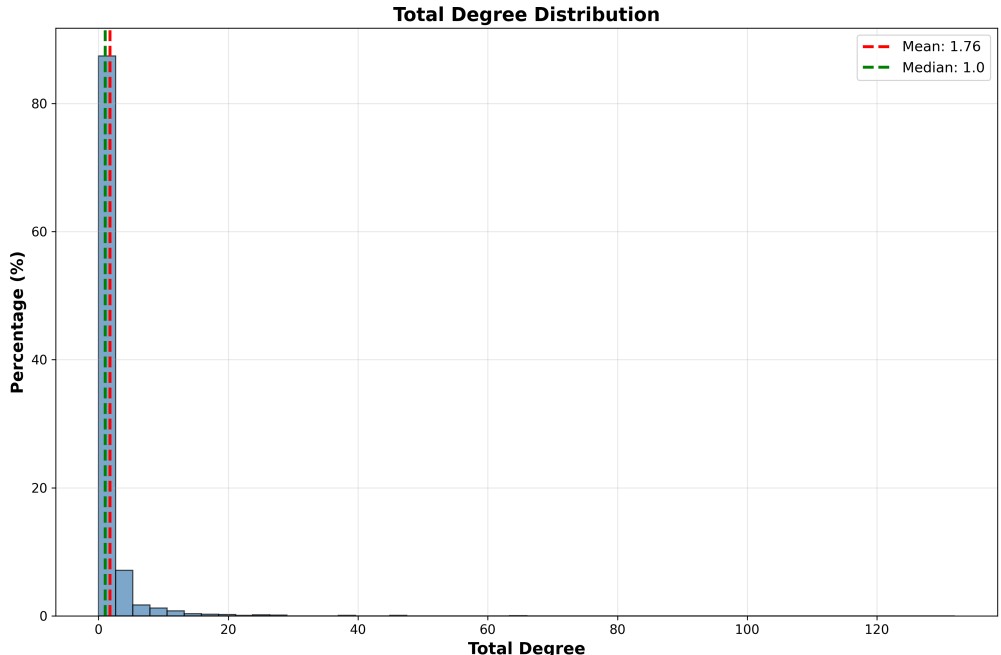

Figure 11: **Degree distribution across canonical entities** (NarrativeQA, 10 scripts). The distribution is heavily skewed: most nodes have low degree, while a small long tail of structurally central entities governs cross-scene interactions.

GOAL-CONDITIONED CONTEXT SELECTION

Before attribute enrichment begins, the system applies a **goal-conditioned content selector** that filters long narrative text into short, attribute-relevant evidence snippets. Here, the *goal* defines what information is currently needed for the entity being enriched. At the first pass, the goal consists of the target entity and the specific attribute to be extracted (e.g., "Person.Name", "Event.Trigger"). During later iterations, the goal is expanded with unresolved issues produced by the reflection module—such as missing core participants, ambiguous timestamps, or conflicting descriptions—allowing the selector to retrieve only the passages that help resolve these points. This progressively focuses the evidence and reduces noise in downstream reflection and correction stages, improving both stability and efficiency.

**Goal-Conditioned Content Extraction Prompt.**

> ### Goal-Conditioned Content Extraction Template
>
> You are an expert in Chinese reading comprehension and key information extraction. Given a **goal** (the target attribute and any open issues from previous reflection) and a piece of source **text**, select and succinctly summarize only the content that directly supports this goal, within {max_length} characters.
>
> **Requirements**
>
> - Retain only information clearly relevant to the goal; ignore all other details.
> - Do not hallucinate or infer facts not grounded in the text.

- Summaries may paraphrase but must preserve the original meaning.
- If no relevant evidence exists, return an empty string.

**Output (JSON only)**

```
{
   "related_content": "<content relevant to the goal>"
}
```

## SINGLE-ROUND ATTRIBUTE EXTRACTION

Before any iterative enrichment takes place, the system performs an initial *single-round attribute extraction* stage. This step is applied to entities whose evidence is largely self-contained, meaning that most of their attribute-relevant information can be recovered from a compact summary rather than requiring multi-pass refinement. The input to this stage is the goal-conditioned snippet assembled in the previous step, where long narrative segments have already been filtered and condensed into a short piece of text focused on the target entity.

In this setting, the extractor attempts to fill all schema-defined attributes in one pass. Unlike the incremental enrichment phase that follows, this stage does not revise attributes based on reflection feedback, introduce new fields, or merge information across multiple occurrences. Its purpose is to produce a clean, schema-aligned initial profile that can be evaluated—and, if necessary, refined—by later stages.

**Standard Attribute Extraction Prompt.**

> **Standard Attribute Extraction Prompt Template**
>
> You are a senior knowledge graph engineer. Your task is to extract structured attributes for a target entity using the provided context and the predefined attribute schema.
> Entity information:
>
> ```
> Name: {ENTITY_NAME}
> Type: {ENTITY_TYPE}
> Type description: {DESCRIPTION}
> ```
>
> Context:
>
> ```
> {TEXT}
> ```
>
> Attribute schema:
>
> ```
> {ATTRIBUTE_DEFINITION}
> ```
>
> **Rules**
> - Fill each attribute with an empty string when evidence is missing.
> - Use attribute names exactly as defined; do not add or remove fields.
> - The field `new_description` must be a concise, non-empty summary.
>
> **Output JSON**
>
> ```
> {
>    "new_description": "...",
>    "attributes": {
>       "Attribute1": "...",
>       "Attribute2": "..."
>    }
> }
> ```

## INCREMENTAL ATTRIBUTE EXTRACTION

For long narratives segmented into multiple chunks, attribute extraction must be adaptive and iterative. The incremental extractor integrates new evidence, inherits stable values, and extends the attribute schema when a chunk introduces genuinely new semantics.

**Incremental Attribute Extraction Prompt.**

---

**Incremental Attribute Extraction Prompt Template**

You are a senior knowledge graph engineer following the **CMP (Common-sense & Manuscript Prior)** principle.

This is an **incremental** extraction task: You receive one text chunk at a time, along with the entity's previously extracted attributes and definitions. Your task is to refine, expand, and reconcile attributes using the new evidence.

**Inputs**

```
entity_name: {entity_name}
entity_type: {entity_type}
description: {description}

text: {text}

attribute_definitions: {attribute_definitions}
prev_attributes: {prev_attributes}
prev_description: {prev_description}
```

**Goals**

- Inherit correct past attributes.
- Complete missing fields using the new text.
- Update conflicting or outdated values strictly based on evidence.
- Introduce new attributes only when necessary and semantically stable.
- Extend attribute definitions for any new fields.
- Produce a concise, updated `new_description`.

**CMP Principles**

- Evidence $\geq$ inference.
- Stable inference allowed; unstable inference becomes empty string.
- Attribute names cannot be modified once established.
- All fields must remain logically consistent.

**Output**

```
{
  "new_description": "...",
  "attributes": { ... },
  "attribute_definitions": { ... }
}
```

---

ATTRIBUTE REFLECTION (INSTANCE-LEVEL)

Unlike schema-level reflection performed by the Graph Probing Agent, the attribute reflection module focuses solely on *instance-level* quality: whether each field is complete, correctly grounded in evidence, and internally consistent. It does *not* propose schema edits; instead, it highlights which attributes require retry and why, producing a numerically calibrated score to guide subsequent refinement.

**Attribute Reflection Prompt.**

---

**Attribute Reflection Prompt Template**

You are a knowledge graph quality auditor. Your task is to evaluate the *completeness*, *correctness*, and *evidence grounding* of extracted attributes for a narrative entity.

**Entity**

```
Type: {ENTITY_TYPE}
Description: {DESCRIPTION}
```

---

**Attribute schema**

```
{ATTRIBUTE_DEFINITIONS}
```

**Extracted attributes**

```
{ATTRIBUTES}
```

**Scoring Rubric (0–10)**

- **10** — All fields complete; values fully consistent and strongly grounded in the context.
- **7** — Minor omissions or weak grounding; most fields reliable.
- **5** — Several incomplete or weakly supported fields; usable but needs partial revision.
- **3** — Many empty or unsupported fields; major revisions required.
- **0** — Unusable extraction (missing fields, contradictions, or no valid grounding).

**Evaluation Criteria**

- **Completeness**: Are required fields filled? Which remain empty?
- **Correctness**: Are the values justified by the available textual evidence?
- **Grounding**: Are any fields speculative or unsupported?
- **Retry Guidance**: Identify fields needing revision and explain each issue ($\geq 30$ words).
- **Important**: Do not suggest schema changes; evaluate only the instance.

**Output JSON**

```
{
  "score": 0-10,
  "feedbacks": ["Brief evaluator comments..."],
  "attributes_to_retry": ["Field1", "Field2"]
}
```

## D.7 ENTITY NORMALIZATION AND DISAMBIGUATION

**Step 1: Dual embeddings and similarity.** For each candidate name $i$, compute a surface–name embedding $e_i^{(n)}$ and a description (or summary) embedding $e_i^{(d)}$. Cosine similarities yield two matrices $S_{ij}^{(n)} = \cos(e_i^{(n)}, e_j^{(n)})$ and $S_{ij}^{(d)} = \cos(e_i^{(d)}, e_j^{(d)})$. Combine them as

$$S = \alpha S^{(n)} + (1-\alpha) S^{(d)}, \qquad \alpha \in (0,1) \ \text{(default} \approx 0.8).$$

**Step 2: $k$-NN graph and Laplacian.** Form a $k$-NN adjacency by connecting each node $i$ to its top-$k$ neighbors under $S$ (excluding $i$), and symmetrize: $A \leftarrow \max(A, A^\top)$. Let $\mathbf{1}$ be the all-ones column vector; the degree matrix is

$$D = \mathrm{diag}(A\mathbf{1}),$$

and the (unnormalized) Laplacian is $L = D - A$.

**Step 3: Estimating the number of clusters.** Let $\lambda_1 \leq \lambda_2 \leq \cdots \leq \lambda_n$ be the eigenvalues of $L$. Compute gaps $g_r = \lambda_{r+1} - \lambda_r$ and select

$$\hat{k} = \arg\max_{r \geq 2} g_r,$$

i.e., an eigengap heuristic that ignores the first gap to reduce sensitivity to trivial modes. To stabilize small-$n$ regimes, shrink $\hat{k}$ toward $n/2$ via a convex combination and clamp to be at least 2:

$$k_{\text{final}} = \max\left(2, \left\lfloor \tfrac{1}{2}\hat{k} + \tfrac{1}{4}n \right\rfloor\right).$$

(Default weights are conservative to avoid under-segmentation.)

**Step 4: $k$-means in a joint embedding space.** Embed each name by concatenating modalities with a balancing weight:

$$\tilde{e}_i = \left[\beta e_i^{(n)} ; (1-\beta) e_i^{(d)}\right], \qquad \beta \in (0,1) \ \text{(default} \approx 0.5).$$

Run $k$-means with $k = k_{\text{final}}$ on $\{\tilde{e}_i\}$ to obtain candidate clusters; discard singletons (size threshold $\geq 2$) before adjudication.

**Step 5: LLM adjudication and application.** Serialize each remaining cluster into a compact, narrative-aware context and evaluate with an LLM enforcing: (i) instance-over-category, (ii) version/model separation, and (iii) naming priority (canonical proper names preferred). Accepted aliases yield a rename map applied consistently to entity nodes and relation endpoints.

---

**Entity Disambiguation Prompt Template**

You are an expert in knowledge-graph construction. Your task is to decide which of the following names refer to the same entity and which should remain distinct.

Rules (must follow):

- Similar spelling is supportive but not sufficient.
- Merge only if identity, role, and narrative function are consistent.
- Do not merge disguises, substitutes, parallel versions, or different life stages.
- Names with model/version identifiers denote distinct entities.
- Do not merge an individual (proper name, nickname, unique ID) into a generic class. If both occur, prefer the individual as canonical.
- Apply naming priority when merging: Proper name > nickname > code name > role/office > class/species.

Input Entity Information:

```
{ENTITY_DESCRIPTIONS}
```

Return the result strictly in the following JSON format:

```
{
  "merges": [
    {
      "canonical_name": "Main name",
      "aliases": ["Alias1", "Alias2"],
      "reason": "Short explanation citing rules"
    }
  ],
  "unmerged": [
    {
      "name": "Unmerged name",
      "reason": "Short explanation citing rules"
    }
  ]
}
```

---

# E  IMPLEMENTATION DETAILS OF EVENT-CENTRIC GRAPH REFINEMENT

## E.1  EVENT CAUSALITY ADJUDICATION

Each surviving pair is represented by their event cards and enriched with common-neighbor context from the KG. These are then passed to an LLM to classify the relation into one of four types: `CAUSES`, `INDIRECT_CAUSES`, `PART_OF`, or `None`. The model additionally outputs temporal ordering, a short rationale, and a confidence score.

---

**Event Relation Classification Prompt Template**

You are an expert in event graph modeling. Perform rigorous step-by-step reasoning internally, but output JSON only (never reveal the reasoning).

Task goals:

---

- Decide a single relation from `CAUSES`, `INDIRECT_CAUSES`, `PART_OF`, or `NONE`.
- Decide temporal order: `E1_before_E2`, `E2_before_E1`, `Overlap`, or `Unknown`.

Event 1 Information:
`{EVENT_1_INFO}`

Event 2 Information:
`{EVENT_2_INFO}`

Internal chain-of-thought steps (DO NOT output):

1. Align key information: names, time cues, locations, participants, and evidence; mark unknown if missing.

2. Temporal order: decide E1_before_E2, E2_before_E1, Overlap, or Unknown.

3. Relation type:
   - If E1 is a substage of E2, choose `PART_OF`.
   - If mediated by another condition, choose `INDIRECT_CAUSES`.
   - If direct trigger words and clear sequence appear, choose `CAUSES`.
   - Otherwise, choose NONE.

4. Confidence & output: assign a score in [0,1] based on evidence strength; give a short (20–40 word) justification, then output JSON only.

Return the result strictly in the following JSON format:

```
{
  "relation": "CAUSES",
  "temporal_order": "E1_before_E2",
  "reason": "justification without exposing hidden reasoning",
  "confidence": 0.0
}
```

Only output the JSON above. No extra text.

## E.2 CAUSAL GRAPH PRUNING WITH SABER

The adjudication stage typically yields a dense causality graph that may contain cycles and redundant edges. To refine this structure, we propose **SABER (Semantic-Aware Breaker of Event Relation)**, which systematically prunes edges while preserving valid causal chains. SABER enhances both the interpretability of the event graph and the stability of downstream analysis by combining structural heuristics with semantic validation. The detailed procedure is summarized in Algorithm 2.

**Stage I: Handling Strongly Connected Components.** Cycles indicate contradictions with narrative logic. For each strongly connected component (SCC), SABER sorts edges by confidence $c(e)$ and removes the lowest-confidence edges until the cycle is resolved. If multiple candidate edges remain, the decision is delegated to the LLM for semantic adjudication.

**Stage II: Refining Weakly Connected Structures.** Large weakly connected components often exhibit *flattened causal patterns*, where a source $A$ connects directly to $C$ as well as indirectly via an intermediate $B$. In such cases, SABER first applies a confidence-based filter, discarding edges below threshold $\tau_c$. When ambiguity remains (e.g., whether $A \rightarrow C$ should be replaced by $A \rightarrow B \rightarrow C$), an LLM compares semantic plausibility and selects the more coherent path.

**Iteration and Convergence.** SABER runs iteratively, updating the graph structure after each pruning round. The process terminates once no cycles or redundancies remain, or after reaching a maximum number of iterations. To ensure auditability, each deletion is logged with its edge identifier, confidence score, and adjudication rationale.

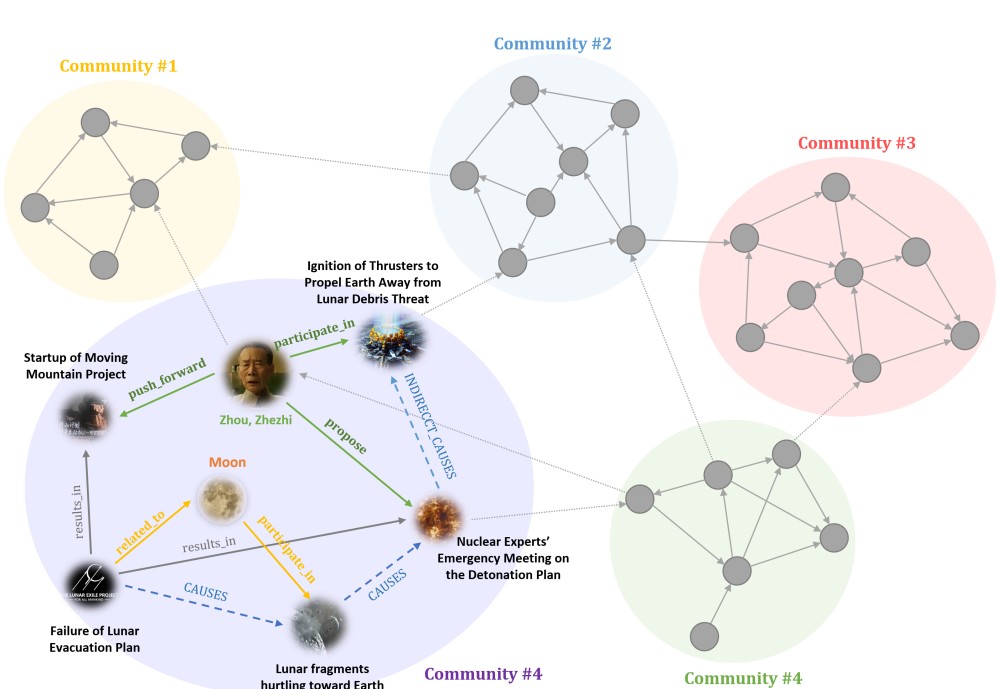

Figure 12: **Candidate event-pair selection on the KG.** Events are proposed as causal candidates when they co-occur within communities and satisfy structural or semantic proximity (e.g., common neighbors and bounded hops). If the extraction stage has already produced an explicit `results_in` edge between two events, it is directly promoted to CAUSES. Otherwise, an LLM adjudicator assigns CAUSES, INDIRECT_CAUSES, PART_OF, or NONE with calibrated confidence.

---

**Algorithm 2** SABER: Semantic-Aware Breaker of Event Relation

---

**Require:** Event causality graph $G = (V, E)$ with confidence scores $c(e)$
**Ensure:** Pruned DAG $G'$
1: $G' \leftarrow G$
2: **repeat**
3:     Identify SCCs in $G'$
4:     **for** each SCC **do**
5:         Remove lowest-confidence edges; resolve ties via LLM
6:     **end for**
7:     Detect flattened causal patterns $(A, B, C)$
8:     **for** each $(A, B, C)$ **do**
9:         **if** $c(A \rightarrow C) < \tau_c$ **then**
10:        Remove $A \rightarrow C$
11:      **else**
12:         Compare $A \rightarrow C$ vs. $A \rightarrow B \rightarrow C$ using LLM
13:      **end if**
14:     **end for**
15: **until** no cycles or redundancies OR max iterations reached
16: **return** $G'$

---

### E.3 From Event Chains to Plots

As illustrated in Figure 13, the annotated causal chains undergo three stages of refinement—confidence-based preprocessing, trunk–branch segmentation, and redundancy consolidation—to produce concise, structurally coherent subchains. These refined subchains then serve as inputs for plot adjudication and for inferring inter-plot relations, both of which are depicted in the figure.

**Step 1: Confidence-based Chain Preprocessing.** Given a candidate chain $c = \langle e_1, \ldots, e_L \rangle$, each edge $(e_i, e_{i+1})$ is annotated with type $t$ and confidence $\gamma$. We apply a weighted thresholding rule:

$$\gamma_{\text{eff}}(e_i, e_{i+1}) = w_t \cdot \gamma,$$

where $w_t$ is a type-specific weight (e.g., $w_{\text{CAUSES}} = 1.0$, $w_{\text{INDIRECT}} = 0.6$, $w_{\text{PART\_OF}} = 0.0$). Chains are split at edges with $\gamma_{\text{eff}} < \theta$. Segments shorter than two events are discarded, yielding subchains that only retain strong causal continuity.

**Step 2: Trunk–Branch Segmentation.** We insert all confidence-filtered chains into a path trie and emit a segment whenever reaching a branch point (a node with multiple children) or a chain termination. Shared prefixes act as trunks; divergent continuations form branches. To preserve continuity around splits and avoid dropping short leaf branches (e.g., $B \rightarrow G$), we adopt two practical choices: (i) *split-inclusive starts* — new segments may start at the split node, so a split at $B$ can yield segments starting with $B$ (e.g., $[B, \ldots]$); (ii) *terminal backfill* — when a leaf would otherwise yield a length-1 segment, we materialize the final edge as a minimal two-event segment (e.g., $[B, G]$). Optionally, contained segments (exact contiguous substrings of longer ones) are removed for conciseness before redundancy consolidation (Step 3).

**Step 3: Redundancy Consolidation.** We remove redundancy in two passes: (i) strict-subset elimination on event *sets*, and (ii) near-duplicate pruning by set overlap.

**Notation.** For a sequence $x = \langle x_1, \ldots, x_\ell \rangle$, let $\text{set}(x)$ be the set of distinct items in $x$. We use the **Szymkiewicz–Simpson coefficient** (overlap coefficient) between sets $A$ and $B$:

$$s_{\text{SS}}(A, B) = \begin{cases} 1, & A = \emptyset \ \wedge \ B = \emptyset, \\ 0, & \min(|A|, |B|) = 0 \text{ and } A \cup B \neq \emptyset, \\ \dfrac{|A \cap B|}{\min(|A|, |B|)}, & \text{otherwise.} \end{cases} \tag{1}$$

---

**Algorithm 3** Maximal-Set Filtering via Inverted Index (Strict-Subset Removal)

---

1: **Input:** sequences $\mathcal{S}$
2: **Output:** indices whose event sets are *maximal*
3: Convert to sets: $S_i \leftarrow \text{set}(s_i)$; discard sequences with $|s_i| < k$
4: Sort indices $I$ by $-|S_i|$ (larger sets first), then by $-|s_i|$, then lexicographic $s_i$
5: Initialize inverted index Inv and kept set Keep
6: **for** $i \in I$ **do**
7:      $A \leftarrow S_i$; order $A$ by increasing $|\text{Inv}[e]|$;   $C \leftarrow \bigcap_{e \in A} \text{Inv}[e]$
8:      **if** exists $j \in C$ with $|S_j| > |A|$ **then**
9:          **continue**
10:      **end if**
11:      Keep $i$; update $\text{Inv}[e]$ for $e \in A$
12: **end for**
13: **return** Keep

---

---

**Algorithm 4** Near-Duplicate Pruning by Szymkiewicz–Simpson Overlap

---

1: **Input:** sequences $\mathcal{Q}$; threshold $\tau$; small signature size $r$
2: **Output:** pruned list $\mathcal{Q}^\star$
3: Sort $\mathcal{Q}$ by $-|q|$; compute token frequency $f(\cdot)$
4: Initialize rare-element index RInv; kept sets $\mathcal{K}$; result $\mathcal{Q}^\star$
5: **for** each $q \in \mathcal{Q}$ **do**
6:     $A \leftarrow \text{set}(q)$; choose signature $\text{sig}(q)$ as the $r$ least-frequent elements in $A$
7:     Candidates $C \leftarrow \bigcup_{e \in \text{sig}(q)} \text{RInv}[e]$
8:     **if** exists $j \in C$ with $s_{\text{SS}}(A, \mathcal{K}[j]) \geq \tau$ **then**
9:         **continue**
10:    **end if**
11:    Append $q$ to $\mathcal{Q}^\star$; append $A$ to $\mathcal{K}$; update $\text{RInv}[e]$
12: **end for**
13: **return** $\mathcal{Q}^\star$

---

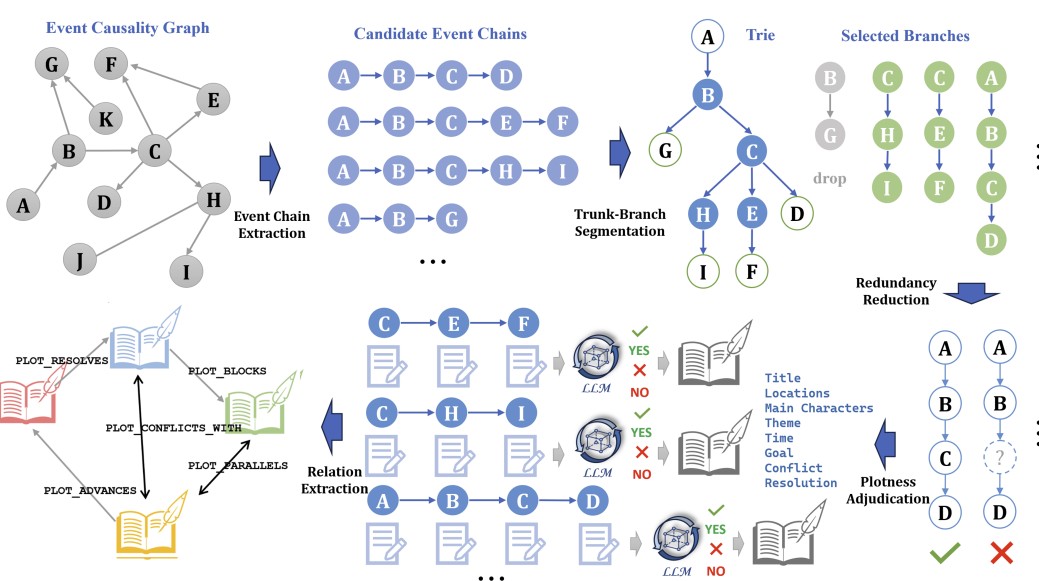

Figure 13: **Promoting causal event chains to plots.** Confidence-filtered event chains are segmented into trunk–branch structures, consolidated to remove redundancies, adjudicated as coherent plot units, and linked with inter-plot relations.

**Prompt Templates for Plot Generation.** We provide the prompt templates used for event chain adjudication and plot relation classification.

---

**Event Chain to Plot Adjudication Prompt Template**

You are a narrative structure analyst. Your task is to decide whether a given event chain constitutes a coherent *plot unit*, and, if so, to generate structured plot information.
A chain is considered a valid plot only if it meets all of the following conditions:

- **Goal consistency**: events revolve around a shared goal, task, or conflict.
- **Order interpretability**: events follow an interpretable sequence; at most one "explanatory segment" is allowed, which must support a real event.
- **Continuity**: consecutive events share participants or locations.
- **Outcome**: at least one event yields a clear result or turning point.

If these conditions are not satisfied, you must reject the chain. Use only the provided input and do not hallucinate external knowledge.

---

The event chain to evaluate is: {EVENT_CHAIN_INFO}
Return your answer strictly in the following JSON format:

```
{
  "is_plot": true or false,
  "reason": "Brief justification",
  "plot_info": {
    "title": "Unique, informative title (<=20 words)",
    "summary": "...",
    "main_characters": ["Character A", "Character B"],
    "locations": ["Location A", "Location B"],
    "time": "Approximate span",
    "theme": "Theme such as sacrifice, betrayal, reunion",
    "goal": "Main task or objective",
    "conflict": "Main obstacle or opposition",
    "resolution": "Result, e.g., success, failure, sacrifice"
  }
}
```

## Plot Relation Classification Prompt Template

You are a narrative structure analyst. Your task is to decide whether two given plots are related, and if so, classify the relation into one of six types.
Rules (must follow):

- Directed relations: – PLOT_PREREQUISITE_FOR: Plot A's outcome is a necessary precondition for Plot B. – PLOT_ADVANCES: Plot A promotes or facilitates Plot B, but is not strictly necessary. – PLOT_BLOCKS: Plot A obstructs or delays Plot B. – PLOT_RESOLVES: Plot A resolves or neutralizes the main conflict of Plot B.

- Undirected relations: – PLOT_CONFLICTS_WITH: The two plots pursue opposing goals or interests. – PLOT_PARALLELS: The two plots are structurally or thematically similar.

- If no sufficient evidence is found, return None.

- For directed relations, output the direction as A->B or B->A.

Input:
Plot A: {PLOT_A_INFO}
Plot B: {PLOT_B_INFO}
Return the result strictly in the following JSON format:

```
{
  "relation_type": "PLOT_PREREQUISITE_FOR",
  "direction": "A->B / B->A / null",
  "reason": "justification using goals, conflicts, ...",
  "confidence": 0.0
}
```

# F  QA Evaluation Details

## F.1  Long-Form Screenplay QA Benchmark Overview

Our QA evaluation is built on a larger ongoing project to construct a comprehensive *long-form screenplay understanding benchmark*. The benchmark covers full-length movie screenplays (20k–40k tokens each) with detailed annotations, including scene-structured scripts, entity and relation extraction, movie-specific knowledge graphs, and professionally authored question–answer pairs. Because the project is still under anonymized review and undergoing copyright verification, the dataset name is withheld and a partial version will be released only after publication.

**Motivation.**  Feature-length screenplays span tens of thousands of tokens and exhibit long-range dependencies across scenes, evolving character states, cross-scene causal chains, and recurring props or locations. A realistic QA benchmark for this setting must therefore evaluate: (i) structural grounding, (ii) temporal reasoning, (iii) cross-scene coherence tracking, and (iv) graph-aware inference over movie-specific knowledge.

**Practitioner Subset (PSQA-CN).**  The Practitioner Screenplay QA (PSQA-CN) subset is constructed from five full-length Chinese screenplays, each accompanied by a set of professionally authored questions. These questions were written by film directors, screenwriters, script supervisors, and other industry practitioners as part of their routine screenplay analysis workflow, where they identify scene structure, character motivations, object states, timeline relations, and cross-scene causal dependencies. As a result, the questions reflect the types of fine-grained reasoning that arise in real production settings rather than crowd-sourced or automatically generated prompts.

Table 6 summarizes the script lengths and their respective question counts. The screenplays range from approximately 13k to 84k chinese characters, covering youth romance, crime comedy, historical martial arts, science fiction, and art-house drama. This diversity in narrative scale and structural complexity yields a challenging benchmark for evaluating graph-based and tool-augmented reasoning systems, particularly on tasks involving scene continuity, temporal grounding, and multi-entity interactions.

Table 6: Chinese screenplays included in the practitioner QA subset, with script length (character count) and number of annotated questions.

| Film (English Title) | CN Characters | # QA |
|---|---|---|
| *The Grandmaster* | 12,965 | 49 |
| *Our Times* | 27,766 | 61 |
| *Farewell My Concubine* | 32,489 | 68 |
| *Let the Bullets Fly* | 70,135 | 80 |
| *The Wandering Earth II* | 83,562 | 45 |

The full category distribution is shown in Figure 14, highlighting a strong skew toward structural grounding and cross-scene reasoning.

**Per-movie annotations.**  Although the full benchmark will be released after publication, each movie in our internal dataset contains: (i) structured screenplay files, (ii) refined scene-level entity and relation extraction, (iii) canonical entity linking across scenes, (iv) attribute annotations, and (v) per-movie knowledge-graph schemas. These artifacts ensure that QA pairs can be grounded in both symbolic structure and raw script evidence.

**Release note.**  Due to copyright and anonymization constraints, the complete long-form benchmark will be released only after publication with its final dataset name and movie ID mapping. In this submission, we publicly release all materials associated with the *Practitioner Chinese Screenplay QA* subset used in the experiments, including the relevant screenplay segments, refined annotations, and QA pairs. These files are provided in anonymized form to enable full reproducibility of the results reported in this paper.

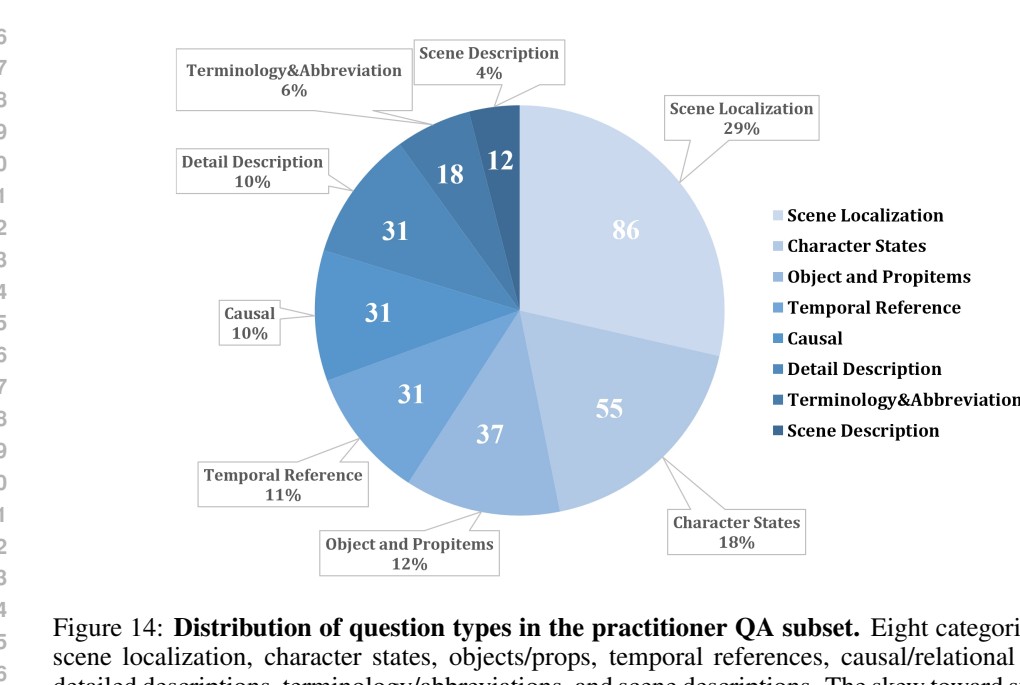

Figure 14: **Distribution of question types in the practitioner QA subset.** Eight categories cover scene localization, character states, objects/props, temporal references, causal/relational queries, detailed descriptions, terminology/abbreviations, and scene descriptions. The skew toward structural grounding motivates the need for graph-aware reasoning.

## F.2 LLM JUDGE PROMPT AND EVALUATION CRITERIA

To compare the quality of answers produced by different QA systems, we employ an LLM-based comparative evaluator. For each question, two system answers are presented to the evaluator along with an explicit evaluation criterion. The evaluator determines which answer better satisfies the criterion and outputs a structured JSON object. This procedure is repeated five times per (question, system pair, criterion), and the majority decision is used to compute the LLM-judge scores reported in our QA evaluation.

**Evaluation Criteria.** We use four complementary criteria designed to capture different dimensions of answer quality. These definitions correspond exactly to the criteria used in the evaluation scripts.

- **Comprehensiveness** — Evaluate which answer covers more aspects of the question while remaining concise and non-redundant. A comprehensive answer should provide all key information without omitting important elements.

- **Diversity** — Assess which answer offers more heterogeneous perspectives, dimensions, or types of information, rather than repeating a single narrow viewpoint.

- **Directness** — Determine which answer responds more directly and succinctly to the question, minimizing irrelevant or overly verbose content.

- **Empowerment** — Judge which answer better helps the user understand the underlying concept, make informed decisions, or reason further, while avoiding confusion or misleading statements.

These criteria jointly measure coverage, specificity, instructional quality, and informational breadth—four essential aspects of QA performance.

> **LLM Judge Comparative Evaluation Prompt Template**
>
> You are a helpful assistant responsible for grading two answers to a question that are provided by two different systems.
> Your task is to read the question and the two answers, then decide which answer is better according to the evaluation measure:

```
{CRITERIA_TEXT}
Rules (must follow):
```

- Judge **only** based on the given criterion; ignore other unrelated aspects.
- Identify how well each answer satisfies the criterion in relation to the question.
- Produce a decision: 1 = Answer 1 is better,   2 = Answer 2 is better,   0 = tie.
- If the two answers are similarly good or similarly poor under the criterion, output 0.
- Provide one concise reasoning sentence supporting your decision.
- Output must be valid JSON with no additional commentary.

**— Question —**

```
{QUESTION}
```

**— Answer 1 —** (SYSTEM1_NAME)

```
{ANSWER1}
```

**— Answer 2 —** (SYSTEM2_NAME)

```
{ANSWER2}
```

Return the result strictly in the following JSON format:

```
{
  "winner": 1 | 2 | 0,
  "reasoning": "Answer X is better because <your reasoning>."
}
```

### F.3   LLM-EVALUATED ANSWER CORRECTNESS PROMPT

To measure whether a system-generated answer is factually correct with respect to the reference answer, we employ an LLM-based correctness evaluator. For each (question, system answer, reference answer) triple, the evaluator is queried independently five times with stochastic sampling. Each run outputs a binary correctness label, and the final correctness score for that example is obtained by majority voting. As shown in Appendix F.4, this procedure yields high internal agreement and provides a stable correctness signal for QA evaluation.

**Correctness Criterion.**   The evaluator judges correctness under the following definition:

- **Correctness** — An answer is considered correct if it preserves the key factual meaning of the reference answer, does not contradict core facts, and does not introduce hallucinated or fabricated content. Minor variations in wording or level of detail are acceptable as long as the factual semantics match the reference answer.

The evaluator is instructed to be conservative: if an answer is only partially correct, omits critical information, or mixes correct facts with incorrect details, it should be labeled as incorrect.

**LLM Correctness Evaluation Prompt Template**

You are a careful factual evaluator. Your task is to determine whether the given answer is factually correct with respect to the reference answer.
Judge correctness under the following rule:

- Label the answer as *correct* if it matches the key factual meaning of the reference answer without contradictions or hallucinations.
- Label the answer as *incorrect* if it contradicts the reference, misses essential information, adds fabricated details, or is only partially aligned.

Return a JSON object with the structure:

```
{
  "is_correct": true | false,
  "reason": "Short explanation of the judgment."
```

```
}

— Question —
{QUESTION}
— System Answer —
{SYSTEM_ANSWER}
— Reference Answer —
{REFERENCE_ANSWER}
```

Evaluate whether the system answer is factually correct with respect to the reference answer. Output only the JSON.

### F.4 LLM EVALUATOR AGREEMENT

To assess the stability of the LLM-based correctness evaluator (Appendix F.3), we compute its internal agreement across five independent stochastic judgments for each (question, system answer, reference answer) triple. The evaluator is queried five times, producing five binary correctness labels used for majority voting in our evaluation.

Pairwise agreement is computed over all $\binom{5}{2} = 10$ annotator pairs per example. To quantify multi-annotator consistency, we additionally report Krippendorff's $\alpha$ for nominal-scale labels (Krippendorff, 2011). Let $n_{ic}$ denote how many times label $c \in \{0, 1\}$ appears among the five judgments for example $i$, and let $n_c$ be the total number of occurrences of label $c$ across the dataset. The observed and expected disagreements are

$$D_o = \frac{1}{N} \sum_{i=1}^{N} \frac{2\,n_{i0}n_{i1}}{5 \cdot 4}, \qquad D_e = \frac{2\,n_0 n_1}{(n_0 + n_1)^2 - (n_0^2 + n_1^2)},$$

and Krippendorff's $\alpha$ is

$$\alpha = 1 - \frac{D_o}{D_e}.$$

Because $\alpha$ measures how much the annotators agree beyond what would be expected by random labeling, it is well suited for evaluating the reliability of repeated stochastic LLM judgments.

Table 7: Internal agreement of the correctness evaluator across five independent judgments. Pairwise agreement is averaged over all $\binom{5}{2} = 10$ label pairs per example.

| Dataset | Pairwise Agreement | Krippendorff's $\alpha$ |
|---|---|---|
| NarrativeQA | 0.90 | 0.84 |
| PSQA-CN | 0.83 | 0.78 |

The evaluator shows strong internal consistency, achieving near-perfect agreement on shorter, factoid-style questions (NarrativeQA) and remaining highly stable on more open-ended, longer answers (PSQA-CN). Most disagreements arise in borderline or partially correct answers, confirming that majority voting provides a robust and reliable estimate of semantic correctness.

## G ADDITIONAL EXPERIMENTS AND ANALYSES

This appendix provides extended analyses that complement the main experiments in Section 4.

### G.1 END-TO-END RUNTIME, TOKEN CONSUMPTION, AND SCALING BEHAVIOR

We report the end-to-end processing cost of Narrative Knowledge Weaver across five full-length Chinese screenplays, covering all stages of the pipeline: metadata extraction, entity–relation extraction, attribute filling, CMP extraction, event-centric refinement, and normalization. Runtime measurements were obtained on a 16-thread CPU server with GPU-accelerated LLM calls.

**Overall runtime and token usage.** Table 8 summarizes total CN characters, end-to-end wall clock time, and token consumption. Total runtime scales linearly with input length, and token usage exhibits an almost perfect linear fit (Figure 16).

Table 8: End-to-end processing cost across five screenplays.

| Screenplay | Scenes | CN Characters | Tokens (M) | Time |
|---|---|---|---|---|
| The Grandmaster | 68 | 12,965 | 2.50 | 0.66h (39 min) |
| Our Times | 108 | 27,766 | 4.68 | 1.34h (81 min) |
| Farewell My Concubine | 100 | 32,489 | 5.78 | 1.55h (93 min) |
| Let the Bullets Fly | 184 | 70,135 | 11.78 | 3.33h (200 min) |
| The Wandering Earth II | 373 | 83,562 | 14.29 | 3.92h (235 min) |
| **Total** | **833** | **226,917** | **39.03** | **10.81h (648 min)** |

**Stage-level breakdown.** Across all inputs, the relative contributions of major stages remain stable (Table 9). Entity/Relation Extraction dominates due to reflection-based retries; Attribute Extraction and Event-Centric Refinement form the next largest components.

Table 9: Average stage-wise runtime proportion across all screenplays.

| Stage | Percentage | Description |
|---|---|---|
| Entity/Relation Extraction | 38.3% | Schema-based extraction with reflection |
| Attribute Extraction | 25.5% | Property-level enrichment |
| Event-Centric Refinement | 19.1% | Causal/temporal adjudication |
| CMP Extraction | 12.8% | Costume/Makeup/Props extraction |
| Entity Normalization & Disambig. | 4.3% | Coreference + canonicalization |
| Metadata Extraction / Text Split | 4.3% | Preprocessing pipeline |

**Scaling observations.** Empirically, the system exhibits the following properties:

- **Chunk inflation.** Actual chunk counts are $\approx 1.52\times$ theoretical (CN characters/600) due to scene/chapter boundary alignment.
- **Reflection-driven retries.** On average, each chunk undergoes **1.75 rounds** of extraction + reflection. Retry rates: Entity/Relation 75%, Attribute 70%, CMP 60%.
- **Parallel efficiency.** Running with 16 threads yields a practical efficiency of **70–75%**.
- **Throughput.** Average end-to-end throughput is **21k CN characters/hour**.
- **Linear scaling.** Token consumption vs. characters shows $R^2 = 0.999$ (Figure 16).

## G.2 ABLATION ON GRAPH PROBING.

We first ablate Graph Probing, the schema-induction step before extraction. Without probing, the average reflection score drops (6.83 vs. 7.21) and more attempts are required (2.71 vs. 2.33). This shows that probing improves both stability and efficiency of the extraction loop.

Table 10: Effect of Graph Probing.

| Variant | Reflection Score | Attempts |
|---|---|---|
| Full System | **7.21** | **2.33** |
| – w/o Probing | 6.83 | 2.71 |

## G.3 ABLATION ON ADAPTIVE ATTRIBUTE ENRICHMENT

We ablate the *Adaptive Attribute Enrichment* module introduced in Section 3.2.1, focusing on its contribution to downstream QA performance and the stability–variance tradeoff it introduces. This module enriches high-degree narrative entities through multi-granular evidence aggregation,

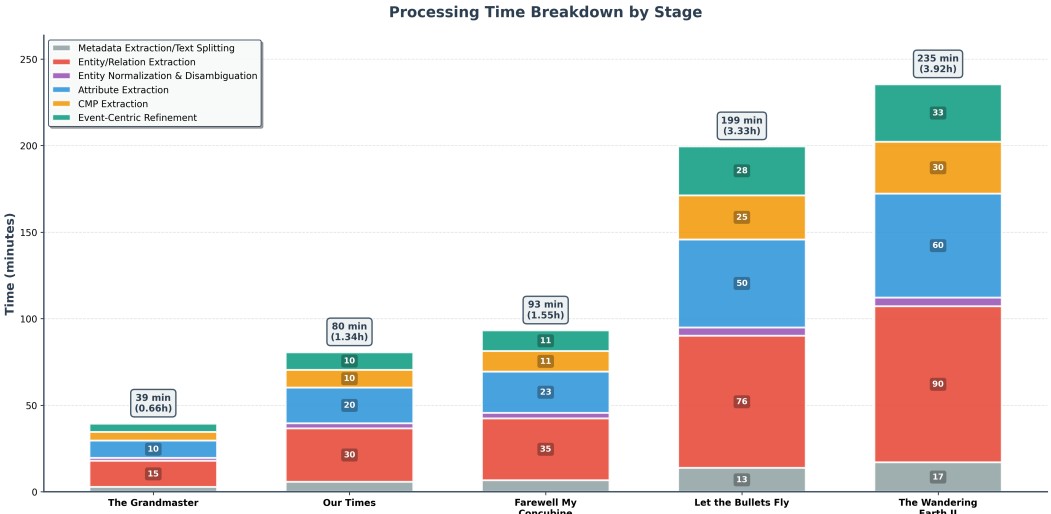

Figure 15: **Processing time by stage** for all screenplays.

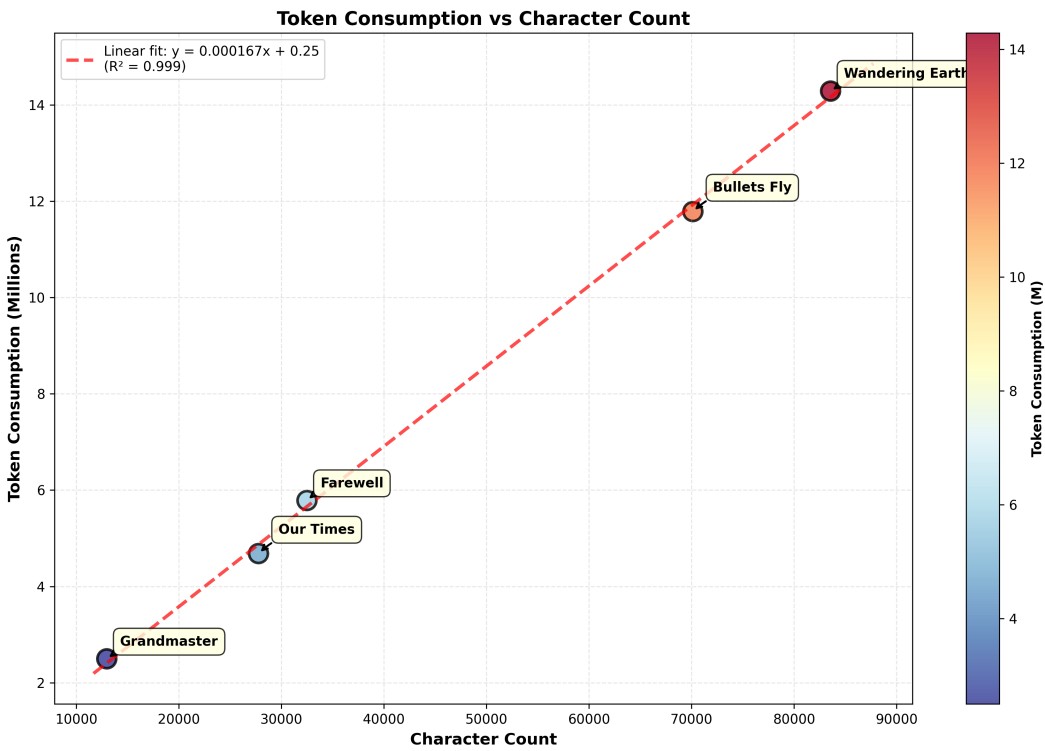

Figure 16: **Token consumption scales linearly** with character count across screenplays.

reflection-guided revision, and schema-adaptive updates. While this produces more informative and provenance-grounded entity profiles, the reflection-driven revision steps introduce additional stochasticity into the pipeline.

To quantify the effect of attribute enrichment, we compare three system configurations:

1. **Full System** — our adaptive enrichment module enabled, allowing incremental updates and schema extension.

2. **Default Schema Only** — attributes are extracted in a single pass using the static schema, without reflection-driven revision or new attribute induction.

3. **No Schema (Raw Descriptions)** — no structured attributes; entities retain only their extraction-time natural language descriptions.

Table 11 reports results across both benchmarks. On **NarrativeQA**, removing adaptive enrichment drops the LLM-judge accuracy from 76.4% to 74.1% under the default schema and further to 71.8% with no schema. On **PSQA-CN**, the effect is most visible at the *question-level*: question accuracy declines from 89.8% to 85.7% (default schema) and 82.0% (no schema). Answer-level accuracy follows a similar trend, though the performance gap between configurations narrows as the system relies less on structured attributes. Notably, the **No Schema** configuration shows the *lowest variance* across runs, consistent with the absence of reflection-driven revision or schema dynamics.

These results demonstrate that adaptive attribute enrichment substantially improves QA quality, particularly for questions requiring multi-entity grounding or fine-grained state tracking. However, similar to the event-centric refinement module, the gains come with the cost of higher variance, reflecting the inherent stochasticity of reflection-driven schema adaptation.

Table 11: Effect of Adaptive Attribute Enrichment on QA benchmarks.

| Variant | NarrativeQA (LLM-judge) Acc. | PSQA-CN (303 questions) Question Acc. | Answer Acc. |
|---|---|---|---|
| Full System | **76.4%** | **89.8%** | **74.3%** |
| Default Schema Only | 74.1% | 85.7% | 72.9% |
| No Schema (Raw Descriptions) | 71.8% | 82.0% | 71.2% |

## G.4 ABLATION ON EVENT-CENTRIC REFINEMENT.

We ablate the Event-centric Graph Refinement module (Section 3.2.4). Table 12 reports results on both benchmarks. On **NarrativeQA**, removing this module lowers the LLM-judge score from 76.4% to 72.1%. On **PSQA-CN**, question-level accuracy drops from 89.8% to 81.5%, and answer-level accuracy from 74.3% to 70.2%. These differences show that refinement mainly benefits questions requiring cross-event reasoning, including causal queries, temporal ordering, and plot-consistency checks.

Table 12: Effect of Event-centric Graph Refinement on QA benchmarks.

| Variant | NarrativeQA (LLM-judge) Acc. | PSQA-CN (303 questions) Question Acc. | Answer Acc. |
|---|---|---|---|
| Full System | **76.4%** | **89.8%** | **74.3%** |
| – w/o Refinement | 72.1% | 81.5% | 70.2% |

## G.5 ABLATION ON THE SLIDING SEMANTIC SPLITTER

We ablate the proposed sliding semantic splitter with two complementary evaluations: (i) an unsupervised segmentation study on Re-DocRED, and (ii) downstream knowledge graph (KG) construction. Together, these experiments examine both intrinsic segmentation quality and extrinsic impact on extraction accuracy.

**Segmentation metrics (formal definitions).** Let a document be segmented into $M$ segments $\{S_m\}_{m=1}^{M}$. Each $S_m$ contains $n_m$ sentence embeddings $\{x_{m,i}\}_{i=1}^{n_m}$ (L2-normalized), and let $\cos(\cdot, \cdot)$ denote cosine similarity. These metrics target three complementary aspects of discourse structure: cohesion within segments, separation across adjacent boundaries, and sharpness of the boundary transition itself.

*Intra-segment cohesion* (higher is better): for a single segment

$$\text{intra\_mean}(S_m) = \frac{2}{n_m(n_m - 1)} \sum_{1 \leq i < j \leq n_m} \cos(x_{m,i}, x_{m,j}),$$

and we report the average over all segments in a document, then average across documents.

*Adjacent-centroid similarity* (lower is better): with segment centroids $\bar{x}_m = \frac{1}{n_m} \sum_{i=1}^{n_m} x_{m,i}$,

$$\text{inter\_adjacent\_sim} = \frac{1}{M-1} \sum_{m=1}^{M-1} \cos(\bar{x}_m, \bar{x}_{m+1}).$$

*Boundary sharpness* (higher is better): for the boundary between $S_m$ and $S_{m+1}$, let $L_m$ be the last $k$ units of $S_m$ and $R_m$ the first $k$ units of $S_{m+1}$. Define

$$\Delta_m = \tfrac{1}{2}\Big(\text{mean\_sim}(L_m) + \text{mean\_sim}(R_m)\Big) - \text{cross\_sim}(L_m, R_m),$$

where $\text{mean\_sim}(A)$ is the mean pairwise similarity within $A$, and $\text{cross\_sim}(L, R)$ is the mean similarity across $L \times R$. We report

$$\text{boundary\_delta\_mean} = \frac{1}{M-1} \sum_{m=1}^{M-1} \Delta_m.$$

**Unsupervised results.** Table 13 shows that the semantic splitter improves intra-segment cohesion and boundary sharpness while reducing cross-boundary similarity. Compared to the recursive character-based baseline, which often introduces cuts within coherent spans, our method produces segments that are more discourse-consistent and yield cleaner transitions. Even small numerical gains across these metrics indicate a more faithful alignment between segment boundaries and semantic shifts.

Table 13: Segmentation quality on Re-DocRED (unsupervised). Higher is better for intra_mean and boundary_delta_mean; lower is better for inter_adjacent_sim.

| Method | Intra_mean ↑ | Inter_adjacent_sim ↓ | Boundary_delta_mean ↑ |
|---|---|---|---|
| Recursive Char Split (baseline) | 0.5174 | 0.7985 | 0.2460 |
| Sliding Semantic Split (ours) | **0.5232** | **0.7677** | **0.2471** |
| $\Delta$ (ours−baseline) | +0.0058 | −0.0308 | +0.0011 |

**Downstream KG construction.** Under the fixed Re-DocRED schema, removing the semantic splitter slightly decreases recall and F1 (Table 14). This confirms that discourse-aware segmentation stabilizes evidence aggregation for both entity and relation extraction. We find that the benefit manifests mainly in recall, indicating that more coherent segmentation reduces *missed links* by preserving complete evidence windows, while precision remains largely stable.

Table 14: KG construction ablation: with vs. without the sliding semantic splitter.

| Setting | Entity | | | Relation | | |
|---|---|---|---|---|---|---|
| | Recall | Precision | F1 | Recall | Precision | F1 |
| w/o splitter | 0.8021 | 0.7195 | 0.7584 | 0.3297 | 0.5660 | 0.4141 |
| Full | **0.8120** | **0.7190** | **0.7627** | **0.3360** | **0.5740** | **0.4239** |

**Qualitative observations.** Manual inspection highlights typical baseline errors, such as segmenting in the middle of dialogue turns or within a multi-sentence event description. The semantic splitter reduces these cases, producing segments that align with natural discourse boundaries. Remaining errors tend to occur in sections with high lexical overlap (e.g., repetitive summaries) or montage-like structures that rapidly alternate contexts.

**Takeaway.** Improved segmentation quality—captured by higher cohesion and sharper boundaries—translates into measurable gains in KG construction, primarily via higher recall and better overall F1. This suggests that segmentation is not just a preprocessing step but a critical design choice that directly shapes evidence windows and the reliability of extracted knowledge graphs.

### G.6 EMBEDDING SELECTION FOR RETRIEVAL AND ENTITY DISAMBIGUATION

We evaluate embeddings in two complementary roles within our framework: (i) **entity disambiguation / semantic locator** for the memory module, and (ii) **vector-database retrieval** for scene-level QA. Both evaluations are grounded in the same narrative sources used in our main experiments.

#### ENTITY DISAMBIGUATION (MEMORY/LEXICON ROLE)

**Task.** Given a surface mention $m$ from a scene, the model retrieves its canonical entity $e^*$ from a candidate inventory $\mathcal{E}$ by nearest-neighbor search in the embedding space. This evaluates how reliably embeddings align noisy mentions (nicknames, titles, transliterations) with canonical memory entries.

**Datasets.** We use the **Practitioner Screenplay QA benchmark** (5 feature films, 303 practitioner-driven questions; *ours*) and the **NarrativeQA-derived English Screenplays** (10 scripts) Kočiský et al. (2018). For each title, the canonical inventory is constructed in two stages. First, during knowledge-graph construction, we extract all *global-scope entities* (persisting across the entire narrative) and aggregate them into a candidate list. Second, professional annotators refine this list to ensure coverage and correctness of three core categories: *characters*, *objects*, and *concepts*. This process consolidates aliases, nicknames, and transliterations into canonical forms. Mentions used for evaluation are sampled from the same scenes appearing in the QA benchmark.

**Evaluation.** We annotate $\sim$200 mentions per film ($N$=1000 total) for the Practitioner Screenplay QA benchmark and $\sim$120 mentions per English script ($N$=1200 total). Candidate sets are evaluated in the **Closed-World** setting (inventory restricted to the same title). Hard negatives such as homographs and near-transliterations are included. Metrics are Recall@1 (R@1) and Precision.

Table 15: Entity disambiguation via retrieval-based linking. Results are reported as Recall@1 (R@1) and Precision (Prec, %). Closed-World setting.

| Model | PractitionerQA (5 films) | | English (10 scripts) | |
|---|---|---|---|---|
| | R@1 | Prec | R@1 | Prec |
| bge-large-zh | **98.40** | 97.40 | – | – |
| bge-large-en | – | – | 97.60 | 97.60 |
| bge-m3 | 95.80 | **97.80** | 98.20 | **98.20** |
| qwen3-embed-4B | 94.70 | 95.20 | 92.90 | 94.00 |
| qwen3-embed-8B | 97.90 | 87.70 | **98.80** | 89.40 |
| qwen3-embed-0.6B | 89.50 | 93.90 | 87.10 | 93.70 |
| m3e-large | 92.60 | 95.70 | 91.20 | 95.10 |
| all-MiniLM-L6-v2 | 28.90 | 94.80 | 84.70 | 95.40 |

**Findings.** For the Practitioner Screenplay QA benchmark, *bge-large-zh* achieves the highest recall (98.40%) with strong precision, while *bge-m3* attains the best precision (97.80%). For English screenplays, *bge-m3* and *bge-large-en* perform comparably, while *qwen3-8B* yields the highest recall (98.80%) but with lower precision due to over-triggering.

#### VECTOR-DATABASE RETRIEVAL (SCENE-LEVEL RECALL)

**Task.** Embeddings are also evaluated as retrieval keys for answering questions. For each scene or chapter we generate 1–2 question–answer pairs using a large language model, strictly grounded in the local passage. If a unit contains no answerable content, it is skipped. Given a question, embeddings are used to retrieve candidate scenes via dense similarity search.

**Datasets.** We again use the **Practitioner Screenplay QA benchmark** (5 films, $N{=}948$ questions; *ours*) and **NarrativeQA-derived English Screenplays** (10 scripts, $N{=}1250$ questions) Kočiský et al. (2018). QA pairs are generated with the following controlled prompt, which enforces self-contained questions, concise answers, and verbatim evidence spans:

---

**Scene-level QA Generation Prompt**

You are an expert question writer skilled in constructing answerable reading comprehension questions. Your task is to generate 1–2 question–answer pairs strictly grounded in the input passage.

**I. Task Description**

- Input: one scene or chapter from a screenplay/novel.
- Generate 1–2 self-contained, specific questions whose answers are explicitly stated in the passage.
- Each answer must be short and accurate ($\leq$30 Chinese characters or 20 English words).
- Do not use external knowledge or generate unanswerable questions.
- Avoid questions requiring exact numbers unless explicitly given in text.
- Each QA pair must also include verbatim supporting evidence from the passage.

**II. Output Format** Return strict JSON (no additional text):

```
{
  "results": [
    {"question": "Question text",
     "answer": "Answer text",
     "evidence": "Exact supporting span from passage"}
  ]
}
```

**III. Additional Requirements**

- If the passage contains no answerable content, return an empty array.
- Do not include explanations, comments, or formatting outside JSON.
- All strings must be valid JSON values to ensure parsing.

**IV. Text to be Processed**

```
{PASSAGE}
```

---

**Evaluation.** We report **Top-1** and **Top-5** scene-level recall against annotated provenance. Retrieval is performed over scene-level embeddings, and results are aggregated per corpus.

Table 16: Scene-level retrieval on the Practitioner Screenplay QA benchmark and NarrativeQA-derived English scripts. Percentages relative to annotated provenance.

| Model | Practitioner Screenplay QA (5 films) | | English (10 scripts) | |
| --- | --- | --- | --- | --- |
| | Top-1 | Top-5 | Top-1 | Top-5 |
| bge-m3 | **55.40** | **74.20** | **58.80** | **78.40** |
| bge-large-zh | 50.10 | 67.50 | – | – |
| bge-large-en | – | – | 52.80 | 72.80 |
| qwen3-embed-8B | 50.70 | 72.10 | 54.40 | 74.40 |
| qwen3-embed-4B | 51.10 | 72.70 | 52.00 | 72.40 |
| qwen3-embed-0.6B | 49.10 | 70.30 | 48.00 | 68.80 |
| m3e-large | 38.20 | 59.20 | 41.60 | 62.40 |
| all-MiniLM-L6-v2 | 9.50 | 20.90 | 32.00 | 56.00 |

**Findings.** Across both corpora, *bge-m3* delivers the strongest retrieval performance, achieving 55.40% / 74.20% (Top-1/Top-5) on the Practitioner Screenplay QA benchmark and 58.80% / 78.40% on English scripts. *qwen3-8B* and *qwen3-4B* are competitive, while smaller models (e.g., *all-MiniLM-L6-v2*) lag significantly.

### G.7 ABLATION ON STRUCTURED EVENT REPRESENTATIONS

This ablation quantifies how *structured event cards* (context engineering) influence both causal graph extraction and downstream plot induction. Each screenplay can be processed under two conditions: (i) *with event cards*, where each event is summarized into a structured representation with participants, actions, and outcomes; and (ii) *without event cards*, where only the raw text span and linked entities are provided, without any summarization. We report results on one representative title (*The Wandering Earth 2*); the other four Chinese screenplays in our benchmark show nearly identical trends. For each condition, we performed ten independent runs and aggregated the outcomes.

**Causal-graph level.** Event cards improve run-to-run stability and lead to a systematic shift in edge labeling. With event cards, intra-run Jaccard rises to $0.512$ (vs. $0.386$ without), indicating higher reproducibility. Despite the relatively low overlap between consensus graphs from the two conditions (Jaccard $= 0.284$), both remain acyclic and admit transitive reduction, condensing to compact backbones (A: $497{\rightarrow}193$, B: $330{\rightarrow}151$).

To assess whether the two strategies differ significantly at the edge level, we use paired statistical tests. For the binary outcome (`causal` vs. `none`), we apply *McNemar's test*. Given a $2{\times}2$ table of paired decisions

|  | B: causal | B: none |
|---|---|---|
| A: causal | $n_{11}$ | $n_{10}$ |
| A: none | $n_{01}$ | $n_{00}$ |

the test statistic is

$$\chi^2_{\text{McNemar}} = \frac{(n_{10} - n_{01})^2}{n_{10} + n_{01}},$$

which under the null hypothesis of symmetry follows a $\chi^2$ distribution with 1 degree of freedom. In our case, $n_{10}{=}740$, $n_{01}{=}228$, yielding $p{=}1.072{\times}10^{-63}$, showing that the no-card condition systematically assigns more edges as causal.

For the multi-class outcome (`CAUSES`, `INDIRECT_CAUSES`, `NONE`), we apply *Bowker's test of symmetry*, a generalization of McNemar. For each off-diagonal pair $(i, j)$, it computes

$$\chi^2_{\text{Bowker}} = \sum_{i<j} \frac{(n_{ij} - n_{ji})^2}{n_{ij} + n_{ji}},$$

which follows a $\chi^2$ distribution with $k(k-1)/2$ degrees of freedom for $k$ categories. This tests whether relabeling asymmetries exist (e.g., edges more often reassigned from `NONE` to `INDIRECT_CAUSES` than vice versa). We obtain $\chi^2{=}290.419$, $df{=}3$, $p{=}1.195{\times}10^{-41}$, confirming a strong directional distributional shift between strategies.

Table 17: Causal-graph level ablation on *The Wandering Earth 2*. Stability is measured by intra-run Jaccard; edge-level consistency is evaluated by McNemar's test on (`causal` vs. `none`) and by Bowker's test on (`CAUSES`, `INDIRECT_CAUSES`, `NONE`).

| Metric | With Event Cards (B) | Without Event Cards (A) |
|---|---|---|
| Intra-run Jaccard (mean $\pm$ std) | $0.512 \pm 0.057$ | $0.386 \pm 0.071$ |
| Consensus Jaccard (A vs. B) | $0.284$ | |
| Consistency (McNemar; $n_{10}$, $n_{01}$, $p$) | $(740, 228, 1.072{\times}10^{-63})$ | |
| Multi-class paired (Bowker; $\chi^2$, $df$, $p$) | $(290.419, 3, 1.195{\times}10^{-41})$ | |
| Edges $\rightarrow$ after transitive reduction | $330 \rightarrow 151$ | $497 \rightarrow 193$ |

**Plot level.** Effects at the causal-graph layer propagate to plot induction. Event cards yield more compact plot structures, lower run-to-run variance, and more balanced distributions of relation types. For instance, the number of plots with event cards is $180.5 \pm 13.6$ (vs. $392.3 \pm 28.7$ without), while inter-plot relations remain well distributed across categories. Full results are given in Table 18.

Table 18: Plot-level ablation on *The Wandering Earth 2*: impact of event cards on induced plot graphs.

| Relation Type | With Event Cards | | Without Event Cards | |
|---|---|---|---|---|
| | Mean $\pm$ Std | Ratio (%) | Mean $\pm$ Std | Ratio (%) |
| **Plots** | $180.5 \pm 13.6$ | – | $392.3 \pm 28.7$ | – |
| **Total Relations** | $612.9 \pm 46.2$ | – | $1284.2 \pm 109.9$ | – |
| PLOT_ADVANCES | $10.0 \pm 3.0$ | 1.6 | $31.5 \pm 7.5$ | 2.4 |
| PLOT_BLOCKS | $8.7 \pm 2.8$ | 1.4 | $6.1 \pm 2.5$ | 0.5 |
| PLOT_CONFLICTS_WITH | $15.2 \pm 7.1$ | 2.5 | $11.5 \pm 6.4$ | 0.9 |
| PLOT_PARALLELS | $160.4 \pm 26.0$ | 26.2 | $302.2 \pm 54.7$ | 23.5 |
| PLOT_PREREQUISITE_FOR | $410.7 \pm 33.0$ | 67.0 | $912.8 \pm 70.8$ | 71.1 |
| PLOT_RESOLVES | $7.9 \pm 3.9$ | 1.3 | $20.2 \pm 5.6$ | 1.6 |

**Takeaways.** Event cards (i) improve intra-run stability of causal judgments; (ii) induce a statistically significant distributional shift in edge labels—toward *fewer* causal assignments overall relative to the no-card baseline (as shown by McNemar and Bowker tests); and (iii) yield compact, regular cores after transitive reduction. These effects propagate to plot induction, producing fewer yet more coherent plots with balanced inter-plot relations. Overall, structured event representations regularize causal extraction and promote reproducibility and interpretability across levels of abstraction.

### G.8 COMPARISON WITH STRUCTURAL CYCLE-BREAKING BASELINES

To assess the effectiveness of SABER in preserving narratively meaningful causal structure, we compare it against two purely structural pruning baselines using the event graphs constructed for the **Practitioner Screenplay QA (PSQA-CN)** benchmark.

**Baseline 1: DFS Back-Edge Removal (DFS-BER).** A depth-first search is run on the adjudicated event graph. Whenever a back-edge $u \to v$ is detected (i.e., $v$ is currently on the recursion stack), a directed cycle is recorded. For each detected cycle, the edge with the *lowest confidence* is removed. Because DFS only detects ancestor–descendant cycles and depends on traversal order, DFS-BER is sensitive to initialization and may remove high-confidence edges that are narratively important.

**Baseline 2: Greedy Cycle-Breaking (GCB).** All causal edges are globally sorted by descending confidence (ties broken by node importance). Edges are added to an initially empty graph in sorted order. An edge $a \to b$ is included only if $b$ is not reachable from $a$ in the current graph, ensuring that no directed cycle is introduced. This Kruskal-style greedy strategy yields a maximum-confidence acyclic subgraph, but does not address flattened causal structures (e.g., $A \to B \to C$ vs. $A \to C$) or semantic plausibility.

**SABER.** Our full pruning pipeline (Appendix E.2) integrates (i) confidence-guided cycle removal within strongly connected components, (ii) detection and resolution of flattened causal patterns, and (iii) LLM-based semantic adjudication between alternative paths. Thus SABER combines structural and semantic evidence to preserve coherent, multi-hop causal progressions.

### EXPERIMENT A: SEMANTIC COHESION OF PLOT EVENT SETS

For each pruning method, we reconstruct the Event Plot Graph and compute **intra-plot semantic cohesion** as the average pairwise cosine similarity between event embeddings within each generated plot unit. Higher values indicate more coherent groupings of events around shared goals, participants, and outcomes.

Results on the PSQA-CN corpus are shown in Table 19.

SABER achieves the highest semantic cohesion, reflecting its ability to remove redundant shortcut edges and preserve events that form narratively consistent causal segments. GCB, while stronger than DFS-BER in retaining high-confidence edges, still produces plots with lower internal coherence due to unresolved flattened structures.

Table 19: Semantic cohesion of plot event sets on PSQA-CN (average pairwise cosine similarity).

| Method | Mean | Median | Std |
|---|---|---|---|
| DFS-BER | 0.39 | 0.38 | 0.07 |
| GCB | 0.48 | 0.47 | 0.06 |
| **SABER** | **0.57** | **0.56** | **0.05** |

EXPERIMENT B: EVENT-SENSITIVE QA ACCURACY

Among the 303 PSQA-CN questions, we identify 67 *event-sensitive* queries that at least trigger event-related tools once in 5 trials:

- `retrieve_entity_by_name` with entity type = `Event`;

- `query_similar_entities` involving event embeddings;

- `find_related_events_and_plots`.

We rebuild the event graph and Event Plot Graph under each pruning method and evaluate TARA on the same 67 queries.

Table 20 summarizes the results.

Table 20: Correctness on 67 event-sensitive PSQA-CN questions, evaluated using the LLM-based correctness assessor with five-way majority voting.

| Method | Accuracy |
|---|---|
| DFS-BER | 0.55 |
| GCB | 0.57 |
| **SABER** | **0.61** |

The correctness values in Table 20 are computed using the the same procedure as in Section 4.3, in which each prediction is judged five times under stochastic sampling, and correctness is determined by majority voting .

Under this evaluation protocol, GCB achieves a small but consistent improvement over DFS-BER. Because GCB preserves high-confidence causal edges during cycle-breaking, it produces fewer prematurely truncated event chains, enabling TARA to retrieve relevant events more reliably.

SABER obtains the highest correctness score. By resolving flattened causal patterns and employing semantic adjudication to choose between alternative causal paths, SABER better preserves multi-hop narrative structure. As a result, answers generated using SABER's event and plot graph contain fewer factual omissions or conflicts relative to the reference answers, leading to improved correctness on event-sensitive PSQA-CN queries.

**Summary.** Structural cycle-breaking strategies (DFS-BER, GCB) remove cycles but do not adequately preserve narrative causality. SABER consistently produces more coherent plot structures and yields the highest accuracy on event-sensitive PSQA-CN queries, highlighting the importance of semantic-aware causal pruning.

## H   TOOL-AUGMENTED REASONING AGENT (TARA).

This appendix provides detailed documentation of the components supporting the **Tool-Augmented Reasoning Agent (TARA)**. We include (i) the full specification of the tool layer used by TARA during inference, (ii) comprehensive tool-usage statistics across both NarrativeQA and PSQA-CN, and (iii) the complete strategy library employed to stabilize tool selection for practitioner questions. The main text (Section 4.3) provides a high-level summary of the key findings.

### H.1   IMPLEMENTED TOOLS

We design a tool layer spanning three categories—(i) **Graph-based** utilities operating on the Neo4j knowledge graph, (ii) **Vector-based** utilities built on embedding search, and (iii) **Native** utilities for sparse retrieval and structured SQL queries. Table 21 lists the final set used by the TARA.

| Category | Tool Name | Input (key params) | Output / Usage |
|---|---|---|---|
| Graph-based | retrieve_entity_by_name | query, entity_type | Fuzzy keyword/alias search scoped by entity type |
| | retrieve_entity_by_id | entity_id, contain_properties, contain_relations | Detailed entity info; optionally include properties/relations |
| | search_related_entities | source_id, predicate, relation_types, entity_types, limit, return_relations | Adjacent entities; optionally return (entity, relation) pairs |
| | get_relation_summary | src_id, tgt_id, relation_type | Human-readable relation summary between two entities |
| | get_common_neighbors | id1, id2, rel_types, direction, limit | Shared neighbors; optionally list edge types from A/B |
| | find_paths_between_nodes | src_id, dst_id, max_depth, limit | Natural-language paths (evidence chains) with node/edge descriptions |
| | top_k_by_centrality | metric, top_k, node_labels | Rank nodes by PageRank / Degree / Betweenness |
| | get_co_section_entities | entity_id, include_types | Co-occurring entities in the same scene/chapter |
| | get_k_hop_subgraph | center_ids, k, limit_nodes | k-hop neighborhood subgraph |
| | find_related_events_and_plots | entity_id, max_depth | Events linked to the node and their Plots, with path sketch |
| | query_similar_entities | text, top_k, entity_types, include_meta | Nearest entities via vector index |
| Vector-based | vdb_search_hierdocs | query, limit | Parent–child retrieval (sentence recall with doc aggregation) |
| | vdb_search_docs | query, limit | Document-level retrieval |
| | vdb_search_sentences | query, limit | Sentence-level retrieval |
| | vdb_get_docs_by_chunk_ids | chunk_ids | Fetch paragraphs by chunk IDs |
| Native | bm25_search_docs | query, k | BM25 keyword retrieval |
| | search_by_character | character keyword | Query CMP information by character |
| | search_by_scene | scene/subscene name | Query CMP information by scene metadata |
| | chunk_to_scene | chunk_ids | Map chunks to scene/subscene |
| | scene_to_chunks | scene/subscene | List all chunks in a scene |
| | nlp2sql_query | natural-language query | LLM-to-SQL for costume/-makeup/prop database |

Table 21: Final tool set grouped into Graph-based, Vector-based, and Native utilities.

### H.2   TOOL USAGE IN QA

Table 22 reports the full per-tool usage statistics for NarrativeQA and the Practitioner Screenplay QA benchmark. Figure 5 in the main text provides a visual summary of these distributions.

Table 22: Tool usage comparison across NarrativeQA and the Practitioner Screenplay QA benchmark (ours).

| Rank | Tool | NarrativeQA | | PractitionerQA | |
|---|---|---|---|---|---|
| | | Calls | % | Calls | % |
| 1 | retrieve_entity_by_name | 5781 | 40.6 | 1784 | 36.8 |
| 2 | search_related_entities | 2592 | 18.2 | 659 | 13.6 |
| 3 | find_related_events_and_plots | 1392 | 9.8 | 726 | 15.0 |
| 4 | retrieve_entity_by_id | 954 | 6.7 | 319 | 6.6 |
| 5 | vdb_get_docs_by_chunk_ids | 766 | 5.4 | 199 | 4.1 |
| 6 | get_relation_summary | 705 | 5.0 | 239 | 4.9 |
| 7 | query_similar_entities | 471 | 3.3 | 159 | 3.3 |
| 8 | vdb_search_docs | 403 | 2.8 | 119 | 2.5 |
| 9 | vdb_search_hierdocs | 328 | 2.3 | 99 | 2.0 |
| 10 | bm25_search_docs | 313 | 2.2 | 99 | 2.0 |
| 11 | find_paths_between_nodes | 233 | 1.6 | 79 | 1.6 |
| 12 | get_common_neighbors | 150 | 1.1 | 59 | 1.2 |
| 13 | vdb_search_sentences | 49 | 0.3 | 39 | 0.8 |
| 14 | get_k_hop_subgraph | 37 | 0.3 | 31 | 0.6 |
| 15 | get_co_section_entities | 30 | 0.2 | 27 | 0.6 |
| 16 | chunk_to_scene | 15 | 0.1 | 23 | 0.5 |
| 17 | search_by_scene | 11 | 0.1 | 19 | 0.4 |
| 18 | top_k_by_centrality | 11 | 0.1 | 15 | 0.3 |
| 19 | scene_to_chunks | 7 | 0.1 | 11 | 0.2 |
| 20 | search_by_character | 7 | 0.1 | 19 | 0.4 |
| 21 | nlp2sql_query | – | – | 121 | 2.5 |

## H.3 ABLATION STUDY ON LOW-FREQUENCY TOOLS

Table 22 shows that several tools in our library are used in fewer than 2% of model-invoked tool calls. A reviewer naturally asks whether these tools meaningfully contribute to TARA's performance, or whether they can be removed without harming QA capability.

To address this question, we perform a controlled ablation study, using the same five-sample evaluation protocol as in the main PSQA-CN experiment (Table 3). For each low-frequency tool, we disable only that tool while keeping the rest of the system unchanged, and re-evaluate TARA on the PSQA-CN benchmark. Because answer-level correctness (per-sample accuracy) is more sensitive to breakdowns in multi-step reasoning than question-level correctness, we report answer-level accuracy for this analysis. The full results are provided in Table 23.

Table 23: Ablation of low-frequency tools on PSQA-CN (Answer-level correctness). Baseline = 74.3%.

| Ablated Tool | Acc. (%) | Drop |
|---|---|---|
| *Tools with negligible impact* | | |
| get_k_hop_subgraph | 74.1 | -0.2 |
| vdb_search_sentences | 74.0 | -0.3 |
| top_k_by_centrality | 73.9 | -0.4 |
| *CMP-specific tools (low use but necessary when triggered)* | | |
| search_by_character | 73.3 | -1.0 |
| search_by_scene | 73.1 | -1.2 |
| nlp2sql_query | 71.5 | -2.8 |
| *Cross-indexing tools* | | |
| scene_to_chunks | 72.4 | -1.9 |
| chunk_to_scene | 72.1 | -2.2 |

**Tools with negligible impact.** Three low-frequency tools—get_k_hop_subgraph, vdb_search_sentences, and top_k_by_centrality—lead to less than a 0.5-point

drop when removed. Their functionality is largely superseded by higher-usage operators: `find_paths_between_nodes` and `get_relation_summary` already cover most multi-hop structural queries, while `vdb_search_docs` retrieves the necessary textual evidence without requiring sentence-level granularity. Although these low-frequency tools can be considered optional for PSQA-CN, they may still be valuable in domains with denser graph structure or sentence-sensitive reasoning.

**Tools critical for cross-index reasoning.** `scene_to_chunks` and `chunk_to_scene` are used in fewer than 1% of calls but are essential for bridging symbolic KG structure with textual chunks. Disabling either tool produces a noticeable 2.0–2.2 point drop in answer-level correctness. This confirms that cross-indexing—even when invoked infrequently—is crucial for multi-hop reasoning that requires aligning retrieved events with their supporting text.

**Tools for CMP-specific reasoning.** The three lowest-frequency tools— `search_by_character`, `search_by_scene`, and `nlp2sql_query`—are all tied to CMP (costume/makeup/props) metadata, which appears only in a subset of screenplays and questions. As a result, they are invoked infrequently, yet their removal yields noticeable performance drops (0.8–1.2 points for the first two, and 2.8 points for `nlp2sql_query`). When CMP information is required, these tools provide retrieval and disambiguation capabilities that cannot be replicated by KG traversal or semantic search, making them essential despite their low activation frequency.

**Summary.** Although many tools appear in less than 2% of tool calls, low frequency does not imply dispensability. Some tools are redundant and can be safely removed, while others—especially those enabling cross-indexing or access to structured metadata—provide capabilities that no other operator can substitute. These findings illustrate the importance of maintaining a diverse tool palette: most queries rely on a compact core of operators, but rare tools supply the specific reasoning affordances required for full coverage on narrative QA.

## H.4 STRATEGY LIBRARY AND ABLATION DETAILS

For the Practitioner Screenplay QA benchmark, domain experts distilled a **strategy library** encoding soft preferences for tool selection by question type. This library is intended to stabilize tool usage rather than to improve raw capability, and is used only for the Practitioner benchmark (not for NarrativeQA). The decision guide is summarized in Table 25.

**Impact on QA performance.** The strategy library primarily reduces variance in TARA's multi-step reasoning by steering the agent toward more stable tool-use trajectories. As shown in Table 24, question-level accuracy remains essentially unchanged, while answer-level accuracy improves notably due to more consistent evidence aggregation.

Table 24: Ablation on the Practitioner Screenplay QA benchmark: effect of the strategy library on stability and accuracy.

| Method | Question Acc. | Answer Acc. |
|---|---|---|
| Agent (w/o Strategy Library) | **89.8%** | 74.3% |
| Agent (+ Strategy Library) | 88.9% | **79.6%** |

**Effect on tool usage.** Table 26 further compares tool usage distributions with and without the strategy library, complementing the ablation above. The library shifts usage away from generic entity retrievers toward more targeted event-centric and structured tools, reflecting improved alignment between question types and tool selection.

Table 26 further details how the strategy library reshapes tool usage on the Practitioner Screenplay QA benchmark, complementing the ablation in Table 24 in the main text.

Table 25: Strategy library for tool selection, authored by domain experts for the Practitioner Screenplay QA benchmark.

| Question Type | Preferred Tool Usage |
|---|---|
| Abbreviations, terminology | Keyword retrieval (`bm25_search_docs`) or entity retriever (`retrieve_entity_by_name`); cross-validate multiple passages if needed. |
| Character appearance, behavior, costume | Character/entity retrievers; if scene information is needed, combine with `chunk_to_scene`. |
| Event timeline and outcomes | Prioritize event graph utilities (`find_related_events_and_plots`, `search_related_entities`); supplement with vector-based retrieval for supporting context. |
| Props and state changes | Use keyword retrieval to locate first occurrence, then vector retrieval for contextual updates. |
| Locations and scene changes | Retrieve location entities by name, then complement with event/scene mapping (`chunk_to_scene`). |
| Equipment definitions and attributes | Use structured query (`nlp2sql_query`) for costume/makeup/prop tables; combine with retrieval for context if linked to characters or scenes. |
| High-risk queries (appearance details, enumerations, fictional timelines) | Prioritize keyword retrieval from original documents (`bm25_search_docs`) before generating an answer. |

Table 26: Comparison of tool usage distribution on the Practitioner Screenplay QA benchmark, with and without the strategy library.

| Rank | Tool | w/o Strategy Library (%) | + Strategy Library (%) |
|---|---|---|---|
| 1 | `retrieve_entity_by_name` | 36.8 | 25.5 |
| 2 | `find_related_events_and_plots` | 15.0 | 21.0 |
| 3 | `search_related_entities` | 13.6 | 8.5 |
| 4 | `retrieve_entity_by_id` | 6.6 | 2.5 |
| 5 | `get_relation_summary` | 4.9 | 0.5 |
| 6 | `vdb_get_docs_by_chunk_ids` | 4.1 | 12.5 |
| 7 | `query_similar_entities` | 3.3 | 0.7 |
| 8 | `nlp2sql_query` | 2.5 | 5.5 |
| 9 | `vdb_search_docs` | 2.5 | 0.7 |
| 10 | `bm25_search_docs` | 2.0 | 9.5 |
| 11 | `vdb_search_hierdocs` | 2.0 | 0.5 |
| 12 | `find_paths_between_nodes` | 1.6 | 0.0 |
| 13 | `get_common_neighbors` | 1.2 | 0.0 |
| 14 | `vdb_search_sentences` | 0.8 | 0.0 |
| 15 | `get_k_hop_subgraph` | 0.6 | 0.0 |
| 16 | `get_co_section_entities` | 0.6 | 2.0 |
| 17 | `chunk_to_scene` | 0.5 | 6.5 |
| 18 | `search_by_scene` | 0.4 | 4.5 |
| 19 | `search_by_character` | 0.4 | 2.7 |
| 20 | `top_k_by_centrality` | 0.3 | 0.3 |
| 21 | `scene_to_chunks` | 0.2 | 0.0 |

## H.5 DETAILED PER-CATEGORY QA ANALYSIS

To understand where graph-aware reasoning provides the largest benefits, we analyze accuracy separately for each practitioner-defined question category. The practitioner benchmark contains eight categories reflecting structural grounding, temporal reasoning, causal inference, object and prop tracking, and fine-grained narrative details.

**Per-category accuracy.** Table 27 compares **TARA (Ours)** with the stronger **Hybrid Retriever** baseline used in our main evaluation. Results are based on the full **PSQA-CN** benchmark of **303** practitioner-authored questions. Across all eight categories, TARA yields consistently higher accuracy. The largest improvements occur in *scene localization*, *character states*, and *temporal and causal reasoning*, where resolving a question requires integrating evidence across multiple scenes or using structured knowledge to maintain narrative consistency. Categories relying primarily on surface lookup (e.g., terminology or abbreviations) show smaller gains, as expected.

Table 27: Per-category QA accuracy (%) on the PSQA-CN benchmark (303 questions). TARA integrates graph reasoning, while the baseline uses hybrid sparse–dense retrieval.

| Category | TARA(Ours) | Hybrid Retriever | Δ | Example |
|---|---|---|---|---|
| Scene localization | 92.9 | 47.6 | +45.3 | "Which scenes take place in Africa?" |
| Character states | 90.7 | 44.4 | +46.3 | "Does Tu Hengyu have a scar near his right eye at age 29?" |
| Objects and props | 85.4 | 60.3 | +25.1 | "What ring did Liu Peiqiang use to propose to Han Duoduo?" |
| Temporal references | 77.8 | 50.0 | +27.8 | "In which year was the back-up plan to ignite the Moon proposed?" |
| Causal/relational | 64.3 | 22.2 | +42.1 | "How did Tu Hengyu waterproof his laptop underwater?" |
| Detail descriptions | 72.7 | 34.1 | +38.6 | "What did Ma Zhao write in his suicide note?" |
| Terminology/abbreviations | 66.7 | 63.0 | +3.7 | "What is the full form of FRAMER?" |
| Scene descriptions | 87.5 | 15.6 | +71.9 | "What was shown on the screen during the UEG assembly?" |

**Error profile.** Most of the baseline's incorrect predictions arise in categories where answering requires linking entities to specific scenes, maintaining temporal order, or following cross-scene causal dependencies. TARA improves accuracy in these settings by grounding its reasoning in the narrative KG and event-centric structure, which stabilizes entity references and enforces consistent temporal and causal constraints.

## H.6 ERROR ANALYSIS OF LOW-CORRECTNESS QUESTIONS

To better understand the error patterns underlying the QA results reported in the main text, we conduct a qualitative analysis of the PSQA-CN questions on which TARA achieves low correctness under the five-sample evaluation protocol. Rather than running an additional evaluation, we directly examine the same per-question correctness scores reported in the main experiment and analyze the types of questions for which TARA's multi-step reasoning most frequently fails. Questions with low question-level or answer-level correctness reveal specific forms of narrative understanding that remain challenging for the system.

We focus on two groups: **completely incorrect questions** (0/5) and **mostly incorrect questions** (1–2/5). These groups highlight distinct failure modes across the eight narrative question categories used in PSQA-CN: scene localization, character states, objects/props, temporal references, causal/relational queries, detailed descriptions, terminology/abbreviations, and scene descriptions.

**A. Completely incorrect questions (0/5).** The following representative examples illustrate patterns where TARA fails to ground any of the five answers:

- **Q0 (Scene Localization).** *"Where did the county magistrate Lao Tang depart to take office?"* The scene involves brief geographical references embedded within transitional narration. TARA often retrieves related scenes but fails to isolate the precise location, yielding repeatedly incorrect answers.

- **Q22 (Character State).** *"What was Xiaoliuzi's state during the conversation?"* The relevant evidence is conveyed through subtle behavioral cues. TARA consistently provides high-level paraphrases rather than grounded micro-state descriptions, leading to unanimous incorrect judgments.

- **Q23 (Scene Description).** *"Where did the scene of the adviser being scalded to death occur?"* This event is embedded within a multi-part action sequence. TARA struggles to extract the correct sub-location, producing answers with plausible but incorrect scene anchors.

These cases share two characteristics: (i) the gold information is narrowly localized and appears only once in the narrative, and (ii) retrieval frequently returns broader contextual segments that obscure the ground-truth detail.

**B. Mostly incorrect questions (1–2 out of 5).** A second group consists of questions where TARA occasionally succeeds but remains inconsistent:

- **Q19 (Character Action).** *"What did Zhang Muzhi do after firing the shot?"* Although the relevant action is described explicitly, retrieval sometimes surfaces adjacent dialogue rather than the action frame, resulting in partially grounded but often incomplete answers.

- **Q33 (Detailed Description).** *"What bodily movements did Huang Silang display while observing the distant scene?"* Micro-action descriptions tend to be brief and dispersed; answers vary between correct fine-grained descriptions and incorrect generalizations, yielding mixed correctness.

- **Q73 (Character State + Role Description).** *"Where does Lao Wu fall in age among his brothers, and what personal state did he reveal?"* This requires integrating two pieces of information presented far apart in the narrative. TARA intermittently retrieves only one of them, creating partial answers that the evaluator marks as incorrect.

Mostly incorrect questions tend to require multi-hop integration of dispersed details (e.g., action + motivation, ordering + state), where retrieval noise or over-generalization leads to inconsistent outcomes across the five attempts.

**Summary.** Low-correctness questions reveal four major challenge clusters across the eight-category taxonomy:

- **Fine-grained scene localization** in composite or transitional scenes.

- **Subtle character behavioral or psychological states** expressed through micro-actions or indirect cues.

- **Multi-hop detail synthesis** across distant narrative segments.

- **Complex scene descriptions** involving intertwined spatial and event cues.

These systematic errors suggest directions for future improvements, including hierarchical sub-location modeling, richer representations of behavioral and affective states, and enhanced cross-segment evidence aggregation.

# I  DOWNSTREAM APPLICATIONS

## I.1  PRODUCTION CONTINUITY CHECKING

This section describes a downstream application enabled by the explicit scene–entity structure of our narrative knowledge graphs: *production continuity checking*. The goal is to determine whether two scenes can be filmed using the same production setup—a practical consideration in film and television where set layout, dressing, lighting, and core costume/props must remain visually coherent across shots. We implement continuity checking as an agent-style pipeline that combines graph-derived structural cues, LLM-based pairwise judgments, and an LLM reflection step for chain-level refinement.

**Overall pipeline.**  The continuity checker operates as an agent on top of the narrative KG. Rather than a fixed sequence of operations, the agent iteratively interacts with the graph, queries scene metadata, performs LLM-based judgments, and updates its internal continuity hypotheses. Each cycle consists of three core behaviors:

- **Candidate retrieval from the KG.** Querying all $\mathcal{O}(N^2)$ scene pairs quickly becomes intractable as $N$ grows. Instead, the agent first retrieves a sparse set of *candidate* scene pairs from the scene–entity graph. Two scenes are considered as candidates only if they share at least one entity node, such as a character, location, or salient prop. This simple structural prior already yields a substantial reduction in search space: for example, in *The Wandering Earth 2* (173 scenes), naive all-pairs checking would require $173 \times 172/2$ pairwise decisions, whereas our entity-based filtering produces on the order of 1,500–2,000 candidates. The agent can further tighten this set by incorporating additional priors such as semantic similarity between scene summaries and constraints on script-order distance (e.g., only considering pairs whose scene indices differ by at most $k$).

- **LLM-based assessment and hypothesis update.** For each candidate pair, the agent invokes a structured continuity prompt that conditions on scene metadata (including summary and CMP information) and asks whether the two scenes can share a production setup. The resulting binary decisions (with rationales) are used to update an evolving *continuity graph* over scenes, where edges represent hypothesized continuity relations.

- **Reflective chain-level refinement.** After accumulating pairwise judgments, the agent constructs preliminary continuity chains by extracting maximal cliques or connected components from the continuity graph. It then invokes a second LLM prompt—the *chain critic*—to reassess each chain holistically. The chain critic examines the entire sequence, identifies internal inconsistencies, over-extended clusters, or incompatible CMP metadata, and proposes merges, splits, or drops that refine the chain structure.

This iterative loop—KG-based candidate retrieval, LLM pairwise evaluation, and LLM chain-level reflection—constitutes the full continuity-checking agent. Visualization of the resulting continuity chains (Fig. 17) is a presentation layer built on top of these outputs rather than a step in the agent itself.

**Pairwise continuity prompt.**  We use the following full prompt to assess whether two scenes belong to the same production setup.

---

**Pairwise Production Continuity Judgment Prompt**

**Role:** You are a professional *script supervisor* evaluating whether two scenes share the same *production setup*.
**Definition (Production Continuity):**
Two scenes are production-continuous if they can reuse the same: - physical set and spatial layout,
- lighting baseline,
- dressing and props configuration,
- core costume/makeup baseline for key characters.
Narrative time jumps are allowed; the decision concerns *production design*, not story order.

---

**Scene 1**
**Name:** {scene_name1}
**Synopsis:** {summary1}
**Costume/props info:** {cmp_info1}

**Scene 2**
**Name:** {scene_name2}
**Synopsis:** {summary2}
**Costume/props info:** {cmp_info2}

**Shared entities (from KG):**
{common_neighbor_info}

**Decision Rules**

1. **Spatial Layout** — Same or compatible functional space ⇒ continuity.

2. **Costume/Props** — Minor variations acceptable; major shifts break continuity.

3. **Lighting/Mood** — Day/night differences may still be achieved on the same set.

4. **Ignore Narrative Time** — Only visual/production coherence matters.

**Output strictly in JSON:**

```
{
   "is_continuity": true | false,
   "reason": "One-sentence justification referencing set,
              lighting, dressing, and costume/props."
}
```

**LLM chain-reflection prompt.** After extracting initial chains, an agent-style reflection step evaluates whether the chain is globally coherent and suggests refinements.

Continuity Chain Critic Prompt

**Role:** You are a senior script supervisor reviewing an automatically grouped *continuity chain*. Your task is to judge whether the chain is visually coherent as a single production block.
**Scenes (ordered):** {scene_ids}

**Scene metadata:**
{per_scene_metadata}

**Evaluate:**

1. Spatial/set-layout coherence across all scene.

2. Lighting and mood compatibility

3. CMP (costume/makeup/props) consistency

4. Internal graph consistency

5. Potential redundancy or over-extension

**Output strictly in JSON:**

```
{
   "coherence_score": 0.0-1.0,
   "confidence: 0.0-1.0,
   "keep_decision": "keep" | "split" | "drop",
   "suggested_splits": [
     ["scene_A", "scene_B"],
     ["scene_C"]
   ],
   "rationale": "Short paragraph explaining the chain assessment."
}
```

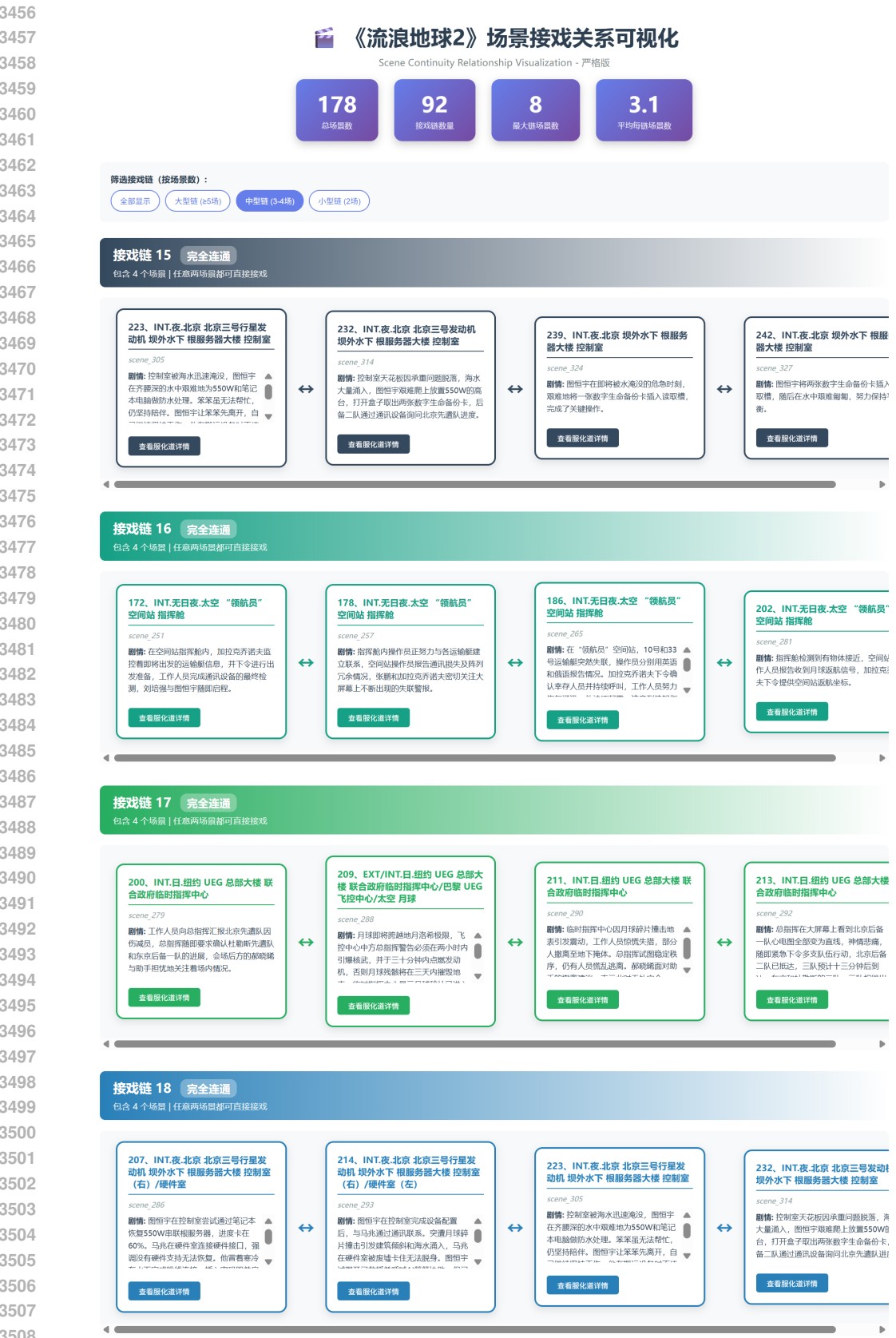

Figure 17: **Continuity chain visualization.** Chains appear as ordered sequences of scene cards annotated with summaries and CMP descriptors.

I.2 CHARACTER STATE TRACKING

This component derives fine-grained, per-scene character states from screenplay text. Whereas the main extraction pipeline (§3.2.1) focuses on entities, relations, and event-centric structure, the character-state module targets a different layer of narrative grounding: short, observable descriptions of what important characters are doing in each scene.

**Iterative extraction with reflection.** Following the reflection-driven framework used across our system, the agent performs a recurrent *extract → reflect → revise* cycle. For each scene, the system (1) retrieves candidate characters from the scene–entity graph, (2) extracts only *visible, non-speculative* states using a structured LLM prompt, and (3) applies a critic model that checks coverage, grounding, and precision. Feedback is reinjected into the next extraction pass until convergence or a retry limit is reached. This loop yields a stable, scene-aligned collection of character states across the entire screenplay.

**Observable state representation.** Each character state captures only what is externally observable—actions, expressions, posture, and tone—excluding psychological or symbolic interpretation. Outputs are stored in a minimal JSON structure listing only important characters for each scene.

**Visualization.** To support interactive exploration, we render the extracted dataset using an web interface featuring three complementary views.

1. TIMELINE SWIMLANE VIEW. A global *character–scene matrix* showing all characters (rows) across all scenes (columns). This view offers a high-level visualization of appearance frequency, co-occurrence structure, and participation patterns, enabling users to quickly identify which characters move together across the narrative.

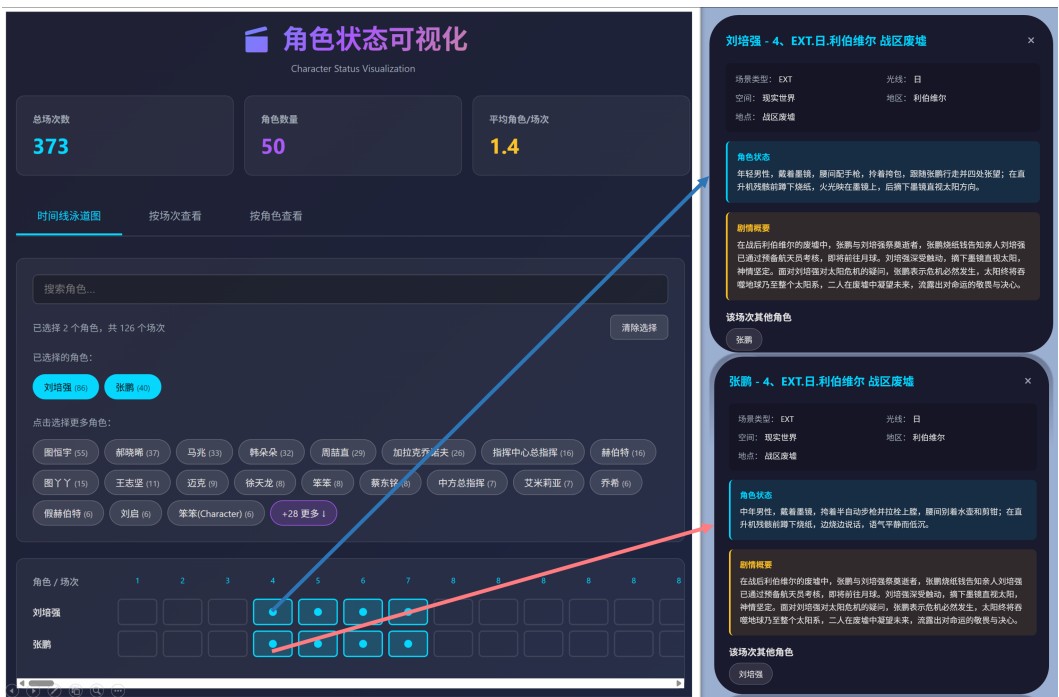

Figure 18: **Timeline Swimlane View.** A screenplay-wide character–scene matrix enabling global inspection of appearance patterns.

2. PER-SCENE VIEW. A chronological walk-through of the screenplay, presenting each scene as a structured card containing metadata (INT/EXT, lighting, spatial setting), a plot summary, and the

set of characters involved, each annotated with their extracted state. This view supports scene-level reasoning and narrative inspection.

3. PER-CHARACTER VIEW. A character-centric timeline showing all scenes in which a selected character appears, together with their extracted state and the surrounding narrative context. This view enables analysis of character trajectories and role progression across the story.

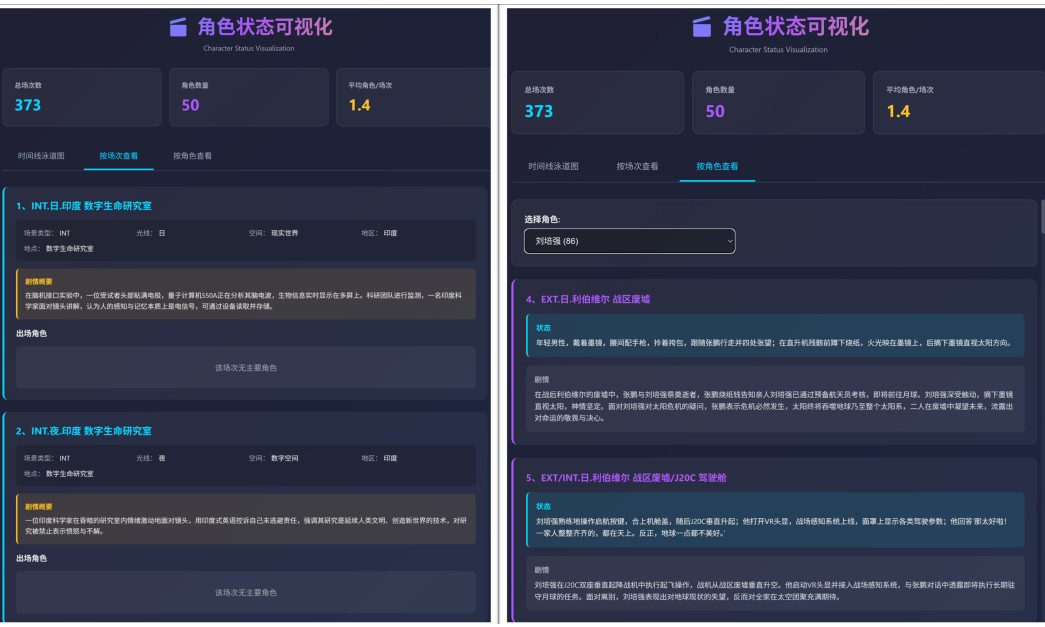

Figure 19: **Per-Scene View (left) and Per-Character View (right).** Two complementary detail views: scene-centric inspection and character-centric trajectory tracking.