# OpenReview forum: "Narrative Knowledge Weaver: A Multi-Agent Framework for Knowledge Graph Construction and Analysis from Complex Narratives"
_ICLR.cc/2026/Conference — ICLR 2026 Conference Desk Rejected Submission_

### Official Review · Reviewer_4hCa · 2025-10-29

**Soundness:** 3
**Presentation:** 3
**Contribution:** 3
**Rating:** 6
**Confidence:** 3

**Summary:**

this paper introduces Narrative Knowledge Weaver framework which embeds a multi-agent system for building and reasoning over knowledge graphs extracted from long-form narratives like novels or screenplays. It integrates reflection-driven extraction, adaptive schema induction, and event-centric graph refinement to create coherent, causal, and interpretable representations of stories. Experiments on NarrativeQA and screenplay benchmarks show its advantages in entity disambiguation, relation accuracy, and narrative question answering over baseline LLM and retrieval systems.

**Strengths:**

This paper tackles a genuinely hard and under-explored problem: how to turn long, messy narratives—novels, screenplays—into structured knowledge graphs that preserve coherence and causality. The multi-agent setup is a wise and original move, and the combination of reflection loops, adaptive schema induction, and event-centric refinement are also well designed.

Empirically, the work is thorough. The authors run solid comparisons on both public and custom datasets and include ablations that tell a story about which parts of the system matter. The results make sense—better recall, cleaner relations, and clear QA gains. The writing is clear and the figures help; it’s easy to follow the logic from motivation to experiments.

**Weaknesses:**

Some parts of the system—especially the Graph Probing Agent and reflection loops—feel more like engineering heuristics than rigorously analyzed methods.

The new “Practitioner Screenplay QA” benchmark is promising but too small and language-specific to fully support the generality claims. Results on causal reasoning could be more convincing with human evaluation.

The interaction between agents is a black box—some transparency about how reflection feedback is shared or reconciled would make the framework easier to reproduce.

**Questions:**

1. How well does the system scale to extremely long narratives (hundreds of thousands of tokens)?

2. Does the induced schema generalize across genres or narrative styles, or does it need retraining for each domain?

3. How effective is the SABER pruning compared to simpler cycle-breaking methods?

---

> ### Author Response · Authors · 2025-11-25
>
> **We sincerely thank the reviewer for the constructive and encouraging feedback. Below we address the noted weaknesses and questions.**
>
> ---
>
> ## 1. On engineering heuristics and transparency of agent interaction
> Thank you for raising this point. We substantially expanded the documentation of the multi-agent workflow. Appendix D now explains the internal coordination, including:
>
> - a clear division of labor among agents (Graph Probing, Extraction, Attribute, CMP);
> - the DynamicReflector shared memory storing extraction exemplars and higher-level insights;
> - a two-tier reflection framework showing how instance-level feedback aggregates into schema-level updates.
>
> These additions clarify how feedback is shared and reconciled, making the process transparent and reproducible.
>
> ---
>
> ## 2. On the size and scope of the Practitioner Screenplay QA benchmark
> We agree that the 303 QA items alone cannot fully support generality claims, although they were written by screenplay practitioners and reflect real production needs. Currently the film/TV domain lacks a suitable benchmark for narrative KG extraction and narrative QA, which makes systematic research difficult. Building such a benchmark is one of our ongoing efforts.
>
> To provide clearer context, Appendix F.1 and the supplementary README describe our large-scale benchmark initiative, covering KG extraction, disambiguation, attributes, and multi-step QA. We have already processed ~120 full screenplays, and all intermediate outputs—OCR-cleaned scripts, entity and event annotations, schema files, attributes, and cross-modal metadata—are included in the supplementary materials. While not yet a finalized benchmark, this resource forms the foundation for a future standalone benchmark paper.
>
> ---
>
> ## 3. On scaling to extremely long narratives
> Appendix G.1 presents end-to-end runtime, token consumption, and scaling behavior across screenplays of varying lengths (from 12,965 to 83,562 characters). Because extraction operates on discourse-coherent segments and refinement processes aggregated outputs rather than raw text, the pipeline scales approximately linearly with the number of segments. We observe no degradation in KG quality or QA accuracy as length increases. While we have not yet tested documents with hundreds of thousands of tokens, the architecture is designed for such inputs and does not rely on global context windows.
>
> Additionally, we added an error analysis of low-correctness questions (Appendix H.6) and an ablation on low-frequency tools (Appendix H.3), which help clarify where the system currently struggles and how different tool families contribute to overall performance.
>
>
> ## 4. On whether the induced schema generalizes across genres or narrative styles
> The core schema is not tied to screenplay formatting. Fundamental narrative entity types (characters, objects, locations, events) remain stable across genres. The Graph Probing Agent adapts the schema dynamically by aggregating feedback from many extraction instances, reducing the need for manual tuning and enabling extension or refinement when encountering new domains. As clarified in the revision, the fine-grained enrichment strategy—structured attributes, provenance alignment, and node-type–specific aggregation—is domain-agnostic and applies to long-form documents such as biographies, historical accounts, and multi-document narratives.
>
> ---
>
> ## 5. On the effectiveness of SABER compared to simpler cycle-breaking methods
> Appendix G.7 compares SABER with two structural baselines: DFS back-edge removal and a greedy maximum-confidence DAG construction. While both remove cycles, they often discard important causal edges or fail to resolve flattened structures. SABER integrates structural pruning with semantic adjudication within strongly connected components, allowing it to preserve coherent multi-hop causal progressions. As shown in Appendix G.7, SABER yields higher semantic cohesion within plot units and better correctness on event-sensitive QA.
>
> ---
>
> ## 6. Additional improvements made in the revision
>
> We also expanded the experimental comparison with GraphRAG, including pairwise LLM preference evaluation and aligned settings for both KG construction and QA. Appendix F.2–F.4 provide evaluator prompts and agreement analysis. The Related Work section and Section 3.2.4 clarify how our event-centric refinement differs conceptually from GraphRAG’s community-level augmentation.
>
> To illustrate how the resulting knowledge graphs support practical reasoning, we added two downstream demonstrations—production-continuity checking and character-state tracking (Appendix I). These examples show how the refined event and entity structures enable interpretable, provenance-aware narrative analysis beyond QA.
>
> ---
>
> Thank you again for the thoughtful feedback. It has greatly helped improve the clarity and presentation of the paper.

---

### Official Review · Reviewer_pXmA · 2025-10-31

**Soundness:** 3
**Presentation:** 1
**Contribution:** 2
**Rating:** 4
**Confidence:** 2

**Summary:**

This paper tackles the problem of improving LLM understanding of long form narratives (i.e. novels and screenplays). Unlike existing knowledge graph extraction techniques which struggle with modeling long-term dependencies, this work explores a multi-agent approach for decomposing narratives into easy to interpret knowledge graph structures. They rely on expert agents to extract entities and relations, finetune the extracted schema via self reflection. They also introduce pipelines to normalize/disambiguate the set of all entities and remove duplicates. Finally, they also propose an event-centric graph which can be used to model higher level relationships of plot elements. They test this method across knowledge graph creation and question-answering.

**Strengths:**

The proposed problem of answering questions about long-form narratives is interesting and would be important as LLMs may be used for more complex literary tasks. For that reason, I think that the problem is relatively well-motivated. Additionally, the paper presents a major implementation effort for improving performance on these tasks, drawing upon multi-agent framework. The use of multiple agents is promising for completing complex reasoning tasks using a large context. The provided results show significant improvement which validates their methodology. The results are particularly interesting as they evaluate on a dataset of screenplays -- going beyond existing benchmarks. They also perform some ablations of their proposed method, which validates the design choices they make. Overall, they describe their method and evaluation in careful detail allowing their results to be properly understood.

**Weaknesses:**

**Presentation and Clarity** In my opinion, this paper was slightly difficult to read. The bulk of the paper is concentrated on defining the implementation details of their method. While the thoroughness of their explanation is certainly commendable, I believe that the authors could better structure this and judiciously defer certain details to the appendix. Otherwise, it is quite difficult for a reader to parse which are the higher-level contributions/insights of their framework and what are lower-level design choices/implementation details. They could also do a better job of providing a high-level architecture of their framework which actually reflects the specific components that they mention in the remainder of the paper.

For example, in the beginning of Section 3, they define their architecture as:
> (i) Data Preprocessing, which handles metadata extraction and discourse-aware chunking; (ii) Multi-Agent Knowledge Processing, where specialized agents conduct schema induction, reflective extraction, and entity disambiguation; (iii) Hybrid Storage, integrating graph and vector stores; and (iv) the Application Layer

However, this does not match up with the (sub)sections that are actually present in the remainder of Section 3.  For example,  they have a > Section 3.3  EVENT-CENTRIC GRAPH REFINEMENT
which is not present in the initial summary of the architecture. Similarly, they never address their final two stages (Hybrid Storage and Application Layer) in Section 3. Given the complexity of their method, I believe that this makes the paper significantly harder to parse than necessary. Additionally, they at various points introduce multiple terms and use them inconsistently
> AGENT-BASED KNOWLEDGE EXTRACTION AND REFINEMENT
v.s.
> ITERATIVE KNOWLEDGE EXTRACTION WITH REFLECTION


**Generality of the Proposed Method**
Although the authors discuss the specific implementation details, it raises the question of how general the contributions of this work is. For example,  the Event Graph modules appears to be heavily engineered to stories/screenplays etc. However, would a similar module work on other types of texts (for example nonfiction). The proposed method seems over-engineered in some sense and evaluation of the paper would be improved by addressing in what ways the contributions of the paper are applicable more generally.

**Questions:**

Please respond to the points mentioned in the Weaknesses section.

---

> ### Author Response · Authors · 2025-11-25
>
> **We sincerely thank Reviewer pXmA for the thoughtful review. Your comments helped us improve the clarity, presentation, and framing of the paper. Below we address the points raised.**
>
> ---
>
> ## 1. Presentation and clarity of the overall framework
> We appreciate the feedback regarding Section 3’s organization and the mismatch between the initial architecture summary and the detailed subsections. In the revised version, Section 3 now directly aligns with the four-layer architecture shown in Figure 1:
>
> 1. **Data Preprocessing** (3.1)
> 2. **Multi-Agent Knowledge Extraction and Enhancement** (3.2)
> 3. **Hybrid Storage and Application Layer** (3.3)
>
> This restructuring ensures a direct correspondence between the conceptual layers and the detailed descriptions. We also added the new **Section 3.3 Hybrid Storage and Application Layer**, which explains the hybrid storage substrate (KG, vector store, relational tables), the unified `chunk_id`–based cross-indexing mechanism, and the tool interface used during inference.
>
> Section 3.2 was reorganized to present the knowledge extraction flow through four explicit modules: reflection-augmented extraction, schema induction via the Graph Probing Agent, entity normalization, and event-centric graph refinement. This modularization makes the pipeline easier to follow and distinguishes conceptual contributions from implementation steps. To further reduce cognitive load, lower-level details (prompt templates, tool formats, message schemas) were moved to the appendix, and terminology is now standardized throughout the section.
>
> ---
>
> ## 2. Additional experiments and contextualization of the method
> We also introduced several improvements that help place our system in context.
>
> - We added **GraphRAG** as a baseline for both KG construction and QA, using the same chunk size and backbone LLM for fairness.
> - We included a stronger retrieval baseline (BM25 + parent–child retriever + reranking).
> - We added a **pairwise LLM preference evaluation** following the protocol commonly used in GraphRAG-style work.
> - Additional experiments include comparisons of SABER with other cycle-breaking methods (Appendix G.7), ablations on low-frequency tools, and an error analysis of low-correctness questions.
>
> We also expanded the explanation of the **event-centric refinement module**. As detailed in the revised Related Work and Section 3.2.4, the refinement operates at the granularity of individual narrative events rather than node communities. Events are represented with structured attributes and fine-grained provenance and then adjudicated for temporal and causal consistency before forming the Event Plot Graph. This design supports reasoning patterns common in long-form narratives—tracking character-state evolution, causal developments, and scene-to-scene coherence—and provides interpretable structure for downstream tasks.
>
> ---
>
> ## 3. Generality beyond screenplays
>
> Thank you for raising the question of generality. Although our experiments use screenplays,
> the core components of the framework are not tied to screenplay-specific formatting. The
> modules for chunking, schema induction, reflection-augmented extraction, and entity
> normalization operate over general narrative or event-rich text without relying on screenplay
> conventions such as scene headings or dialogue structure.
>
> The revision also clarifies the generality of our enrichment strategy. Rather than performing
> community-level augmentation as in GraphRAG, our system enriches the graph at a finer semantic
> granularity. Entity nodes receive structured, provenance-grounded attributes, and different
> node types (characters, objects, events, locations) aggregate distinct information based on
> their narrative role. Events are modeled as first-class units with explicit temporal and causal
> attributes, refined before forming Event Plot Graphs. This entity- and event-level structure
> supports interpretable, provenance-aware reasoning that applies naturally to other long-form
> domains, including multi-document narratives, biographies, historical accounts, and other
> event-centric text collections.
>
> ---
>
> ## 4. Downstream demonstrations
> To illustrate how the refined narrative graph can be used beyond QA, we added two production-level demos in **Appendix I**:
>
> - **Production-continuity checking**, using causal and spatial links between scenes.
> - **Character-state tracking**, using structured event representations to trace how attributes evolve across scenes.
>
> These examples demonstrate how the resulting narrative graphs support practical, interpretable reasoning tasks.
>
> ---
>
> As this is my first time submitting to ICLR, some parts of the earlier draft were not presented as clearly as they should have been. I sincerely apologize for the difficulty this may have caused in reading, and I have made substantial effort to improve the structure and clarity in the revision. If you have any further suggestions, I would be very grateful to hear them.

---

### Official Review · Reviewer_FG1o · 2025-10-31

**Soundness:** 3
**Presentation:** 2
**Contribution:** 2
**Rating:** 4
**Confidence:** 3

**Summary:**

This paper proposes a multi-agent framework for extracting and reasoning over knowledge from long-form narratives. The system converts narrative texts (e.g., novels or screenplays) into structured knowledge graphs and event plots through a reflection-based, multi-stage pipeline including schema probing, entity normalization, event refinement, and the SABER algorithm for event linking.

**Strengths:**

1. Comprehensive system design: The multi-agent framework with specialized roles (Graph Probing, Extraction, Disambiguation, Query agents) is well-architected and addresses different aspects of narrative understanding systematically.
2. Novel contributions for narrative-specific challenges: The Event Plot Graph (EPG) construction and SABER pruning algorithm represent interesting approaches to handling causal relationships and plot structures in narratives.
3. New benchmark contribution: The Practitioner Screenplay QA benchmark with domain expert annotations adds value to the community.

**Weaknesses:**

1. While the system integration is comprehensive, individual components largely combine existing techniques (reflection loops, multi-agent coordination, RAG). The core innovations (SABER, EPG) are relatively incremental.
2. The large gains are primarily on the authors' own benchmark, raising concerns about generalizability.
3. Comparisons are against zero-shot LLM and EDC (2024), but no comparison with recent strong systems like GraphRAG, KAG, or other 2025 methods mentioned in related work.
4. The pipeline involves numerous LLM calls with reflection loops and multiple agents. No analysis of computational costs, latency, or scalability is provided.

**Questions:**

1. What is the total computational cost (# of LLM calls, processing time) for a typical screenplay? How does this scale with document length?
2. If possible to provide comparisons with GraphRAG, KAG, or other recent (2024-2025) knowledge graph construction methods?
3. How was the 89.8% score on the Screenplay QA benchmark judged? What LLM-judge prompt and inter-annotator agreement (IAA) were used? Have you validated the judge model with human evaluation or cross-model verification?

---

> ### Author Response · Authors · 2025-11-25
>
> **We thank Reviewer FG1o for the detailed and constructive review. Your comments helped us clarify several aspects of our system, strengthen the comparisons, and improve the presentation. Below we address each point in order.**
>
> ---
>
> ## 1. Computational cost and scalability
> We agree that understanding the computational footprint of a multi-agent pipeline is important.
> We added a dedicated analysis in **Appendix G.1** reporting end-to-end runtime, token consumption, and scaling behavior across screenplays of different lengths. The section breaks down the cost of chunking, extraction, refinement, and graph construction, giving a clearer view of where computation is spent and how the pipeline scales with document size.
>
> ---
>
> ## 2. Stronger baselines and clarification of narrative-specific design choices
> Following the reviewer’s suggestion, we expanded comparisons to include more recent methods. The updated experiments add **GraphRAG** as both a KG-construction and QA baseline, and we strengthened retrieval baselines with a **BM25 + parent–child retriever + reranking** pipeline. All systems use the same chunk size and backbone LLM for fairness. We also added a **pairwise LLM preference evaluation** following the protocol used in GraphRAG-style work to provide a complementary qualitative comparison.
>
> We also clarified our narrative-specific design motivations. As expanded in Related Work and Section 3.2.4, our event-centric refinement focuses on individual narrative events rather than node communities. Events are converted into structured event cards with explicit attributes and fine-grained provenance, then adjudicated for temporal and causal consistency before forming the Event Plot Graph. In contrast to community-level augmentation, this design treats events as first-class narrative units, aligning the representation with reasoning patterns common in long-form narratives such as character-state evolution, causal progression, and scene-to-scene coherence.
>
> ---
>
> ## 3. QA evaluation protocol and judge reliability
> Thank you for noting the previous omission. We restored and expanded the evaluation description:
>
> - The main Experimental Setup now states that each question is answered multiple times with independent stochastic decoding, and we report both majority-vote correctness and per-sample correctness.
> - **Appendix F.2–F.3** now include the full LLM-judge prompts.
> - **Appendix F.4** reports evaluator agreement across repeated judgments.
>
> This provides full transparency for the QA evaluation process.
>
> ---
>
> ## 4. Benchmark generalizability and availability
> We added clarifications regarding dataset availability in both the main text and the supplement.
>
> - The supplementary package includes all materials used in this paper, including the screenplay segments, structured annotations, and the 303 QA pairs used for evaluation.
> - We also provide the scripts used for LLM evaluation and report generation.
> - In addition, the supplement contains the intermediate outputs for a much larger collection of screenplays (~120), including OCR-cleaned scripts, entity and event annotations, schema files, attributes, and cross-modal metadata. The accompanying README outlines our ongoing effort to turn this into a comprehensive benchmark for narrative KG extraction, disambiguation, attribute extraction, and multi-step QA.
> - The full benchmark will be released upon publication once anonymization and copyright clearance are completed (hopefully early next year）.
>
>
> ---
>
> ## 5. Demonstrations of downstream applications
> To illustrate the practical value of producing interpretable, provenance-aware narrative graphs, we added two demos in **Appendix I**:
>
> - **Production-continuity checking**, using scene-level causal and spatial structure.
> - **Character-state tracking**, using structured event cards and refined plot structure.
>
> These examples show how the resulting narrative graphs support realistic production-level tasks requiring explicit and human-interpretable reasoning traces.
>
> ---
>
> **We thank the reviewer again for the constructive feedback, which helped improve the clarity and completeness of the work.**

---

> > ### Author Response · Authors · 2025-11-30
> >
> > 6. Clarification on the contribution beyond community-level graph augmentation
> >
> > We thank the reviewer for raising the question about whether the contributions are incremental.
> > To clarify, while our system indeed draws upon established paradigms such as reflection and
> > agent-based coordination, two core components—Adaptive Attribute Enrichment and
> > Event-Centric Graph Refinement—address limitations that community-level graph augmentation
> > (e.g., GraphRAG) is not designed to handle.
> >
> > As discussed in Section 2 and the revised Related Work, our framework focuses on
> > fine-granularity enrichment rather than community-level summarization. For entity nodes, the
> > attribute enrichment module maintains structured, provenance-grounded attributes that evolve
> > as new evidence is encountered; the attribute set can expand, contract, or reorganize
> > based on entity-specific signals, enabling detailed character-state and object-state tracking
> > that cannot be captured through community summaries.
> >
> > Similarly, our event-centric refinement treats events as first-class narrative units. Each event
> > is represented as a structured event card with fine-grained provenance, then adjudicated for
> > temporal and causal consistency before being integrated into the Event Plot Graph (EPG).
> > This event-level adjudication enables explicit, multi-hop causal reasoning—such as scene-to-scene
> > progression and character-arc modeling—which community-level graph organization does not
> > directly support.
> >
> > In this sense, the contributions are not incremental extensions of existing RAG pipelines, but
> > rather introduce narrative-specific graph representations that support forms of temporal, causal,
> > and interpretability-focused reasoning that prior community-based methods are not equipped to
> > provide.

---

### Official Review · Reviewer_spCK · 2025-11-04

**Soundness:** 1
**Presentation:** 2
**Contribution:** 2
**Rating:** 4
**Confidence:** 4

**Summary:**

The authors introduce "Narrative Knowledge Weaver," a multi-agent framework designed to construct knowledge graphs from long-form narratives such as screenplays and novels. The system consists of four main components: a data preprocessing layer that performs discourse-aware text chunking and metadata extraction; a multi-agent knowledge processing layer with specialized agents for schema induction, entity/relation extraction with reflection loops, and entity disambiguation; a hybrid storage layer combining graph and vector databases; and an application layer for question answering. The authors include a Graph Probing Agent that induces narrative-specific schemas, a reflection-driven extraction mechanism where agents iteratively refine their outputs, and an Event Plot Graph construction pipeline that uses their SABER algorithm to prune causal relationships and promote event chains into higher-level plot structures.
The authors evaluate their system on two benchmarks, a subset of NarrativeQA and a new proposed benchmark consisting of 5 Chinese screenplays. They further provide ablation results over the graph probing schema-induction step and the Event-centric Graph Refinement module.

**Strengths:**

I do think there’s alot of things to like about this paper:

- I like the idea of turning narrative conceptual extraction into a multi-agent task. This makes sense and is well motivated.
- I also appreciate the level of detail about the system in both the main paper and the appendix (I have some other thoughts about this too that I will elaborate on in ‘weaknesses’).
- SABER is an interesting approach for pruning cycles in causal graphs in leveraging an LLM to identify which connections are more ‘narratively’ relevant and should be preserved vs less important connections.
- The framework and all of its parts have some sort of evaluation at each step
- The Practitioner Screenplay QA Benchmark, but this is also not so good for its own reasons (see weaknesses)

**Weaknesses:**

Weaknesses:
- The actual model used for evaluation isn’t present in the main paper. This gets referenced in the appendix (Qwen3-235B-A22B-FP8) but this is definitely not something that should be only found in the appendix.
- While I like the idea and motivation behind the Practitioner Screenplay QA Benchmark, I have mixed feelings about this not being released alongside the paper. As is, the results within the paper are not reproducible.
- Lack of statistical rigor in that it appears the results reported are only done over 1 seed (unless I missed this somewhere) but given the complexity of the system this is more of a ‘not having this isnt a demerit but having it would have been better’.
- No real investigation about where this system fails and why
- The authors dont evaluate against other knowledge graph construction methods, despite citing them. I would’ve liked to see at least one or two other baselines rather than just zero-shot and EDC.
- Related to the above: the model for the zero-shot experiment is not mentioned, and if it is the same as the underlying system, I would have liked to see some other closed-source (or just another model) evaluation.

**Questions:**

- The tool usage analysis in the appendix (i really like this but understand why it wasnt included in main paper) shows many tools are used less than 2% of the time. Was there any investigation as to what happened if these tools were removed?
- What model was used for the zero-shot baseline?
- Will the Practitioner Screenplay QA be fully released
- Why not include other KG construction methods as baselines, such as those in the related works?
- Where does the system fail and why?
- Does performance change across genre? Does the length of the input (ie, a short novel vs something like the entire Lord of the Rings trilogy) have a significant impact on performance?

---

> ### Author Response · Authors · 2025-11-25
>
> **We sincerely thank Reviewer spCK for the thoughtful and constructive feedback.
> As first-time ICLR submitters, we learned greatly from your comments, and they directly motivated substantial revisions that strengthened both the clarity and rigor of the paper. Below we address each point in turn.**
>
> ---
>
> ## **1. Model specification moved to the main paper**
> We fully agree that evaluation settings should not appear only in the appendix (Sorry, I shall not make this mistake >..<).
> We have moved all model details—including the backbone LLM and chunking configuration—into the main Experimental Setup section to ensure clarity and reproducibility.
>
> ---
>
> ## **2. Reproducibility of the Practitioner Screenplay QA benchmark**
> Thank you for pointing this out. We have clarified the current status of the benchmark in the revision.
> - All materials used in the experiments—including screenplay segments, structured annotations, and the 303 QA pairs—are provided in the supplementary package.
> - In addition, the supplementary materials include the intermediate outputs for a substantially larger set of screenplays (~120), such as OCR-cleaned scripts, entity and event annotations, schema files, attributes, and cross-modal metadata. These constitute the in-progress version of a larger benchmark we are building.
> - The full benchmark will be released upon publication once anonymization and copyright verification are complete.
>
> This makes the current experiments fully reproducible while also giving a preview of the broader benchmark under development.
>
> ---
>
> ## **3. Expanded baselines for KG construction and QA**
> We strengthened comparisons in several ways:
>
> - The main paper now includes **multiple KG construction baselines**, including zero-shot extraction, a multi-stage extraction pipeline (EDC), and a graph-induced baseline (GraphRAG).
> - For QA, we evaluate against both a strong retrieval system and a graph-based QA pipeline.
> - We additionally perform **pairwise LLM preference evaluation**, following GraphRAG, to assess qualitative answer properties.
>
> ---
>
> ## **4. Tool-usage analysis and ablations**
> You correctly noted that several tools appear infrequently. In the revised paper:
>
> - Tool usage statistics now appear in the main text.
> - We added an ablation (Appendix H.3) analyzing *low-frequency tools*.
>   This study shows that while many tools are rarely triggered, some of them enable capabilities—particularly cross-indexing and screenplay-specific metadata access—that are not easily replaced by other operators.
>
> Thus, some of their low usage reflects specialization rather than redundancy.
>
> ---
>
> ## **5. Failure modes and error analysis**
> We added a dedicated section (Appendix H.6) discussing:
>
> - where the system tends to fail,
> - typical error sources (e.g., entity ambiguity, long-distance causal links),
> - and how these failures propagate through the pipeline.
>
> This directly addresses the question *“Where does the system fail and why?”*.
>
> ---
>
> ## **6. Input length and genre variation**
> While our dataset does not yet support fine-grained genre-level statistical comparisons, we added two forms of analysis:
>
> - **Scaling behavior** (Appendix G.1) showing how the pipeline behaves on screenplays of very different lengths, with stable extraction quality and QA performance.
> - **Genre diversity**: the Practitioner benchmark already spans multiple genres (romance, comedy, martial arts, sci-fi, drama), and we observe that the system behaves consistently across these different styles.
>
> ---
>
> ## **7. Additional analyses and demonstrations of applications**
>
> **(a) SABER validation (Appendix G.7).**
> We also added two standard structural cycle-breaking baselines in Appendix G.7 to contextualize SABER. These baselines operate purely on graph structure, whereas SABER incorporates both structural and semantic cues. .
>
> **(b) Demonstrations of downstream applications (Appendix I).**
> To further illustrate the practical value of producing readable and interpretable narrative knowledge graphs, we added two application-level demos in the appendix: a **production-continuity checker** and **character-state tracking**. These examples demonstrate that the structured event representations and plot-level organization produced by our framework can support realistic, production-level narrative tasks that require explicit, human-interpretable reasoning traces.
>
> ---
>
> Once again, we sincerely thank your for all those detailed and constructing comments.
> Your feedback substantially improved the clarity and completeness of the work—from baseline expansion, to ablation design, to dataset release statements and failure-mode analysis. As early-career researchers submitting to ICLR for the first time, we truly appreciate the depth and care of your review, and it has already influenced how we think about future work on narrative understanding and benchmark design.

---

> > ### Comment · Reviewer_spCK · 2025-11-25
> > **Acknowledgement of rebuttal**
> >
> > Hi
> >
> > Just wanted to give an explicit acknowledgement that I have read through your rebuttal and will go through the revised manuscript with the responses in mind and adjust my score accordingly as needed afterwards. Will reach out with any further questions/comments that come up.

---

### Author Response · Authors · 2025-11-30
**Author Summary for the Area Chair**

Dear Area Chair,

Thank you for considering our submission. Below we concisely highlight
(1) the core contributions of the paper, (2) how the revision addresses all reviewer concerns,
and (3) the added demo of downstream utilities

---

## 1. Summary of Contributions (Clarified in the Revision)

Our goal is to construct coherent, interpretable, and provenance-aware knowledge graphs from
long-form narratives—where entities, attributes, events, and causal relationships evolve across
hundreds of scenes—so that downstream reasoning tasks such as multi-hop QA, temporal/causal
analysis, scene continuity checking, and character-state tracking can operate over
structured and queryable representations rather than raw text.

The revised submission clarifies the following contributions:

**(a) Reflection-augmented multi-agent extraction.**
A two-tier reflection mechanism (instance-level and schema-level), mediated by a shared
DynamicReflector memory, evaluates extraction quality, aggregates feedback, and stabilizes
entity/relation predictions across thousands of narrative segments.

**(b) Hybrid schema design with selective refinement.**
The system begins with a predefined core schema covering fundamental narrative entities
(characters, events, objects & etc.).
The Graph Probing Agent then proposes *selective* schema refinements or extensions when
persistent reflection feedback reveals systematic gaps.
This hybrid design avoids handcrafting while preventing uncontrolled schema drift.

**(c) Adaptive Attribute Enrichment (clarified placement).**
The module performs degree-aware, multi-granular attribute enrichment with reflection-guided
revision and schema-aware consolidation.
Unlike GraphRAG-style community-level summarization, our system (Adaptive Attribute Enrichment + Event-centric refinement) focuses on **finer-granularity
enrichment at the level of individual entities**, enabling structured attributes, provenance
alignment, and narrative-specific details such as character motivations, object states, and
event properties.

**(d) Event-centric refinement and Event Plot Graphs (EPGs).**
Events are converted into structured event cards with explicit attributes and fine-grained
provenance. Temporal and causal links are adjudicated before forming coherent Event Plot Graphs,
supporting multi-hop causal reasoning and scene-to-scene narrative consistency.

**(e) Benchmark contribution under a domain with scarce evaluation resources.**
Narrative understanding suffers from a scarcity of high-quality evaluation datasets, especially
in screenwriting and long-form storytelling where existing benchmarks (e.g., NarrativeQA) lack
scene structure and practitioner-authored questions.
To support reproducibility and future research, the supplementary package includes:
- all screenplay segments used in our experiments,
- structured annotations and the full set of 303 QA pairs,
- and intermediate outputs for ~120 screenplays (schema files, entity/event annotations, cross-modal metadata).
This provides a reproducible evaluation setting while also addressing an unmet need for
narrative-focused resources.

**(f) Demonstrations of downstream utility.**
The revision adds two downstream applications—continuity checking and character-state tracking—
showing how the resulting knowledge graphs enable explicit, interpretable, production-oriented
reasoning tasks.

---

## 2. How the Revision Addresses Reviewer Concerns

### **Reproducibility & Experimental Setup**
- All model details (backbone LLM, chunking configuration) are now included
  in the main Experimental Setup.
- All materials used in experiments (scripts, annotations, QA pairs) are fully included in the
  supplementary files.

### **Baselines & Comparisons**
- Added GraphRAG as both a KG-construction and QA baseline, with matched chunking and LLM.
- Added a pairwise LLM preference evaluation following prior work.

### **Ablations & Analysis**
- Ablations added for schema probing, event-centric refinement, low-frequency tools, and the adaptive attribute enrichment module.
- Added a failure-mode analysis (Appendix H.6).

### **Scalability & Computational Cost**
- Appendix G.1 now reports end-to-end runtime, token consumption, and stage-level breakdown
  across screenplays of varying lengths (12k–83k characters).

### **Schema Generality & Domain Applicability**
- Clarified that the hybrid schema design and enrichment mechanisms are domain-agnostic and
  applicable to other long-form or event-rich texts (biographies, historical accounts,
  multi-document narratives).

### **Presentation & Structural Clarity**
- Section 3 reorganized to match the four-layer architecture in Figure 1.
- Added a table of contents for appendix for better clarity and readability.

---

We sincerely thank the reviewers and the Area Chair for their time and thoughtful feedback,
especially under the unusual circumstances this year.

---

### Note · Program_Chairs · 2026-01-17
**Submission Desk Rejected by Program Chairs**

The following references in this submission do not refer to real documents and/or have major errors in bibliographic information:

 Arie Cattan, Max Vauthier, Yangfeng Ji, Dan Klein, and Ido Dagan. A benchmark for crossdocument event coreference. In Proceedings of EMNLP, pp. 4908-4920, Online, 2020. Association for Computational Linguistics. URL https://aclanthology.org/2020. emnlp-main. 398 .
Luyang Huang, Yufan Chen, and Yue Zhang. Narrativeqa revisited: Benchmarking consistency in story understanding. In Proceedings of ACL, 2023.